# Limited oxygen in standard cell culture alters metabolism and function of differentiated cells

Joycelyn Tan [1,16], Sam Virtue [1,16✉], Dougall M Norris[1,16], Olivia J Conway[1], Ming Yang[2,3], Guillaume Bidault[1], Christopher Gribben[4], Fatima Lugtu [4], Ioannis Kamzolas [1,5], James R Krycer [6,7] Richard J Mills[6,7], Lu Liang[1], Conceição Pereira [8], Martin Dale[1], Amber S Shun-Shion[1], Harry JM Baird [1], James A Horscroft[9], Alice P Sowton [9], Marcella Ma [1], Stefania Carobbio [1,10], Evangelia Petsalaki [5], Andrew J Murray [9], David C Gershlick [8], James A Nathan [11], James E Hudson[6,7,12], Ludovic Vallier [4], Kelsey H Fisher-Wellman [13,14,15], Christian Frezza [2,3], Antonio Vidal-Puig [1,10✉] & Daniel J Fazakerley [1✉]

## Abstract

The in vitro oxygen microenvironment profoundly affects the capacity of cell cultures to model physiological and pathophysiological states. Cell culture is often considered to be hyperoxic, but pericellular oxygen levels, which are affected by oxygen diffusivity and consumption, are rarely reported. Here, we provide evidence that several cell types in culture actually experience local hypoxia, with important implications for cell metabolism and function. We focused initially on adipocytes, as adipose tissue hypoxia is frequently observed in obesity and precedes diminished adipocyte function. Under standard conditions, cultured adipocytes are highly glycolytic and exhibit a transcriptional profile indicative of physiological hypoxia. Increasing pericellular oxygen diverted glucose flux toward mitochondria, lowered HIF1α activity, and resulted in widespread transcriptional rewiring. Functionally, adipocytes increased adipokine secretion and sensitivity to insulin and lipolytic stimuli, recapitulating a healthier adipocyte model. The functional benefits of increasing pericellular oxygen were also observed in macrophages, hPSC-derived hepatocytes and cardiac organoids. Our findings demonstrate that oxygen is limiting in many terminally-differentiated cell types, and that considering pericellular oxygen improves the quality, reproducibility and translatability of culture models.

**Keywords** Cell Culture; Oxygen Tension; Hypoxia; Metabolism/Adipocytes

**Subject Categories** Metabolism; Methods & Resources

## Introduction

Recently, there have been increasing efforts to improve the fidelity of cell culture models by optimising the nutrient composition of cell culture media to be more physiological (Vande Voorde et al, 2019; Cantor et al, 2017). However, the importance of oxygen tension in cell culture has not received the same attention. Cell culture is usually assumed to be hyperoxic (Ast and Mootha, 2019), due to incubators containing ~18% oxygen (Wenger et al, 2015). As a result, most work studying oxygen tension in cell culture reduces atmospheric oxygen levels to 5% or even 1% to limit oxygen availability. However, the notion that cell culture is hyperoxic relies on the assumption that oxygen delivery through the medium exceeds cellular demand, but this may not always be the case.

First, under 'standard culture conditions' (e.g. 1 mL of medium per well of a 12-well plate) the depth of medium varies between 2.4 and 2.9 mm due to the meniscus effect. Most mammalian cells range between 40 and 100 μm in diameter, such that oxygen must diffuse over a distance equivalent to between 27 and 58 cell diameters to reach the cell monolayer. Most cells in vivo are directly subtended by capillaries which deliver oxygen over significantly shorter diffusion

[1]Metabolic Research Laboratories, Wellcome-Medical Research Council Institute of Metabolic Science, University of Cambridge, Cambridge CB2 0QQ, UK. [2]MRC Cancer Unit, University of Cambridge, Cambridge Biomedical Campus, Cambridge CB2 0XZ, UK. [3]CECAD Research Center, Faculty of Medicine, University Hospital Cologne, Cologne 50931, Germany. [4]Wellcome-MRC Cambridge Stem Cell Institute, University of Cambridge, Cambridge CB2 0AW, UK. [5]European Molecular Biology Laboratory, European Bioinformatics Institute, Wellcome Genome Campus, Hinxton CB10 1SD, UK. [6]QIMR Berghofer Medical Research Institute, Brisbane, Queensland 4006, Australia. [7]Faculty of Health, School of Biomedical Sciences, Queensland University of Technology, Brisbane, Queensland 4000, Australia. [8]Cambridge Institute for Medical Research, University of Cambridge, Cambridge CB2 0XY, UK. [9]Department of Physiology, Development and Neuroscience, University of Cambridge, Cambridge CB2 3EL, UK. [10]Centro de Investigacion Principe Felipe, Valencia 46012, Spain. [11]Cambridge Institute of Therapeutic Immunology and Infectious Disease (CITIID), Jeffrey Cheah Biomedical Centre, Department of Medicine, University of Cambridge, Cambridge CB2 0AW, UK. [12]Faculty of Medicine, School of Biomedical Sciences, The University of Queensland, Brisbane, QLD 4072, Australia. [13]Department of Physiology, Brody School of Medicine, East Carolina University, Greenville, NC 27834, USA. [14]East Carolina Diabetes and Obesity Institute, East Carolina University, Greenville, NC 27834, USA. [15]UNC Lineberger Comprehensive Cancer Center, University of North Carolina at Chapel Hill School of Medicine, Chapel Hill, NC 27599, USA. [16]These authors contributed equally: Joycelyn Tan, Sam Virtue, Dougall M Norris. ✉E-mail: sv234@cantab.ac.uk; ajv22@medschl.cam.ac.uk; djf72@medschl.cam.ac.uk

distances (microns) than the medium column (millimetres). Thus, standard culture conditions pose a significant barrier to oxygen delivery. Second, cultured cells have a large range of oxygen consumption rates (Place et al, 2017). If these exceed the rate of oxygen delivery from the air-medium interface, cells will experience hypoxia at the cell monolayer (Keeley and Mann, 2019; Wenger et al, 2015). While the concept that cellular respiration can outstrip oxygen delivery in cell culture is known (Stevens, 1965; Metzen et al, 1995; Boag, 1970), the functional consequences of limiting oxygen under standard culture conditions on cell metabolism, gene expression and function has not been comprehensively addressed. This is an important gap in knowledge since pericellular oxygen, which is influenced by both cellular oxygen consumption and factors affecting delivery to cells (e.g. cell density, medium depth), are currently often not reported in literature and yet may substantially impact experimental outcomes. Understanding the impact of limiting oxygen in standard cell culture on cell phenotypes may pave the way for greater consideration of these factors in experimental design and method reporting.

Many cultured cell lines exhibit high rates of glycolysis, generating lactate from glucose to meet their ATP demands (Krycer et al, 2020; Henry et al, 2011). This glycolytic metabolism is often attributed to an intrinsic metabolic rewiring of cultured cells, similar to the aerobic glycolysis/Warburg effect observed in cancer cells (Warburg et al, 1927). Indeed, a high rate of glycolysis is a key metabolic adaptation of proliferating cells (Vander Heiden et al, 2009; Luengo et al, 2021; Han and Simon, 2022; Liberti and Locasale, 2016). However, this reasoning does not hold for the high rates of glycolysis observed in many terminally differentiated non-proliferative cell types in culture (Krycer et al, 2020; Gilglioni et al, 2018; Gstraunthaler et al, 1999). One possible explanation as to why glycolysis appears almost ubiquitous in cell culture is that unphysiologically high medium glucose promotes glycolysis, also referred to as the Crabtree effect (Crabtree, 1929). However, another arguably simpler explanation for the high glycolytic rates of many cultured cells is that oxygen is limiting, with cells responding to hypoxia by switching to anaerobic metabolism. While data regarding the glucose and lactate metabolism by many terminally differentiated cell types is consistent with the hypothesis that these cells are hypoxic, it remains to be formally proven and, perhaps more importantly, the impact of increasing oxygen supply on cellular phenotypes is largely unexplored.

To test whether oxygen can be limiting in standard cell culture, and the functional significance of this, we turned to cultured adipocytes. These are a valuable model for studying multiple aspects of adipocyte biology, including adipocyte differentiation (Fu et al, 2005; Chen et al, 2005; Ren et al, 2002), lipid metabolism (Green et al, 2016; Fernández-Galilea et al, 2012), and hormonal responses (Tan et al, 2015). Importantly, adipocyte cell lines exhibit substantially greater lactate production than adipocytes in vivo (Krycer et al, 2020; Hodson et al, 2013). Further, oxygen availability in adipose tissue is decreased in obesity (Trayhurn et al, 2008; Cifarelli et al, 2020), leading to activation of the hypoxia-inducible factor (HIF) 1α pathway (Todorčević et al, 2021; Pasarica et al, 2009). Since adipose tissue hypoxia is associated with diminished adipocyte function (differentiation, adipokine secretion, lipid metabolism, insulin action, and inflammation) (Yin et al, 2009; Halberg et al, 2009; Hosogai et al, 2007; Anvari and Bellas, 2021;

Trayhurn, 2013), these cells allowed us to assess the functional impact of limited oxygen in cell culture.

Here, we provide experimental evidence that the high glycolytic activity of cultured adipocytes results from hypoxia due to limitations in oxygen diffusion through the medium column. We demonstrate that increasing oxygen availability reprogrammed cells away from glycolytic and toward oxidative metabolism, increased pyruvate dehydrogenase (PDH) flux, tricarboxylic acid (TCA) cycle activity and oxygen consumption, promoted lipid anabolism, destabilised HIF1α, and induced over 3000 transcriptional changes within 16 h. Providing more oxygen also improved key adipocyte functions, including adipokine secretion, lipolysis and insulin responses, suggesting that hypoxia in standard cell culture compromises how faithfully cultured adipocytes model healthy adipose tissue. Critically, these changes in metabolism and functional improvements were validated in several additional terminally differentiated metabolic cell types, such as brown fat cells, myotubes, macrophages, hepatocytes, and cardiac organoids. Our findings highlight that understanding and controlling oxygen availability is vital to more accurately translate in vitro models to their in vivo counterparts, as well as to increase the reproducibility of cell culture experiments.

# Results

## Oxygen is limiting for adipocyte respiration under standard culture conditions

To study whether oxygen is limiting in standard cell culture, we first sought to investigate oxygen consumption and pericellular oxygen levels in terminally differentiated cells. The first cell line we used was 3T3-L1 adipocytes, which are differentiated from fibroblast precursors (Green and Kehinde, 1975) and have been a workhorse cell line for studying a range of adipocyte biology (Fu et al, 2005; Chen et al, 2005; Ren et al, 2002; Green et al, 2016; Tan et al, 2015). These cells have a high oxygen consumption rate (OCR) (200 fmol/mm$^2$/s) (Fig. 1A,B) and are highly glycolytic (Krycer et al, 2020). However, whether their high OCR exceeds the diffusion rate of oxygen through the medium column and what impact this has on cellular functions is unestablished.

Fick's first law of gas diffusion (Fig. 1C) states that oxygen diffusion rates are proportional to medium depth. To support an OCR of 200 fmol/mm$^2$/s we calculated that medium depth could not exceed 2.43 mm (e.g. a 5 mm medium depth would limit OCR to just 100 fmol/mm$^2$/s). Medium height is not constant in cell culture wells due to the meniscus, but the minimum depth of medium in a 12-well plate containing 1 mL of medium was measured at ~2.4 mm (Figs. 1C and EV1A). Based on the OCR of 3T3-L1s, Fick's law predicted that oxygen concentrations at the cell monolayer would be virtually 0 μM (Fig. 1C). Indeed, under standard culture conditions (100 μL per well in a 96-well plate), the oxygen concentration measured 0.55 mm above the cell monolayer (the minimum measurable depth by the Resipher system) was ~15 μM (11 mmHg) (Figs. 1D and EV1A), well below incubator oxygen concentrations (181 μM; 140 mmHg) (Place et al, 2017; Wenger et al, 2015). However, in a closed system, 3T3-L1 adipocytes respired maximally down to ~4 μM oxygen (Fig. 1E).

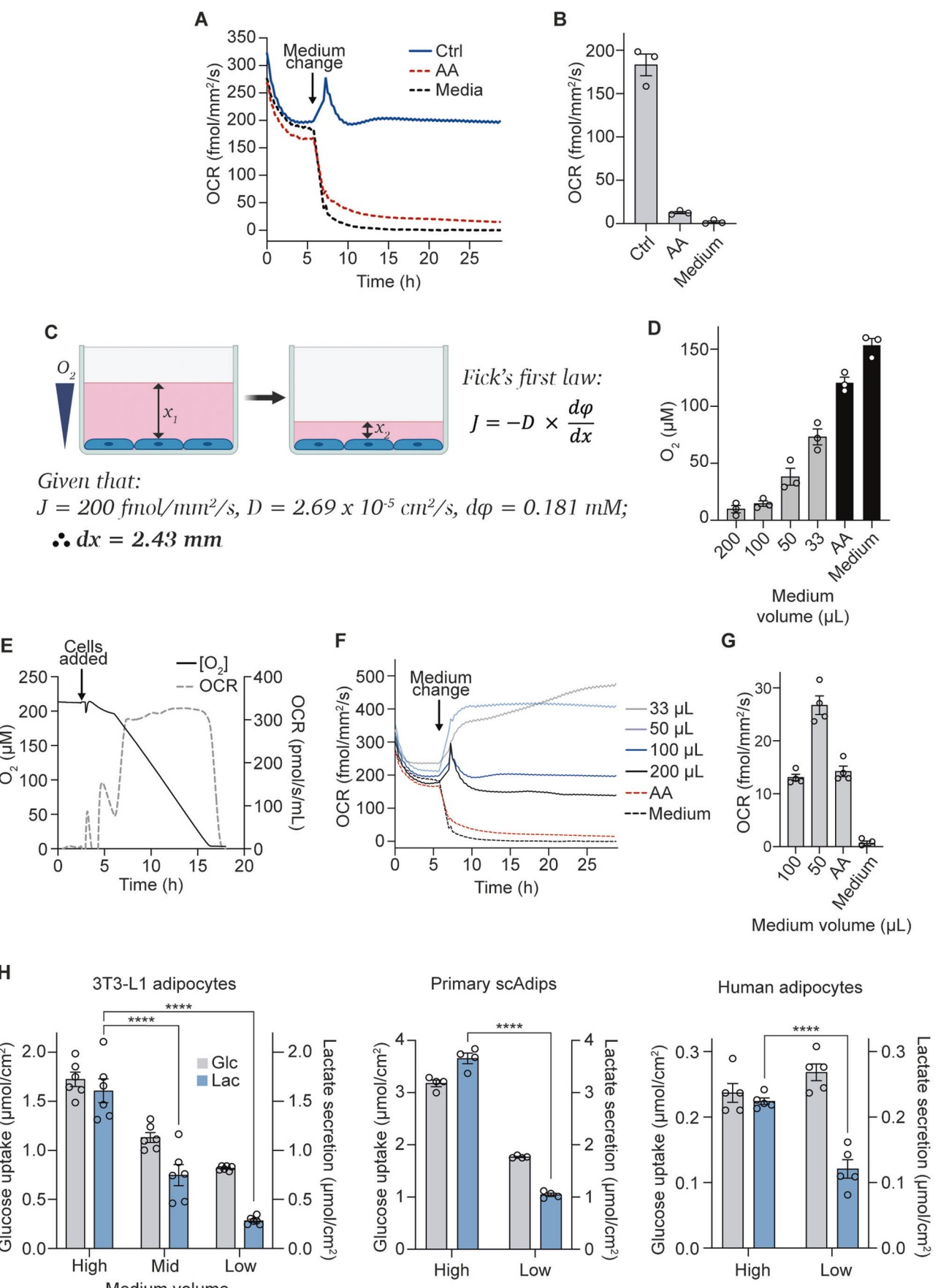

**Figure 1. Oxygen is limiting for adipocyte respiration under standard culture conditions.**

(A) 3T3-L1 adipocyte oxygen consumption rate (OCR) under control (100 μL), mitochondrial complex III inhibitor antimycin A (AA) treatment, or cell-free/medium-only conditions (96-well plate). Baseline OCR was measured for 6 h before starting AA treatment and removing cells from medium-only wells (representative of $n = 3$ biological replicates). (B) OCR measurements were taken after 20 h from (A) ($n = 3$ biological replicates). (C) Schematic representation of experimental setup and Fick's first law of diffusion, which states that the rate of diffusion ($J$) is proportional to the concentration gradient ($\varphi$) and is inversely proportional to the diffusion length ($x$), where $D$ is the diffusion coefficient. (D) Pericellular oxygen concentration at different medium volumes in 96-well plates ($n = 3$ biological replicates). (E) Oroboros measurements showing oxygen concentration and rate of OCR of 3T3-L1 adipocytes in a closed system ($n = 3$ biological replicates). (F) 3T3-L1 adipocyte OCR at different medium volumes, AA treatment, or medium-only conditions in 96-well plates. Baseline OCR (at 100 μL medium) was measured for 6 h before altering medium volumes or starting AA and medium-only treatments (representative of $n = 3$ biological replicates). (G) OCR under 1% oxygen with different medium volumes in 96-well plates ($n = 4$ biological replicates). (H) Extracellular medium glucose and lactate measurements from 3T3-L1 adipocytes ($n = 6$ biological replicates), primary subcutaneous white adipose tissue (scAdips) ($n = 4$ biological replicates), and human adipocytes ($n = 5$ biological replicates) 16 h after medium volume change. High = 1 mL, Mid = 0.67 mL, Low = 0.33 mL (in a 12-well plate) from hereafter unless stated otherwise. Data information: Data were represented as mean ± SEM (B, D, G, H). ****$p < 0.0001$ by two-way ANOVA with Šidák correction for multiple comparisons (H). See also Fig. EV1. Source data are available online for this figure.

These data suggest a near total use of available oxygen (to less than 4 μM) at the cell monolayer under standard culture conditions.

To increase oxygen at the cell monolayer, we lowered medium volumes, reducing the distance from the air-medium interface to the cell monolayer (Fig. 1C). As expected, lowering medium volumes increased pericellular oxygen (Fig. EV1B), with oxygen concentrations 0.55 mm above the cell monolayer reaching between 38–73 μM (29–56 mmHg) for cells cultured in 50 or 33 μL medium in a 96-well plate, respectively (Fig. 1D). These values were more similar to those measured in mouse (~60 mmHg) (Midha et al, 2023) and human (40.5–73.8 mmHg) (Pasarica et al, 2009) adipose tissue than cells grown under standard conditions. Therefore, lowering medium volumes resulted in an oxygen tension closer to in vivo and provided a simple experimental system to assess the impact of oxygen tension on 3T3-L1 adipocyte metabolism (Sheng et al, 2014).

Switching cells from standard (100 μL per well in 96-well plates) to low (33 μL) medium conditions rapidly (within 3–4 h) doubled the oxygen consumption rate from 200 to 400 fmol/mm²/s (Fig. 1F), showing that oxygen is limiting for cell respiration under standard culture conditions. Increased oxygen consumption was independent of changes in mitochondrial content as measured by western blotting (Fig. EV1C) or oxygen consumption in permeabilised cells (Fig. EV1D). Further, lowering oxygen provision by culturing cells in 1% $O_2$ significantly reduced OCR to just 13 fmol/mm²/s in 100 μL and severely attenuated the increase in OCR when medium volumes were lowered (Fig. 1G). These data suggest that greater oxygen delivery drives increased OCR in adipocytes cultured in lower medium volumes. Given their substantial initial OCR and large increases in both OCR and pericellular oxygen concentration in response to reduced oxygen diffusion distance, 3T3-L1 adipocytes represented an excellent model to determine the phenotypic effects of increased oxygen delivery to terminally differentiated cells in culture.

## Increasing pericellular oxygen tension decreases lactate production in adipocytes

Next, we sought to determine the phenotypic effects of increased oxygen delivery to cells. Anaerobic glycolysis is a hallmark of low oxygen availability, generating lactate instead of $CO_2$ as the end product. As 3T3-L1 adipocytes are highly glycolytic under standard culture conditions, we first determined the effects of increasing oxygen tension on adipocyte glycolytic metabolism by measuring lactate production. Under standard culture conditions (e.g. 1 mL DMEM/10% FCS per well in 12-well plates) (Fig. EV1A), 3T3-L1 adipocytes were highly glycolytic as evidenced by high lactate production relative to their glucose uptake (Fig. 1H). High rates of lactate production were also seen in two other in vitro models of adipocytes, primary subcutaneous adipocytes (scAdips), and human adipocytes (hMADS (Bezaire et al, 2009)). Estimates from human subcutaneous adipose tissue (in vivo) and mouse epididymal adipose tissue (ex vivo) place tissue lactate production at 15–30% of glucose uptake (Krycer et al, 2020; Hodson et al, 2013), which is substantially lower than observed for each of these adipocyte cell lines under standard culture conditions (close to 50%). We next determined if supplying more oxygen to adipocyte cell lines could decrease their reliance on anaerobic glycolysis.

To increase oxygen delivery to cells, we switched each of the adipocyte cell lines from 1 mL (high) to 0.67 mL (mid) or 0.33 mL (low) medium in 12-well plates. Reducing medium depth lowered glucose uptake and caused an even more drastic reduction in lactate production (Fig. 1H). To confirm that these metabolic responses to low medium conditions were due to a primary change in oxygen availability, we cultured 3T3-L1 cells in gas-permeable cell culture plates (Lumox). These plates allow oxygen diffusion from the bottom of the well, bypassing the medium layer. Similar to the low-medium conditions, gas-permeable plates lowered glucose use and lactate production (Fig. EV1E). Notably, the extent of lactate production from glucose in 3T3-L1 adipocytes cultured at a higher oxygen tension (17% lactate relative to glucose) is more similar to the in vivo values of 15–30% in human subcutaneous abdominal adipose tissue (Hodson et al, 2013).

This change in glucose use and lactate production upon switching to lower medium volumes was not driven by depletion of glucose, which remained >12 mM under all medium volumes after 16 h (Fig. EV1F). Additionally, the lower lactate-to-glucose ratio in low medium conditions was independent of starting medium glucose concentrations (Fig. EV1G), suggesting that the change in glucose metabolism was not an artefact of culturing cells in non-physiological medium glucose (Crabtree, 1929). Instead, our results indicated a considerable reduction in anaerobic glycolysis under conditions of greater oxygen availability (Fig. 1H).

Our data to this point suggested that high oxygen use by the adipocyte monolayer results in low pericellular oxygen and that limits mitochondrial oxidation of glucose and promotes anaerobic glycolysis. To further test this, we lowered cell density per well to

decrease the OCR at the monolayer. This revealed that the extent of lactate production was proportional to cell density and oxygen consumption. Seeding fewer cells per well lowered OCR (Fig. EV1H), as expected, but also the amount of lactate produced relative to glucose uptake (Fig. EV1I). Accordingly, cells seeded at a lower confluence had a diminished response to the low-medium intervention (Fig. EV1I). Together, these data support a model whereby 3T3-L1 adipocytes exhibit high rates of lactate production under standard culture conditions due to OCR/well-outstripping oxygen delivery, in turn driving anaerobic glucose metabolism. Additionally, they highlight how cell density can drive distinct cellular metabolic profiles.

## Lowering medium volumes increases de novo lipogenesis and mitochondrial glucose oxidation in adipocytes

Our observations that cells were consuming more oxygen and producing less lactate indicated a more oxidative use of glucose. However, the cells were taking up only half as much glucose in low medium relative to standard conditions, likely due to lower GLUT1 protein expression (Fig. 2A). This raised the question as to whether the cells were now able to more efficiently utilise glucose for anabolism. To test this, we first investigated whether reduced glucose uptake impacted lipid synthesis, a key anabolic process in adipocytes. Despite the 50% reduction in glucose uptake to the cells, strikingly, adipocytes cultured in lower medium volumes (0.33 mL versus 1 mL), or in Lumox plates, increased de novo lipogenesis (DNL) (Fig. 2B,C). This indicated the cells were substantially more metabolically efficient; able to produce sufficient ATP and obtain enough carbon to support DNL from less glucose. The increased efficiency of the cells was consistent with the fact that generation of ATP by converting glucose to lactate only produces 2 molecules of ATP, whereas complete oxidation in mitochondria generates up to 36 (Nelson et al, 2008). In agreement with greater mitochondrial oxidation of glucose, PDH phosphorylation was decreased following the switch to low medium volumes (Fig. 2D), consistent with decreased negative regulation of PDH by pyruvate dehydrogenase kinases (PDKs) and therefore increased PDH activity (Kolobova et al, 2001; Kim et al, 2006). Direct measurement of PDH flux using mass isotopomer distribution analysis (MIDA) of palmitate (Hellerstein and Neese, 1992) revealed a ~40% increase in glucose entry to the TCA cycle via PDH when cells were cultured in lower medium volumes (Fig. 2E).

In line with greater PDH activity, $U^{13}C$-glucose tracing (Fig. 2F) revealed that transitioning cells to low medium (0.33 mL in 12-well plates) for either 4 h (Fig. EV2A) or 16 h (Fig. 2G) increased $^{13}C$-labelling of all TCA cycle intermediates (Dataset EV1). Additionally, enrichment of $m + 3$ isotopologues for fumarate and malate suggested increased pyruvate carboxylase activity, which allows for TCA anaplerosis by replenishing oxaloacetate. In total, the abundance of 45 metabolites was changed by culturing cells in low medium for 16 h, indicating comprehensive metabolic rewiring of cells beyond just the TCA cycle (Fig. EV2B). Of note, α-ketoglutarate (α-KG) was the most upregulated TCA metabolite under low medium conditions (Fig. EV2C), with the large majority of the additional α-KG containing glucose-derived carbon (Fig. 2H). These results demonstrate that increasing oxygen tension can rewire multiple branches of cell metabolism in addition to increased glucose oxidation via the TCA cycle.

## Standard medium volumes drive a widespread transcriptional response reminiscent of physiological hypoxia

Changes in cellular oxygen tension are known to drive transcriptional changes through multiple mechanisms, many centred on HIF (Dengler et al, 2014; Schofield and Ratcliffe, 2004). In addition to changes in oxygen tension, we also noted Increased α-KG abundance (Fig. 2H), which governs the activity of α-KG-dependent dioxygenase prolyl hydroxylases, which regulate HIF1α stability and activity (Tennant et al, 2009; Iommarini et al, 2017; Pan et al, 2007), as well as reduced protein levels of GLUT1 (Fig. 2A), a canonical HIF1α target. Given that HIF1α is a transcription factor, we confirmed that changes in GLUT1 protein were also present on a transcriptional level. Accordingly, expression of *Slc2a1* (which encodes GLUT1) alongside other HIF1α target genes (*Slc16a3, Pgk1, Pdk1, Car9, Pkm2*) were reduced under low medium conditions (Fig. 3A) and in cells cultured in gas-permeable culture plates (Lumox) (Fig. EV3A). Finally, we confirmed the direct regulation of the HIF1α protein itself. Consistent with the known kinetics of the HIF/prolyl hydroxylase domain (PHD) system (Moroz et al, 2009), HIF1α was rapidly degraded in <5 min after transition to low medium (Fig. 3B), and was still drastically decreased after 16 h (Fig. 3C).

Despite high rates of lactate production under standard culture conditions (Fig. 1H), adipocytes still mounted a considerable lactate response to further hypoxia (1 and 5% oxygen; Fig. EV3B,C). However, the effect of switching to low medium volumes was attenuated in cells cultured in either 1 or 5% $O_2$ when compared to standard conditions (Fig. EV3D). Further, HIF1α stability, as well as its target gene expression, was largely refractory to low medium intervention at 1% $O_2$ (Figs. 3D and EV3E). These data support our conclusion that low medium affects lactate production and HIF1α activity through greater oxygen provision, and demonstrate that the 3T3-L1 cell line has a regulatable and functional oxygen-responsive HIF system, which though is active under standard culture conditions, still had the capacity to respond to even lower oxygen supply.

To more fully characterise the transcriptional responses to low medium, we performed RNAseq analyses. This revealed extensive and widespread changes in gene expression in cells cultured in a low medium, with over 3000 differentially expressed genes. Multiple HIF1α targets were expressed at greater levels in high medium (Fig. 3E; Dataset EV2). VIPER transcription factor prediction software predicted 125 transcription factors to either be inhibited or activated (Dataset EV3) by changing medium volumes, including HIF1α (NES 8.8, FDR $P$ value = $1.6 \times 10^{-16}$). Finally, FGSEA identified 63 altered KEGG pathways (Dataset EV4). Notably, mitochondrial function, TCA cycle activity, and lipogenesis, were all predicted to be increased by low medium (Dataset EV4), consistent with our biochemical analyses (Figs. 1 and 2).

Given the widespread transcriptional response to increased oxygen tension in cultured cells, we next compared our in vitro findings to in vivo adipose tissue to assess whether transcriptional responses observed in adipocytes in standard culture conditions resembled adipose tissue responses to hypoxia. We compared the transcriptional profile of 3T3-L1 adipocytes in low versus high medium culture conditions to subcutaneous white adipose tissue (scWAT) from mice exposed to either normoxia (21% $O_2$) or hypoxia (10% $O_2$, 4 weeks) (Fig. EV3F; Datasets EV5 and EV6).

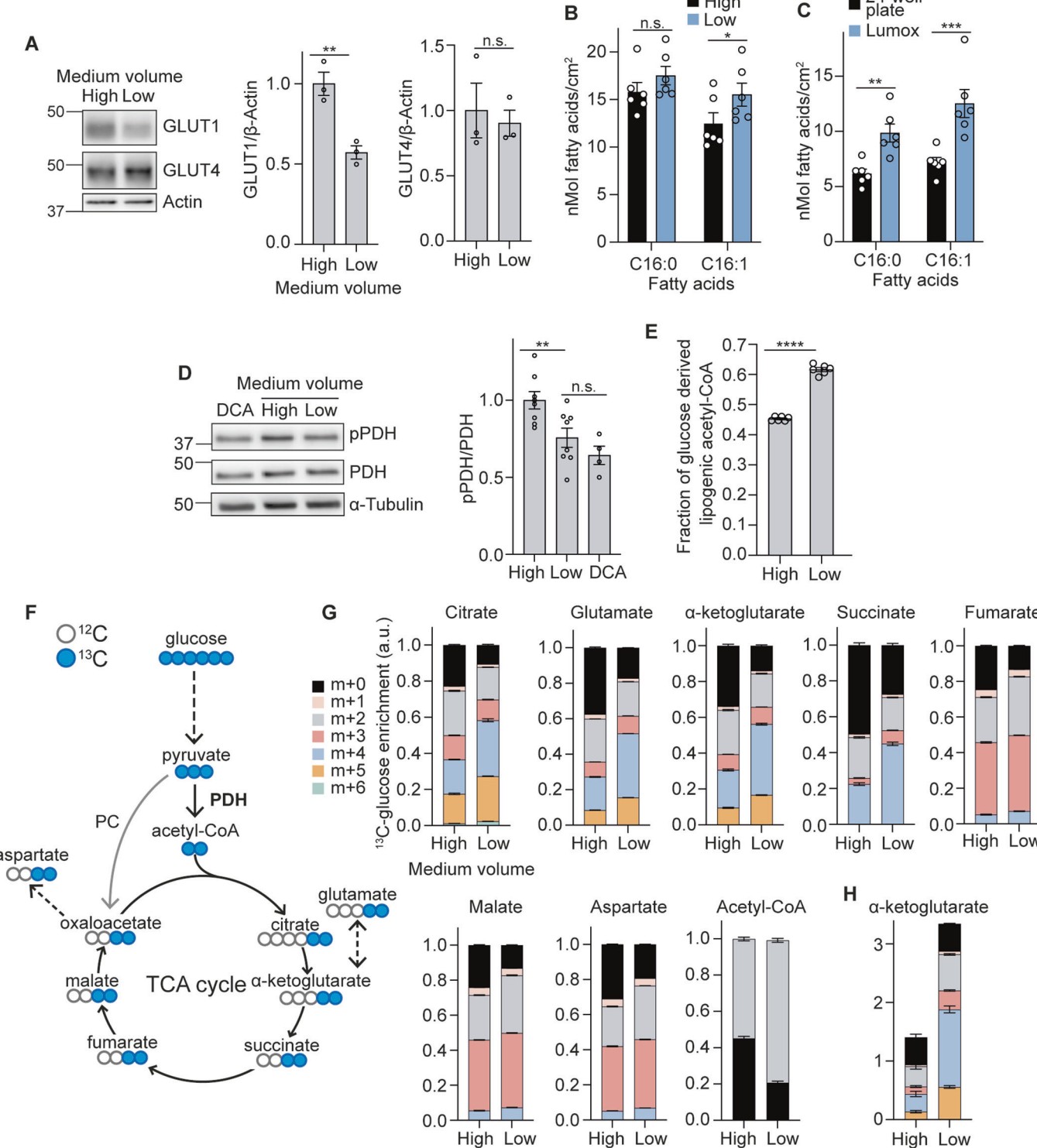

Both individual gene and pathway analyses revealed a high degree of overlap in gene expression responses of cells cultured in high medium and hypoxic adipose tissue. Specifically, of the 2239 differentially expressed genes in hypoxic mice scWAT, 703 were also changed in the 3T3-L1 adipocytes cultured in high medium (Fig. EV3G), with a 77% concordance in the direction of change

(Fig. EV3H). Additionally, out of 312 KEGG pathways tested, 63 were significantly altered in 3T3-L1 adipocytes and 61 in scWAT, of which 34 were significant in both (Chi$^2$ for overlap $p = 6 \times 10E^{-11}$) (Datasets EV4 and EV6). Remarkably, 32 of the 34 common pathways were concordantly regulated (Fig. 3F; Datasets EV4 and EV6). Overall, our data suggested that increasing

Figure 2. Lowering medium volumes increases mitochondrial glucose oxidation.

(A) Western blot and quantification of GLUT1 and GLUT4 after 16 h of medium volume change ($n = 3$ biological replicates). (B) De novo lipogenesis (DNL) of palmitate (C16:0) and palmitoleate (C16:1) during 16 h of culture in high or low medium volumes in 12-well plates ($n = 6$ biological replicates). (C) DNL of palmitate and palmitoleate after 16 h in 24-well or gas-permeable Lumox plates ($n = 6$ biological replicates). (D) Western blot and quantification of phospho-pyruvate dehydrogenase (pPDH) over total PDH after 16 h of medium volume change in 12-well plates, with dichloroacetate (DCA) as a positive control ($n = 4$–8 biological replicates). (E) Fraction of newly synthesised lipids after 16 h of medium volume change and U$^{13}$C-glucose labelling in 12-well plates ($n = 6$ biological replicates). (F) Schematic of $^{13}$C enrichment in TCA cycle metabolites after one cycle of U$^{13}$C-glucose labelling. PC, pyruvate carboxylase. (G) Graphs show the fractional abundance of each isotopologue after 16 h medium volume change ($n = 4$–6 biological replicates). (H) Total abundance of α-ketoglutarate after 16 h medium volume change ($n = 4$–6 biological replicates). Data information: Data were represented as mean ± SEM (A–E, G, H). n.s. non-significant; *$p < 0.05$, **$p < 0.01$, ****$p < 0.0001$ by paired two-tailed Student's *t*-test (A, D and E) or by two-way ANOVA with Šidák correction for multiple comparisons (B, C). See also Fig. EV2. Source data are available online for this figure.

oxygen concentrations relative to standard culture conditions drove a similar pattern of transcriptional changes to that seen between hypoxic and normoxic adipose tissue. These data demonstrate that gene expression changes observed in 3T3-L1 adipocytes under standard conditions (compared to low medium conditions) were a physiological response to limiting oxygen reminiscent of adipose tissue responses to hypoxia.

## Lowering medium volumes improves adipocyte function in both 3T3-L1s and primary scAdips

Adipose tissue plays a key role in whole-body energy homoeostasis through adipokine secretion and energy storage and release. These processes are dysregulated in hypoxia and in obesity (Yin et al, 2009; Halberg et al, 2009; Hosogai et al, 2007). Indeed, the most altered KEGG pathways in 3T3-L1 adipocytes cultured in high-medium volumes were similarly changed in subcutaneous adipose tissue from obese individuals (van der Kolk et al, 2021). Therefore, we next tested how increased oxygen provision impacted adipocyte adipokine secretion, lipolysis, and insulin responsiveness. We cultured 3T3-L1 adipocytes in high or low medium volumes for 48 h, during which their metabolic switch to glucose oxidation in low medium was maintained (Fig. EV4A). At this time point, low medium increased both leptin and adiponectin secretion (Fig. 4A,B), consistent with gene expression data for *Lepn* and *AdipoQ* (Dataset EV2). Additionally, extended culture in low medium increased the sensitivity of 3T3-L1 adipocytes to the lipolysis-stimulating β3-adrenergic receptor agonist CL316,243 more than 15-fold (EC$_{50}$ 0.21 versus 3.43 nM) (Fig. 4C). Despite higher lipolytic sensitivity, insulin was equally effective at suppressing lipolysis in response to CL316,243 (Fig. 4D). Low medium also markedly increased insulin-stimulated GLUT4 translocation at both physiological and supraphysiological insulin concentrations (Fig. 4E). Similarly, while total 2-deoxyglucose (2-DG) uptake in the presence of insulin was similar in cells cultured in low or high medium volumes (Fig. EV4B), the treatment of cells with the GLUT4-inhibitor indinavir demonstrated that 2-DG uptake was more GLUT4-dependent in low medium (Fig. EV4C), despite no changes in total GLUT4 protein (Fig. EV4D). Overall, these phenotypic measures suggested that increased oxygen tension improved adipokine release, and optimised the coupling between lipolysis and insulin responses in 3T3-L1 adipocytes.

To confirm that altered insulin responses were a function of increased oxygen provision, we tested GLUT4-translocation responses in cells cultured in 1 or 18% O$_2$. Cells in 1% O$_2$ had impaired insulin responses (Fig. 4F). Further, similar to the effect of ambient hypoxia on the metabolic and HIF response to low

medium (Figs. 3D and EV3B,C), the effect of low medium on GLUT4 translocation responses was severely blunted (Fig. 4F). These data highlight that oxygen availability modulates cellular sensitivity to insulin stimulation, consistent with the observation of decreased adipose tissue oxygenation in cases of obesity-induced insulin resistance (Cifarelli et al, 2020).

Given the stark improvements in these 3T3-L1 adipocyte phenotypes, we tested the translatability of these findings to subcutaneous white adipocytes (scAdips) cultured from primary mouse mesenchymal stem cells, which also exhibited lower lactate production when provided with more oxygen (Fig. 1H). As in 3T3-L1 adipocytes, low medium intervention increased pericellular oxygen tension (Fig. EV4E) and OCR (Fig. EV4F), and decreased HIF1α stability (Fig. 4G) as well as expression of HIF1α target genes (Fig. 4H). Further, greater oxygen availability improved scAdips response to insulin stimulation as measured by increased GLUT4 translocation (Fig. 4I). These data highlight that the improvements to adipocyte function were not limited to 3T3-L1 adipocytes, but also extended to primary cells.

## Lowering medium volumes reduces lactate production and improves functional outcomes in other cell types and organoids

Finally, we expanded our analysis to other cell types/tissues. First, we assessed responses to increased oxygen at the level of lactate secretion and/or HIF activity in other post-mitotic cell lines. pBAT cells (murine brown adipocytes) and L6 myotubes (rat) both reduced lactate secretion in response to lower medium volumes (Fig. 5A). Human induced pluripotent stem cells (hiPSC)-derived neurons showed little change in their lactate response (trend to decrease; Fig. EV5A), however expression of HIF1α target genes *PGK1* and *PKM2* were reduced in low medium (Fig. EV5B), suggesting that these cells responded to increased oxygen and that transcriptional changes to increased oxygen can occur independent of large metabolic changes.

Next, we assessed the functional effects of increasing oxygen tension in primary macrophages, h(i)PSC-derived hepatocytes as well as cardiac organoids, which are all preclinical models used to study cardiometabolic complications. It is generally accepted that pro-inflammatory macrophages rely on glycolysis, while their anti-inflammatory counterparts rely on OXPHOS and fatty acid oxidation (FAO) for ATP production (Van den Bossche et al, 2017). Decreased medium volumes lowered lactate production (Fig. 5B) and enhanced FAO (Fig. 5C). Apart from metabolic changes, expression of HIF1α-regulated *Slc2a1* decreased in low medium, as well as inflammatory marker *Tnfα*, whereas *Cd206*, an

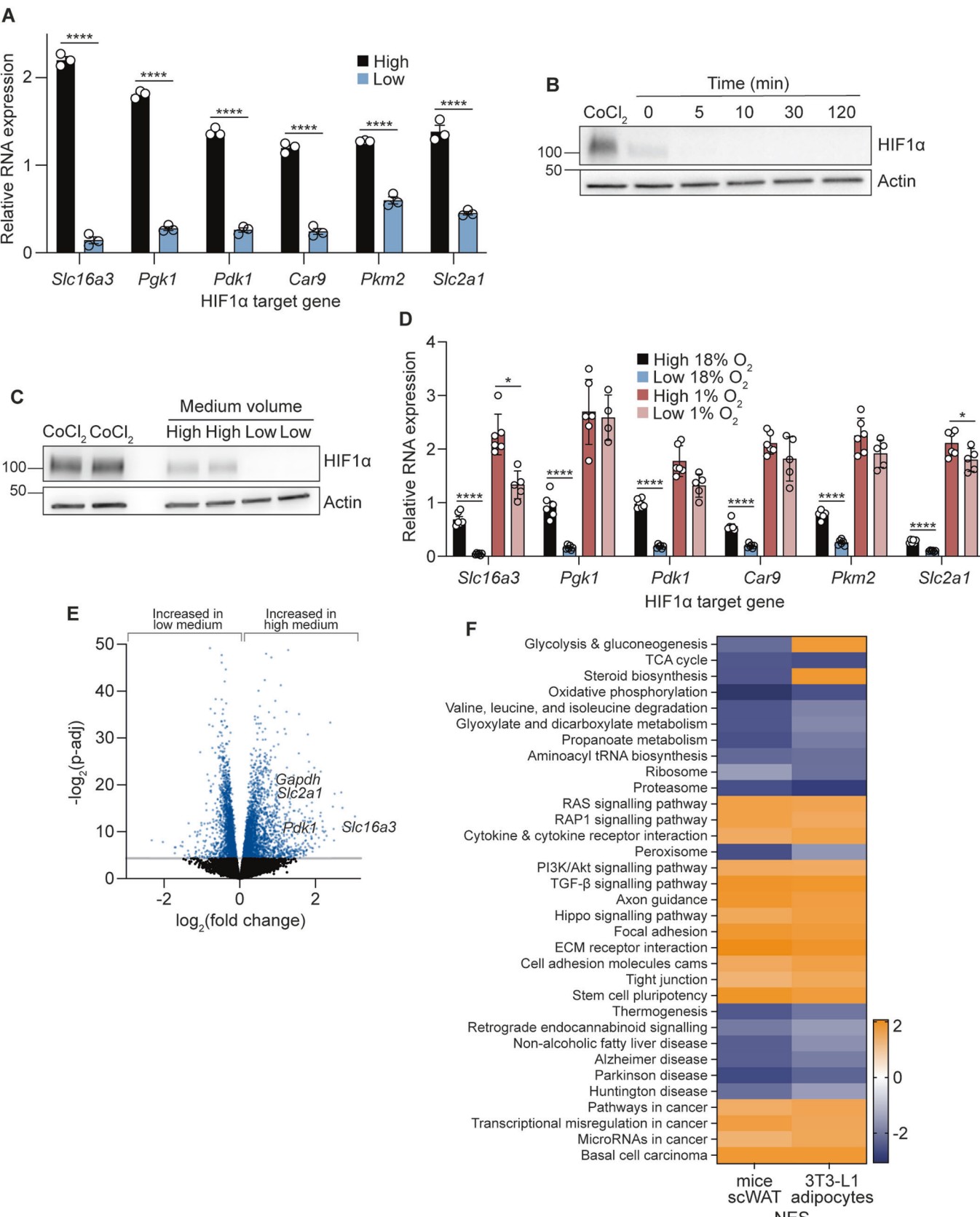

◀ **Figure 3. Lowering medium volumes induces a widespread transcriptional response reminiscent of physiological hypoxia.**

(A) Relative RNA expression of HIF1α target genes after 16 h medium volume change in 12-well plates ($n = 3$ biological replicates). (B) Western blot of hypoxia-inducible factor (HIF) 1α at various time points after the transition to low medium volumes in 12-well plates, with 500 µM $CoCl_2$ as a positive control ($n = 3$ biological replicates). (C) Western blot of HIF1α after 16 h medium volume change in 12-well plates, with 500 µM $CoCl_2$ as positive control ($n = 3$ biological replicates). (D) Relative RNA expression of HIF1α target genes cultured in either 1 or 18% incubator oxygen for 16 h with different medium volumes in 12-well plates ($n = 4$–6 biological replicates). (E) Volcano plot of differentially expressed genes after 16 h medium volume change in 12-well plates ($n = 6$ biological replicates). (F) KEGG pathway analyses of 3T3-L1 adipocytes after 16 h medium volume change in 12-well plates ($n = 6$ biological replicates) and subcutaneous white adipose tissue (scWAT) from mice kept in 10 or 21% oxygen for 4 weeks ($n = 10$ biological replicates). Orange (positive NES) represents upregulated KEGG pathways in high medium (3T3-L1 adipocytes) or in 10% oxygen (mice scWAT). Blue (negative NES) represents upregulated KEGG pathways in low medium (3T3-L1 adipocytes) or in 21% oxygen (mice scWAT). NES, normalised enrichment score. Data information: Data were represented as mean ± SEM (A, D). *$p < 0.05$, ****$p < 0.0001$ by paired Student's $t$-tests (A, D). $p$-adj threshold of 0.05 in (E) is the adjusted $p$ value after controlling for the false discovery rate (FDR) with the Benjamini–Hochberg procedure. The raw $p$ values were determined using the Wald test. See also Fig. EV3. Source data are available online for this figure.

anti-inflammatory marker was increased (Fig. 5D). Together, these suggest that providing more oxygen to cultured macrophages results in both metabolic and transcriptional changes indicating a switch towards a more anti-inflammatory phenotype.

Culturing either chemically-induced (FSPS13B) or forward-programmed (FOP) hiPSC-derived hepatocytes in low medium (0.5 mL versus 1 mL) throughout differentiation decreased lactate secretion (Figs. 5E and EV5D) and HIF1α target gene expression (Fig. EV5C,E), indicating that lowering medium volumes increased oxygen tension in both cell types. Concomitant with changes in HIF1α activity, the hepatocyte differentiation marker *ALB* (albumin) were increased in FSPS13Bs, whereas levels of *SERPINA1* and lineage specification marker *HNF4α* did not (Watt et al, 2003; DeLaForest et al, 2011) (Fig. 5F). On the protein level, hepatocyte albumin content and/or secretion was increased for both FSPS13Bs (Fig. 5G,H) and FOPs (Fig. EV5F), even though *ALB* was not found to be significantly upregulated in the FOPs (Fig. EV5G). Additionally, CYP3A4 activity, a liver-specific metabolic enzyme, increased with a low medium in the FOPs and trended towards an increase for the FSPS13Bs (Figs. 5I and EV5H). Overall our results suggested that lower medium volumes, which decreased lactate production and HIF1α activity, may increase the differentiation potential and/or functionality of hiPSC-derived hepatocytes (Gilglioni et al, 2018; Poyck et al, 2008).

In hPSC-derived cardiac organoid cultures, we again observed less lactate secretion when cultured in lower medium volumes (Figs. 5J and EV5I). Organoids in low medium exhibited greater contractile force (Fig. 5K), without changes in contractile rate, or activation and relaxation times (Fig. EV5J–L). Together, our data in a range of terminally differentiated cell types suggest that manipulating oxygen tension, at least in cells that model major metabolic organs, can substantially impact the metabolism and phenotype of multiple different cell types, which has significant implications on the fidelity and translatability of in vitro culture models.

## Discussion

Using a comprehensive series of biochemical, multi-omics, and phenotypic analyses, we demonstrated that multiple terminally differentiated metabolic cell models experience a degree of hypoxia under standard culture conditions, as a result of oxygen use outstripping oxygen delivery through the medium column. Despite the known importance of oxygen, its importance for understanding

cellular experiments and translatability to in vivo outputs has not been widely considered, perhaps due to a lack of data on how oxygen deficiencies actually affect cells in culture both metabolically and phenotypically. Our study builds upon previous work on oxygen tension in cell culture (Stevens, 1965; Weiszenstein et al, 2016; Gilglioni et al, 2018; Gstraunthaler et al, 1999; Metzen et al, 1995), to provide a comprehensive view of the broad impact of oxygen limitations in standard cell culture. Importantly, we show that increasing oxygen availability resulted in significant metabolic and transcriptional changes, and markedly improved cellular function in several cell types that model major metabolic organs —namely adipocytes, macrophages, hiPSC-derived hepatocytes, and hPSC-derived cardiac organoid cultures. These data provide empirical evidence that limited oxygen can have profound effects on cell metabolism and function under standard culture conditions, and highlight that considering pericellular oxygen can provide substantial benefits when developing cell culture models.

The majority of our study focused on cultured adipocytes, which offered a useful model to investigate how limiting oxygen dysregulates cell metabolism and function. Adipocytes are highly metabolically active, and adipose tissue hypoxia is a common pathological feature in obesity and associated cardiometabolic complications characterised by adipose tissue dysfunction (Todorčević et al, 2021; Trayhurn et al, 2008; Cifarelli et al, 2020). Indeed, the transcriptional response to low-high medium transition largely resembled an in vivo adipose tissue hypoxia response, suggesting that standard culture conditions recapitulate the adaptations to hypoxia in vivo (Fig. 3F). Alleviating hypoxia in culture improved adipokine secretion, insulin sensitivity and lipolysis (Fig. 4), indicating that lowering medium volumes recapitulates a healthier adipocyte phenotype. Of note, obesity and adipose tissue hypoxia stimulate an inflammatory response by macrophages. Consistent with this, providing greater oxygen to cultured macrophages promoted an anti-inflammatory profile (Fig. 5). Together, these data highlight the importance of considering pericellular oxygen in modelling distinct cell types from adipose tissue, and is an especially important consideration when using adipocytes or macrophages for studying obesity and/or other aspects of metabolic disease.

Importantly, these metabolic and functional changes were not limited to adipocytes and macrophages but extended to other relevant cardiometabolic. These included both iPSC-derived hepatocytes and cardiac organoids. There are several possibilities how increased oxygen availability in culture may improve or restore cellular function. The observed impact on cellular phenotypes may be HIF-driven, given the wide range of cellular functions influenced

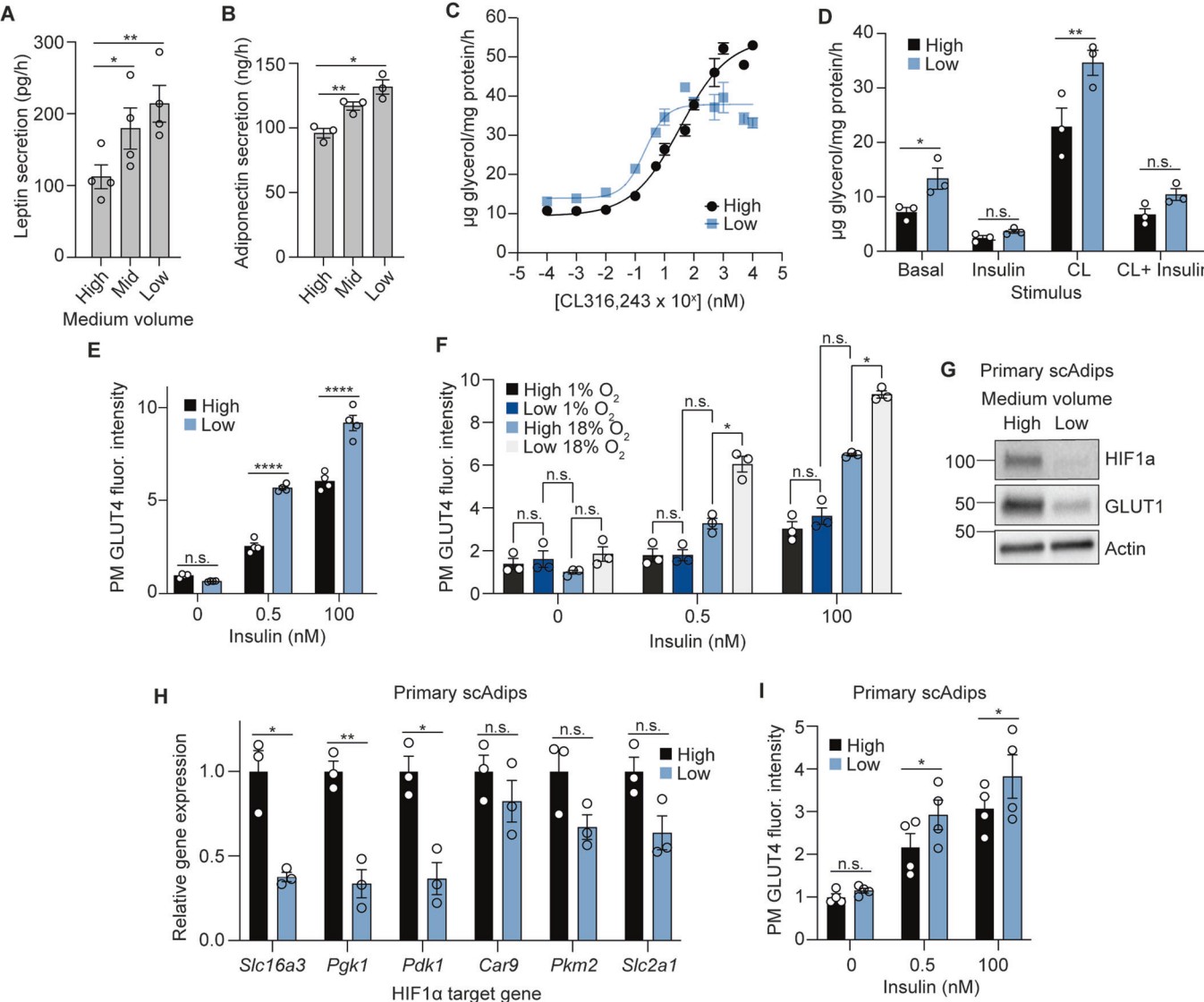

**Figure 4. Lowering medium volumes improves adipocyte function in both 3T3-L1s and scAdips.**

(A) Rate of leptin secretion under different medium volume conditions in 12-well plates ($n = 4$ biological replicates). (B) Rate of adiponectin secretion under different medium volume conditions in 12-well plates ($n = 3$ biological replicates). (C) Dose-response curve of the lipolytic drug, CL316,243 treatment in 12-well plates ($n = 3–7$ biological replicates). (D) Rate of lipolysis measured by glycerol release, upon 100 nM insulin or 1 nM CL316,243 stimulation ($n = 3$ biological replicates). (E) Fluorescence intensity of plasma membrane (PM) GLUT4 upon insulin stimulation after 48 h medium volume change in 96-well plates ($n = 4$ biological replicates). (F) Fluorescence intensity of PM GLUT4 upon insulin stimulation after 48 h culture in either 1 or 18% incubator oxygen with medium volume changes in 96-well plates ($n = 3$ biological replicates). (G) Western blot of HIF1α and GLUT1 from primary scAdips after 16 h medium volume change (12-well plate), with 500 μM $CoCl_2$ as positive control ($n = 4$ biological replicates). (H) Relative RNA expression of HIF1α target genes in primary scAdips after 16 h medium volume change (12-well plate) ($n = 3$ biological replicates). (I) Fluorescence intensity of primary scAdips PM GLUT4 upon insulin stimulation after 48 h medium volume change in 96-well plates ($n = 4$ biological replicates). Data information: Data were represented as mean ± SEM (A, B, D–F, H, I). n.s. non-significant; *$p < 0.05$, **$p < 0.01$, ****$p < 0.0001$ by one/two-way ANOVA with Šidák correction for multiple comparisons (A, B, D–F and I), or by paired Student's $t$-tests (H). See also Fig. EV4. Source data are available online for this figure.

by HIF (Dengler et al, 2014; Catrina and Zheng, 2021). For example, we measured increased rates of lipolysis in cells with greater oxygen availability, consistent with previous findings (Michailidou et al, 2015). However, our RNAseq analyses revealed transcriptional changes beyond the HIF pathway (Fig. 3), suggesting that additional pathways may connect oxygen availability to cellular behaviour. Indeed, signalling molecules such as reactive oxygen species (D'Autréaux and Toledano, 2007) or metabolites

(Baker and Rutter, 2023) may be implicated. Regardless of the mechanism, the functional improvements upon increased oxygen delivery to cells provide a strong impetus for considering oxygen tension as a critical and adjustable parameter for optimising cell culture models. Notably, the functional changes observed were variable between different culture models, highlighting the difficulty in establishing a single formula for the optimal provision of oxygen when considering distinct cell types.

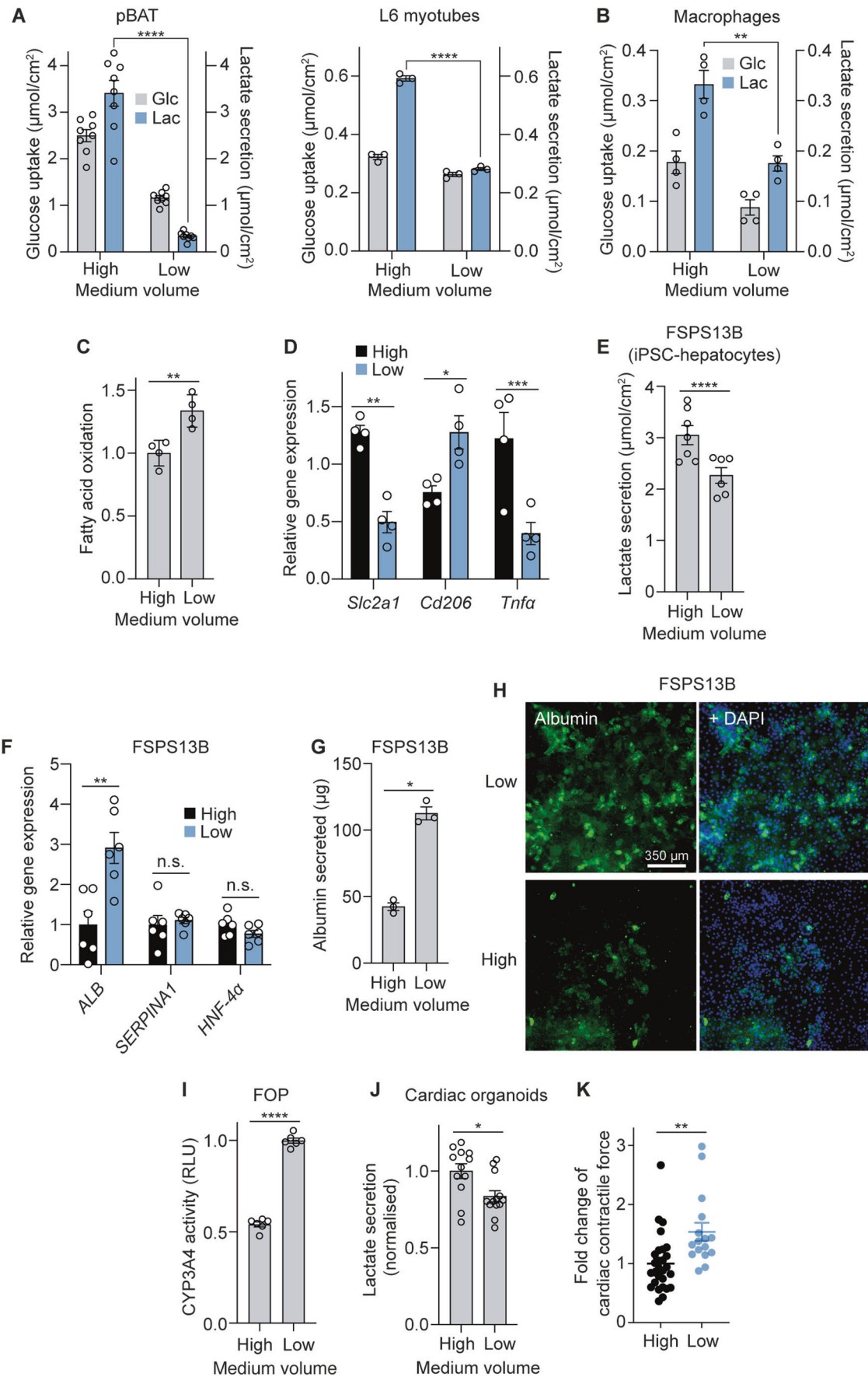

**Figure 5. Lowering medium volumes reduces lactate production and improves functional outcomes in other cell types and organoids.**

(A) Extracellular medium glucose and lactate measurements after 16 h medium volume change in 12-well plates in murine brown adipocytes (pBAT) ($n = 8$ biological replicates) and L6 myotubes ($n = 3$ biological replicates). (B) Extracellular measurements of primary macrophage medium glucose and lactate 16 h after medium volume change in 24-well plates ($n = 4$ biological replicates). (C) Fatty acid oxidation in primary macrophages after 16 h medium volume change in 24-well plates ($n = 4$ biological replicates). (D) Relative RNA expression of HIF1α target gene *Slc2a1*, anti-inflammatory marker *Cd206*, and inflammatory marker *Tnfα* in primary macrophages after 16 h medium volume change in 24-well plates ($n = 4$ biological replicates). (E) Lactate secretion in FSPS13B iPSC-derived hepatocytes (High = 1 mL, Low = 0.5 mL) after medium volume change in 12-well plates throughout differentiation ($n = 6–7$ biological replicates). (F) Relative RNA expression of hepatocyte differentiation marker genes in FSPS13B hepatocytes after medium volume change in 12-well plates throughout differentiation ($n = 6$ biological replicates). (G) Albumin secretion over 24 h by FSPS13B hepatocytes after medium volume change in 12-well plates throughout differentiation ($n = 3$ biological replicates). (H) Immunofluorescence of albumin in FSPS13B hepatocytes after medium volume change throughout differentiation ($n = 3$ biological replicates). Scale bar = 350 μm. (I) Relative CYP3A4 activity in FOP hepatocytes after medium volume change throughout differentiation ($n = 6$ biological replicates). (J) Lactate secretion in cardiac organoids after 48 h of medium volume change in 96-well plates (normalised) ($n = 3–6$ technical replicates from $n = 3$ biological replicates). High = 150 μL, Low = 50 μL. (K) Cardiac contractile force (normalised) ($n = 2–17$ technical replicates from $n = 4$ biological replicates). Data information: Data were represented as mean ± SEM (A–G, I–K). n.s. non-significant; *$p < 0.05$, **$p < 0.01$, ***$p < 0.001$, ****$p < 0.0001$ by one/two-way ANOVA with Šidák correction for multiple comparisons (A, B, D) or by paired/unpaired two-tailed Student's t-test (C, E–G, I, J, K). See also Fig. EV5. Source data are available online for this figure.

Lowering medium volumes provides an accessible method to increase oxygen provision by decreasing oxygen diffusion distances, which can be easily employed by most laboratories. However, this method has its limitations, such as medium evaporation, nutrient depletion and altered concentrations of secreted factors that need to be accounted for, probably through more frequent medium changes. Nevertheless, changes in metabolism, transcription and function observed in our study are attributable to increased oxygen delivery since (1) culturing cells in hypoxic chambers negated the effects of low medium on lactate, HIF1α activity and function and (2) gas-permeable plates resulted in the same responses as decreasing medium volumes.

Other methods to increase cellular oxygen have been employed (Place et al, 2017), but bioreactors or perfusion systems most accurately recapitulate the sophisticated oxygen delivery system of the vasculature since they continuously supply an oxygenated medium (Sucosky et al, 2004). However, scalability and accessibility to this method pose barriers to widespread use. Suspension cell culture employs a similar principle, whereby growing cells under conditions where the medium is continuously agitated likely allows for greater oxygen provision (Cooper et al, 1958). A field where oxygen tension has perhaps been more carefully considered is organoid culture. Air−liquid interface culture models have been adopted for many stem-cell and organoid lines, including the respiratory tract (Abo et al, 2022), brain (Giandomenico et al, 2019), tumour immune microenvironment (Neal et al, 2018), and gastrointestinal tissues (Li et al, 2014; Klasvogt et al, 2017), with many studies highlighting the improved fidelity of in vitro air-liquid interface models to their in vivo counterparts. Our results regarding the impact of increased oxygen on cell differentiation and function, particularly in cardiac organoids (Fig. 5), offer a compelling explanation for the efficacy of these air-liquid interface systems and ultimately provide a means and rationale for further improvements to hPSC-based models.

Overall, our findings on pericellular oxygen concentrations and function have important implications for cell culture models. First, in our study, we observed responses to increased oxygen in all cell types studied. However, both the qualitative and quantitative changes each cell type exhibited varied. The individual response of a given cell line is likely driven by a wide range of factors, including local oxygen tension, which is dependent on OCR, cell number, energy demand, intrinsic metabolic preferences, and differentiation stage. Indeed, while we have focused exclusively on confluent terminally differentiated cell types, our findings are likely more widely applicable. For example, studies in proliferative cell types have reported increased HIF1α activity and stabilisation under standard culture conditions, particularly when grown at higher densities (Dayan et al, 2009; Le et al, 2014). Since cell density is a key determinant of oxygen consumption at the cell monolayer (Fig. EV1H,I) (Dayan et al, 2009), whether specific cells in culture experience hypoxia and would functionally benefit from increased oxygen will depend on both cell confluency during experiments and the oxygen use of the specific cell type.

Second, our study highlights a critical distinction between incubator oxygen levels and the local oxygen concentration experienced by cells. This suggests that reporting of incubator oxygen % is insufficient and that measuring pericellular oxygen is required to reveal the actual oxygen concentration experienced by cells (Rogers et al, 2023). Notably, culturing 3T3-L1 adipocytes in low medium accurately matched in vivo oxygen tensions (Midha et al, 2023; Pasarica et al, 2009) as well as lactate export (Hodson et al, 2013). Shifting cells to a more physiological oxygen tension correlated with improved adipocyte function (Fig. 4). Since cells in vivo exist on a spectrum of oxygen tensions (Ast and Mootha, 2019), we suggest that the best reference point for optimal oxygen tension for a specific cell type is to match the pericellular oxygen to that of the relevant tissue in vivo.

Finally, the powerful effects of adjusting oxygen tension on cell phenotypes highlight the need for researchers to control and report on factors that impact oxygen availability (e.g. medium volumes and cell densities) to ensure reproducibility. For instance, acute changes in medium volumes (e.g. adding more medium to cells over the weekend, or reducing medium volumes during assays to save on expensive reagents) may lead to profound changes in experimental outcomes.

Our findings ultimately highlight that cell culture is a state of variable oxygen tension depending on factors such as oxygen consumption by cells and limitations of oxygen diffusion through the medium column. Manipulating oxygen levels can cause dramatic effects on many aspects of cellular metabolism and function (Jain et al, 2020). As such, these findings complement recent data on the use of more physiological media (Vande Voorde et al, 2019; Cantor et al, 2017), with potentially important implications for the translatability of both cell and tissue culture models to in vivo settings.

# Methods

**Reagents and tools table**

| Reagent/Resource | Reference or Source | Identifier or Catalogue Number |
|---|---|---|
| **Experimental Models** | | |
| 3T3-L1 (*Mus musculus*) | David E. James' Lab (University of Sydney, Australia), originally from Howard Green (Harvard Medical School, Boston, MA) | RRID: CVCL_0A20 |
| C57BL/6 J (*Mus musculus*) | Wellcome-MRC Institute of Metabolic Science Disease Model Core | |
| Primary subcutaneous adipocytes (*Mus musculus*) - extracted from male and female C57BL/6 J mice | This study | |
| hMADS (*Homo sapiens*) | Centre National de la Recherche Scientifique (Ez-Zoubir Amri, iBV, Nice, France) | |
| L6 myoblasts (*Rattus norvegicus*) | David E. James' Lab (University of Sydney, Australia), originally from David Yaffe (Weizmann Institute of Science, Israel) | RRID: CVCL_0385 |
| Male 129Ev/Sv (*Mus musculus*) | Frank Gonzalez (National Institutes of Health, Bethesda, MD) | |
| pBAT (*Mus musculus*) | Ángela María Martínez Valverde (IIBm Alberto Sols (CSIC-UAM)) | |
| FSPS13B iPSC-hepatocytes (*Homo sapiens*) | Wellcome Sanger Institute HipSci initiative, UK | |
| FOP iPSC-hepatocytes (*Homo sapiens*) | Ludovic Vallier's Lab (Cambridge Stem Cell Institute, UK) | |
| iPSC-i3Neurones (*Homo sapiens*) | Michael E. Ward (National Institute of Neurological Disorders and Stroke, MD, USA) | io1001 (bit.bio) |
| Bone marrow-derived macrophages (*Mus musculus*) - extracted from C57BL/6 J mice | This study | |
| HES3 (*Homo sapiens*) | WiCell | RRID: CVCL_7158 |
| **Recombinant DNA** | | |
| N/A | | |
| **Antibodies** | | |
| Mouse anti-α-tubulin, monoclonal, 1:1000 (WB) | Sigma-Aldrich | Cat # T9026 |
| Rabbit anti-β-Actin, monoclonal, 1:1000 (WB) | Cell Signaling Technology | Cat # 8457 |
| Rabbit anti-GLUT1, monoclonal, 1:1000 (WB) | Cell Signaling Technology | Cat # 12939 |
| Rabbit anti-GLUT4, polyclonal, 1:1000 (WB) | Gift from F. Koumanov, University of Bath and affinity-purified in-house | N/A |
| Rabbit anti-HIF1α, monoclonal, 1:1000 (WB) | Cell Signaling Technology | Cat # 14179 |
| Mouse OxPhos Rodent WB Antibody Cocktail, monoclonal, 1:250 (WB) | Invitrogen | Cat # 45-8099 |
| Rabbit anti-PDH, polyclonal, 1:2000 (WB) | Cell Signaling Technology | Cat # 2784 |
| Rabbit anti-pPDH, polyclonal, 1:2000 (WB) | Cell Signaling Technology | Cat # 31866 |
| Goat anti-Rabbit IgG, HRP, polyclonal, 1:2500 (WB) | Invitrogen | Cat # G-21234 |
| Goat anti-Mouse IgG, HRP, polyclonal, 1:2500 (WB) | Invitrogen | Cat # 31430 |
| Human-anti-GLUT4, 2 µg/mL, 1:1000 (IF) | Integral Molecular, PA, USA | Cat # LM048 |
| Goat anti-Human IgG Alexa Fluor 647, polyclonal, 1:500 (IF) | Thermo Fisher | Cat # A48279 |
| Goat anti-albumin, polyclonal, 1:500 (IF) | Bethyl Labs | Cat # A80-229A |
| Donkey anti-Goat IgG Alexa Fluor 488, polyclonal, 1:500 (IF) | Thermo Fisher | Cat # A-11055 |

| Reagent/Resource | Reference or Source | Identifier or Catalogue Number |
|---|---|---|
| Rabbit anti-human albumin, polyclonal (Albumin assay) | Agilent Dako | Cat # F0117 |
| Goat anti-Rabbit SULFO-TAG Labelled, polyclonal (Albumin assay) | Meso Scale Discovery | Cat # R32AB |
| **Oligonucleotides and sequence-based reagents** | | |
| qPCR Primers | This study | Table 1 |
| **Chemicals, enzymes and other reagents** | | |
| DMEM - high glucose | Sigma-Aldrich | Cat # D6546 |
| Foetal bovine serum | Gibco | Cat # 10270106 |
| GlutaMAX | Gibco | Cat # 35050-038 |
| Insulin from bovine pancreas | Sigma-Aldrich | Cat # I5500 |
| 3-isobutyl-1-methylxanthine | Sigma-Aldrich | Cat # I5879 |
| Dexamethasone | Sigma-Aldrich | Cat # D4902 |
| Biotin | Sigma-Aldrich | Cat # B4639 |
| DMEM/F12 | Invitrogen | Cat # 31330-038 |
| Collagenase from *Clostridium histolyticum* | Sigma-Aldrich | Cat # C0130 |
| Bovine serum albumin | Sigma-Aldrich | Cat # A2153 |
| Phosphate buffered saline | Sigma-Aldrich | Cat # 806552 |
| L-glutamine | Gibco | Cat # 25030081 |
| Penicillin/Streptomycin (10,000 U/mL) | Gibco | Cat # 15140122 |
| DMEM - low glucose | Sigma-Aldrich | Cat # D6046 |
| Human fibroblast growth factor 2 (FGF2) | Abcam | Cat # ab9596 |
| Rosiglitazone | Cayman | Cat # 71740 |
| Glucose-free DMEM | Sigma-Aldrich | Cat # D5030 |
| Sodium pyruvate | Gibco | Cat # 11360070 |
| Sodium bicarbonate | Sigma-Aldrich | Cat # S6014 |
| D-(+)-glucose | Sigma-Aldrich | Cat # G7021 |
| Chloroform, HPLC grade | Merck | Cat # 270636 |
| Methanol, HPLC grade | Merck | Cat # 34860 |
| Boron trifluoride-methanol solution | Sigma-Aldrich | Cat # B1252 |
| n-Hexane, HPLC grade | Fisher Scientific | Cat # 10499170 |
| Food Industry FAME Mix | Restek | Cat # 35077 |
| Pierce Protease and Phosphatase Inhibitor Mini Tablets | Thermo Scientific | Cat # A32959 |
| Pierce BCA Protein Assay Kit | Thermo Scientific | Cat # 23227 |
| 4x Laemmli Sample Buffer | Bio-Rad | Cat # 1610747 |
| Pierce TCEP-HCl | Thermo Scientific | Cat # 20490 |
| SuperSignal™ West Pico PLUS Chemiluminescent Substrate | Thermo Scientific | Cat # 34580 |
| Acetonitrile, LC-MS grade | Fisher Scientific | Cat # 10001334 |
| Valine-d8 | Cambridge Isotope Laboratories, Inc. | Cat # DLM-488-PK |
| Cytochalasin B | Sigma-Aldrich | Cat # C6762 |
| 2-Deoxy-D-glucose | Sigma-Aldrich | Cat # D3179 |
| Triton™ X-100 | Sigma-Aldrich | Cat # X100 |
| RNeasy Mini kit | Qiagen | Cat # 74104 |
| GoScript Reverse Transcriptase kit | Promega | Cat # A2801 |

| Reagent/Resource | Reference or Source | Identifier or Catalogue Number |
|---|---|---|
| SYBR Green Master Mix | Applied Biosystems | Cat # 4309155 |
| RNA STAT-60 | AMSBIO | Cat # CS-110 |
| Lysing Matrix D | MP Biomedicals | Cat # 116913050-CF |
| TruSeq® Stranded mRNA Library Prep | Illumina | Cat # 20020594 |
| Mouse Leptin Kit | Meso Scale Discovery | Cat # K152BYC-2 |
| Mouse Adiponectin Kit | Meso Scale Discovery | Cat # K152BXC |
| 16% Formaldehyde | Thermo Scientific | Cat # 28906 |
| Glycine | Millipore | Cat # 3570 |
| Normal swine serum | Jackson ImmunoResearch | Cat # 014-000-121 |
| Lectin-FITC | Sigma-Aldrich | Cat # L4895 |
| Hoechst 33342 | Invitrogen | Cat # H3570 |
| Glycerol | Sigma-Aldrich | Cat # G2025 |
| DABCO (1,4-Diazabicyclo[2.2.2]octane) | Sigma-Aldrich | Cat # D27802 |
| Free glycerol reagent | Sigma-Aldrich | Cat # F6428 |
| Triiodothyronine (T3) | Sigma-Aldrich | Cat # T6397 |
| Indomethacin | Sigma-Aldrich | Cat # I7378 |
| MEM α, nucleosides | Gibco | Cat # 12571063 |
| Insulin-Transferrin-Selenium | Thermo Fisher | Cat # 41400045 |
| Sodium Bicarbonate Solution 7.5% | Thermo Fisher | Cat # 25080060 |
| L-Ascorbic acid 2-Phosphate | Sigma-Aldrich | Cat # A8960 |
| Zebrafish FGF2 | Qkine | Cat # Qk002 |
| TGFß | R&D | Cat # 240-B-500/CF |
| StemPro™ Accutase™ Cell Dissociation Reagent | Thermo Fisher | Cat # A1110501 |
| DPBS, no calcium, no magnesium | Thermo Fisher | Cat # 14190250 |
| ROCK inhibitor (Y-27632) | Selleckchem | Cat # S1049 |
| IMDM | Thermo Fisher | Cat # 21980 |
| Chemically Defined Lipid Concentrate | Thermo Fisher | Cat # 11905031 |
| 1-Thio Glycerol (MTG) | Sigma-Aldrich | Cat # M6145 |
| Human Apo-Transferrin, CF | R&D | Cat # 3188-AT-001G |
| Insulin solution human | Sigma-Aldrich | Cat # I9278 |
| PVA | Sigma-Aldrich | Cat # P8136 |
| RPMI | Thermo Fisher | Cat # 61870010 |
| MEM Non-Essential Amino Acids Solution | Thermo Fisher | Cat # 11140035 |
| B27+insulin | Thermo Fisher | Cat # 17504044 |
| HepatoZYME-SFM | Thermo Fisher | Cat # 17705021 |
| BMP-4 | Qkine | Cat # Qk038 |
| LY29004 | Promega | Cat # V1201 |
| CHIR99021 | Stratech Scientific | Cat # S1263 |
| Hepatocyte Growth Factor (HGF) | Peprotech | Cat # 100-39 |
| Oncostatin M (OSM) | R&D | Cat # 295-OM-01M |
| Doxycycline | Sigma-Aldrich | Cat # 9891 |
| Vitronectin XF | Stem Cell Technologies | Cat # 07180 |
| 0.5 M EDTA | Invitrogen | Cat # AM9260G |
| Diluent 100 | Meso Scale Discovery | Cat # R50AA |
| P450-Glo™ CYP3A4 Assay kit | Promega | Cat # V9001 |

| Reagent/Resource | Reference or Source | Identifier or Catalogue Number |
|---|---|---|
| IL-4 | PeproTech | Cat # AF-214-14 |
| [1-14C]oleate | PerkinElmer | Cat # NEC317250UC |
| Hionic-Fluor scintillation liquid | PerkinElmer | Cat # 6013311 |
| mTeSR PLUS | Stem Cell Technologies | Cat # 100-0276 |
| ReLeSR | Stem Cell Technologies | Cat # 100-0483 |
| IWP-4 | Stem Cell Technologies | Cat # 72552 |
| L-Ascorbic acid 2-phosphate sesquimagnesium salt hydrate | Sigma-Aldrich | Cat # A8960 |
| Collagenase type I | Sigma-Aldrich | Cat # SCR103 |
| MEM α, GlutaMAX | Thermo Fisher | Cat # 32561037 |
| Antimycin A | Sigma-Aldrich | Cat # A8674 |
| Dichloroacetate | Sigma-Aldrich | Cat # 347795 |
| Cobalt chloride | Sigma-Aldrich | Cat # 255599 |
| CL316,243 | Tocris | Cat # 1499 |
| Indinavir | Sigma-Aldrich | Cat # SML0189 |
| MG132 | Sigma-Aldrich | Cat # 474790 |
| 13C6-glucose | Cambridge Isotope Laboratories, Inc. | Cat # CLM-1396-PK |
| Deuterium oxide | Sigma-Aldrich | Cat # 756822 |
| tridecanoic-d25 acid | Cambridge Isotope Laboratories, Inc. | Cat # DLM-1392-PK |
| Complete Essential 8 Medium (iPSC-neurons) | Thermo Scientific | Cat # A1517001 |
| DMEM/F12 with HEPES (iPSC-neurons) | Gibco | Cat # 11330032 |
| N2 supplement | Gibco | Cat # 17502048 |
| 1% NEAA | Gibco | Cat # 11140050 |
| poly-L-ornithine | Sigma-Aldrich | Cat # P3655 |
| BrainPhys neuronal medium | Stem Cell Technologies | Cat # 05790 |
| B27 supplement 50x (iPSC-neurons) | Gibco | Cat # 17504044 |
| BDNF | Peprotech | Cat # 450-02 |
| NT-3 | Peprotech | Cat # 450-03 |
| Laminin | Gibco | Cat # 23017015 |
| **Software** | | |
| Prism 9 | https://www.graphpad.com/features | |
| Lucid Lab | https://lab.lucidsci.com | |
| Agilent MassHunter Workstation Quantitative Analysis (version B.07.00) | https://www.agilent.com/en/product/software-informatics/mass-spectrometry-software/data-analysis/quantitative-analysis | |
| Agilent MassHunter Workstation Qualitative Analysis (version B.07.00) | https://www.agilent.com/en/product/software-informatics/mass-spectrometry-software/data-analysis/qualitative-analysis | |
| ImageJ2 | https://imagej.net/software/imagej2/ | |
| Image Lab (v 6.1) | https://www.bio-rad.com/en-uk/product/image-lab-software?ID=KRE6P5E8Z | |
| Matlab | https://www.mathworks.com/products/matlab.html | |
| R | https://www.r-project.org | |
| Compound Discoverer (v 3.2) | https://www.thermofisher.com/uk/en/home/industrial/mass-spectrometry/liquid-chromatography-mass-spectrometry-lc-ms/lc-ms-software/multi-omics-data-analysis/compound-discoverer-software.html | |
| Tracefinder (v 5.0) | https://www.thermofisher.com/uk/en/home/industrial/mass-spectrometry/liquid-chromatography-mass-spectrometry-lc-ms/lc-ms-software/lc-ms-data-acquisition-software/tracefinder-software.html | |
| FastQC (v. 0.11.9) | http://www.bioinformatics.babraham.ac.uk/projects/fastqc/ | |

| Reagent/Resource | Reference or Source | Identifier or Catalogue Number |
|---|---|---|
| hisat2 (v2.1.0) | http://daehwankimlab.github.io/hisat2/ | |
| HTseq-count (v 0.11.1) | https://htseq.readthedocs.io/en/release_0.11.1/ | |
| DESeq2 (v 3.15) | https://bioconductor.org/packages/release/bioc/html/DESeq2.html | |
| Harmony High-Content Imaging and Analysis Software | https://www.perkinelmer.com/uk/product/harmony-4-9-office-license-hh17000010 | |
| MSD Discovery Workbench | https://www.mesoscale.com/en/products_and_services/software | |
| **Other** | | |
| Tecan Spark 10 M Plate Reader | Tecan | |
| Resipher | Lucid Scientific | |
| 24-well Lumox plates | Sarstedt | |
| OxoPlates | PreSens | |
| Agilent 7890B gas chromatography system | Agilent | |
| Agilent 5977 A mass spectrometer | Agilent | |
| Tecan M1000 Pro Plate Reader | Tecan | |
| Oroboros Oxygraph-2K | Oroboros Instruments, Innsbruck, Austria | |
| Chemidoc MP | Bio-Rad | |
| Millipore Sequant ZIC-pHILIC analytical column (5 μm, 2.1 × 150 mm) | Merck Millipore | |
| Vanquish Horizon UHPLC | Thermo Fisher | |
| Orbitrap Exploris 240 mass spectrometer | Thermo Fisher | |
| TriCarb 2900TR | PerkinElmer | |
| ABI QuantStudio 5 | Thermo Fisher | |
| Whitley H35 Hypoxystation | Don Whitley Scientific | |
| Agilent Bioanalyser 2100 | Agilent | |
| Illumina NovaSeq6000 | Illumina | |
| Opera Phenix | PerkinElmer | |
| MSD Meso Sector S600 Plate Reader | Meso Scale Discovery | |
| Zeiss inverted 710 confocal microscope | Zeiss | |
| GloMax plate reader | Promega | |
| Heart-Dyno | Mills et al, 2017 PNAS | |
| Nikon ANDOR WD Revolution Spinning Disk microscope | Nikon | |

## Methods and protocols

### Cell culture of 3T3-L1 adipocytes

1. 3T3-L1 fibroblasts were cultured in high glucose DMEM supplemented with 10% foetal bovine serum (FBS) and 2 mM glutamax at 37 °C in 10% $CO_2$.
2. For hypoxia experiments, cells were cultured at 37 °C in 5% $O_2$, 5% $CO_2$.
3. For differentiation into adipocytes, fibroblasts were cultured in 10% FBS-supplemented DMEM containing an adipogenic cocktail (350 nM insulin, 0.5 mM 3-isobutyl-1-methylxanthine, and 250 nM dexamethasone for 3 days.
4. Then, the cells were cultured in in DMEM containing 10% FBS and 350 nM insulin for another 3 days.
5. Differentiated adipocytes were maintained in 10% FBS-supplemented DMEM.
6. Adipocytes were used for experiments 9–10 d after the onset of differentiation, with the culture medium renewed every 2 days prior to each experiment.
7. For low medium volume treatments, cells were cultured in either 1000 μL (high), 666 μL (mid), 333 μL (low) medium in a 12-well plate, or 500 μL (high), 333 μL (mid), or 167 μL (low) in a 24-well plate (Fig. EV1A) for the duration specific to each experiment.

3T3-L1 cells tested negative for mycoplasma and were not authenticated, but they were differentiated into adipocytes and only used if they reached >90–95% differentiation as observed by lipid accumulation.

## Cell culture of primary subcutaneous adipocytes (scAdips)

Pre-adipocytes/mesenchymal stem cells (MSC) were isolated from subcutaneous white adipose tissue (scWAT) of 6–12-week old male and female C57BL/6 J mice. Animal work was carried out in the Disease Model Core, and was regulated under the Animals (Scientific Procedures) Act 1986 Amendment Regulations 2012 following ethical review by the University of Cambridge Animal Welfare and Ethical Review Body according to UK Home Office guidelines.

Extraction of preadipocytes/MSCs:

1. The tissues were minced with scissors and transferred to a tube containing 1 mL of digestion buffer (DMEM/F12, 1 mg/mL collagenase Type II, 1% bovine serum albumin).
2. After that, 1 mL of PBS was added and the tissues vortexed for 10 s.
3. The tubes were sealed with parafilm and shaken at 37 °C, 150 cycles/min, for 40 min. In between, the tubes were vortexed every 10 min for 10 s.
4. After 40 min, 500 μL FBS was added, and the digested tissues were passed through a 100 μm filter.
5. 5 mL PBS was used to rinse any unfiltered tissue.
6. The filtered tissues were centrifuged at $600 \times g$ for 5 min, and the overlying medium aspirated, leaving a cell pellet.
7. The pellet was resuspended in 500 μL of 1 x red blood cell lysis buffer and incubated for 3–5 min on ice.
8. After that, 5 mL of maintenance media (MM) containing DMEM/F12, 10% FBS, 2 mM L-glutamine, 1% P/S was added and the tissues centrifuged again at $600 \times g$ for 5 min.
9. After centrifugation, the overlying medium was aspirated, and the cells were resuspended again in 5 mL MM and seeded into a T25 flask.

Differentiation of scAdips:

1. After two passages, cells were seeded at a density of $4.0 \times 10^5$ cells/well into 12-well plates and grown to confluence in "maintenance medium" (10% FBS-supplemented 25 mM glucose DMEM containing 720 nM insulin, 1% P/S, 2 mM L-glutamine).
2. Once confluent (usually ~3 days post-seeding), the medium was then changed to "induction medium" (day 0), containing 0.65 mM IBMX and 1.3 μM dexamethasone in maintenance medium.
3. After 48 h, the cells were fed with the maintenance medium again.
4. The maintenance medium was changed every two days until complete differentiation (about 10 days).

## Cell culture of human adipocytes (hMADS)

hMADS adipocytes were kindly provided by the Centre National de la Recherche Scientifique (Ez-Zoubir Amri, iBV, Nice, France), and cultured as previously described (Rodriguez et al, 2005; Pisani et al, 2018).

1. hMADS were maintained in proliferation medium (DMEM low glucose, 10% FBS, 2 mM glutamax, 1% P/S, supplemented with 2.5 ng/mL of human fibroblast growth factor 2 (hFGF2)).
2. The cells were seeded into 12-well plates at a density of 44,000 cells/mL and kept at 37 °C, 5% $CO_2$.
3. Six days post-seeding, hFGF2 was removed from the proliferation medium.
4. On the next day (day 0), the cells were incubated in differentiation medium (DM; serum-free proliferation medium/Ham's F12 medium containing 10 μg/mL transferrin, 10 nM insulin, 0.2 nM triiodothyronine, 500 μM 3-isobutyl-1-methyl-xanthine, 1 μM dexamethasone and 100 nM rosiglitazone).
5. On day 3, dexamethasone and 3-isobutyl-1-methylxanthine were omitted from DM, and on day 9, rosiglitazone was also omitted.
6. The experiments were carried out between days 12 and 15.

## Measurement of extracellular glucose and lactate concentrations

All cells cultured in 12-well plates were fed with 1 mL (high), 0.67 mL (mid), or 0.33 mL (low) of fresh medium for 16 h or the equivalent volumes in other well sizes as shown in Fig. EV1A unless stated otherwise. iPSC-derived hepatocytes were fed with 1 mL (high) or 0.5 mL (low) of fresh medium for 24 h. hPSC-derived cardiac organoids were fed with 150 μL (high) or 50 μL (low) of fresh medium for 48 h. For experiments using different starting concentrations of glucose, the appropriate amount of D-(+)-glucose was added to glucose-free DMEM (Sigma #D5030) supplemented with 1 mM sodium pyruvate, 10% FBS, 2 mM glutamax, and 44 mM sodium bicarbonate. Following low medium volume interventions, media was collected from wells and sent for glucose consumption/lactate production analysis at the Core Biochemical Assay Laboratory (Addenbrooke's Hospital, Cambridge). Naïve medium was used as a baseline. Medium glucose was measured using an adaption of the hexokinase-glucose-6-phosphate dehydrogenase method described by Kunst et al, (1983) (Siemens Healthcare (product code DF30)). Medium lactate was measured by monitoring absorbance at 340 nm due to NADH production as L-lactate is oxidised to pyruvate by lactate dehydrogenase (Siemens Healthcare (product code DF16)). Calculated glucose use and lactate production were normalised to well areas (except in the case of cardiac organoids). Medium lactate from the cardiac organoids were measured using an in-house enzymatic assay, based on the hydrazine-sink method as described previously (Krycer et al, 2020). Cardiac data were normalised to account for data variability between batches.

## Measurements of pericellular oxygen concentration and oxygen consumption rate

3T3-L1 adipocytes were cultured in Falcon flat-bottom 96-well microplate (Corning #353072) with 100 μL medium throughout differentiation. Oxygen consumption rates (OCR) and oxygen concentration were continuously measured using Resipher (Lucid Scientific) at 37 °C, 10% $CO_2$ over 2–3 days starting from 8 d post-differentiation. The Resipher oxygen sensing lid contains micro probes with optical oxygen sensors that scan between 0.55–0.95 mm above the cells to measure the oxygen concentration gradient within the medium column. Baseline OCR and oxygen concentrations under 100 μL medium were measured for 4–6 h in

each well. Then, medium volumes were changed and replenished every 24 h. About 100 nM antimycin A treatment and trypsinised cells for medium-only wells were used as controls. Data were analysed using the Resipher web application.

## Relative measurements of pericellular oxygen concentration

3T3-L1 cells were differentiated in 96-well OxoPlates (PreSens), which have optical oxygen sensors integrated at the bottom of each well. Prior to starting the experiment, different volumes of fresh medium were added to each well, and the plate was immediately placed into the Tecan Spark 10 M Plate Reader set to 37 °C, 10% $CO_2$ for 24 h. The filters were set up and oxygen levels were calculated according to the manufacturer's protocol. Relative oxygen levels were normalised to the 100 μL condition.

## 3T3-L1 culture in gas-permeable (Lumox) plates

1. Differentiated 3T3-L1 cells were reseeded on day 6 post-differentiation into 24-well plates (control), or 24-well Lumox plates (Sarstedt), which consist of a black polystyrene frame with a foil base made from a thin (50 μm), gas-permeable film.
2. The medium volume was kept at 500 μL throughout the differentiation protocol.
3. On day 9 post-differentiation, 500 μL fresh medium was added and collected after 16 h for glucose and lactate quantification, or cells were scraped for lipid extraction as detailed in the other sections.

## Deuterium tracing and lipid extraction

1. Prior to lipid extraction, cells were cultured in DMEM containing 8% $^2H$ (deuterium)- labelled water, in a 37 °C, 10% $CO_2$ incubator containing 8% $^2H$-labelled water for 16 h.
2. Cells were detached with 100 μL ice-cold PBS and lipids were extracted and derivatised as fatty acid methyl esters (FAMEs).
3. Lipids were extracted according to a modified Folch method (Folch et al, 1957) using 50 μM tridecanoic-d25 acid (Cambridge Isotope Laboratories, Inc.) as an internal standard.
4. 1 mL of HPLC-grade chloroform/methanol 2:1 v/v mixture was added to cell samples in a glass vial.
5. Samples were homogenised by vortexing for 5 min.
6. 200 μL $H_2O$ was added to each sample before vortexing again for 5 min and centrifuging at $4000 \times g$ for 10 min.
7. 600 μL of the lower lipid fraction was transferred to a 7 mL glass tube.
8. A second extraction was performed by 11 adding 600 μL fresh chloroform followed by vortexing and centrifugation as above.
9. Another 600 μL of lower lipid fraction was collected and pooled with the first 600 μL fraction (total = 1200 μL).
10. The collected organic fraction was dried under nitrogen steam and stored at −80 °C for subsequent gas chromatography-mass spectrometry (GC-MS) analysis.

## Fatty acid methyl ester derivatisation and GC-MS analysis

FAME derivatisation:

1. Dried lipids were dissolved in a mixture of 410 μL methanol, 375 μL chloroform, and 90 μL 14% $BF_3$ for the FAMEs, and incubated at 80 °C for 90 min.
2. 500 μL $H_2O$ and 1000 μL hexane were then added to each sample, homogenised by vortexing briefly, and centrifuged at 2000 rpm for 5 min at room temperature.
3. 1 mL of the organic upper layer was then transferred to an autosampler vial for GC-MS analysis.

GC-MS analysis:

1. GC-MS was performed with Agilent 7890B gas chromatography system linked to Agilent 5977 A mass spectrometer, using an AS3000 autosampler and data was acquired using MassHunter Workstation Software.
2. A TR-FAME column (length: 30 m, inner diameter: 0.25 mm, film size: 0.25 μm, 260M142P, Thermo Fisher Scientific) was used with helium as carrier gas.
3. Inlet temperature was set at 230 °C. Dried FAME samples were re - suspended in 200 μL HPLC-grade n-Hexane. 1 μl of this solution was injected for analysis.
4. The oven programme used for separation was as follows: 100 °C hold for 2 min, ramp at 25 °C/min to 150 °C, ramp at 2.5 °C/min to 162 °C and hold for 3.8 min, ramp at 4.5 °C/min to 173 °C and hold for 5 min, ramp at 5 °C/min to 210 °C, ramp at 40 °C/min to 230 °C and hold for 0.5 min.
5. Carrier gas flow was set to constant 1.5 mL/min.
6. If the height of any FAME peaks exceeded 108 units, the sample was re-injected with a 10:1–100:1 split ratio.
7. Identification of FAME peaks was based on retention time and made by comparison with those in external standards (Food industry FAME mix, 35077, Restek).
8. Peak integration and quantification were performed using MassHunter Workstation Quantitative Analysis software (version B.07.00, Agilent).
9. Specific high-abundance ions from total ion chromatograms were chosen to calculate each fatty acid peak.

## Mass isotopomer distribution analysis (MIDA)

To calculate de novo lipogenesis rates from fatty acids, MIDA was performed according to the following protocol. In order to perform the analyses we extracted the $M + 0$ to $M + 4$ ions (e.g. for palmitate methyl ester (m/z 270–274)). From these, we calculated the fractional concentration of each ion. All equations below use fractional concentrations.

We determined a theoretical distribution for a newly synthesised molecule that was dependent on the precursor labelling pool p, whereby p was the fraction of deuterated water (i.e. 0.08). Firstly, we calculated the isotopomer pattern of each fatty acid caused by the presence of additional $^2H$ atoms in the molecule. The number of available sites in a fatty acid that could be deuterated, N, was

determined based on model fit (e.g. for palmitate in 3T3-L1 adipocytes, $N = 14$).

Ions dependent on deuterium incorporation were called M'. For each ion M'0-M'4 the following equation gave the expected relative abundance:

For M'x:

(14!/x!*(14-x!)) * px * (1-p)(14-x)

The pattern of labelling based on deuterium was then corrected for the presence of naturally occurring oxygen and carbon isotopes using the following equations for ions M0n-M4n, where Mxn stood for the ion of a newly synthesised fatty acid molecule:

M' = labelling due to deuterium
M = natural isotopic labelling
M0n = M'0*M0
M1n = M'0*M1 + M'1*M0
M2n = M'0*M2 + M'1*M1 + M'2*M0
M3n = M'0*M3 + M'1*M2 + M'2*M1 + M'3*M0
M4n = M'0*M4 + M'1*M3 + M'2*M2 + M'3*M1 + M'4*M0

These equations provided us with the ion pattern for a newly synthesised fatty acid M0n-M4n.

To determine the relative contribution of newly synthesised fatty acids and existing fatty acids within the cell, we set the following equations, where M0n is newly synthesised, M0 was endogenous pre-existing fatty acids, and M0obs was the observed M0 fractional concentration of the ion measured using the mass spectrometer. M0obs' was the calculated fraction of M0 based on combining newly synthesised and pre-existing fatty acid as follows:

M0obs' = f *M0n + (1-f)*M0
M1obs' = f *M1n + (1-f)*M1
M2obs' = f *M2n + (1-f)*M2
M3obs' = f *M3n + (1-f)*M3
M4obs' = f *M4n + (1-f)*M4

We then calculated M0obs-M0obs' through to M4obs-M4obs' and calculated the sum of squares for these 4 equations. We used the GRG Non-Linear Engine of the Solver function of Excel to minimise the sum of squares of M0obs-M0obs'+…M4obs-M4obs' by changing the values of $p$ and $f$. This enabled us to calculate the fractional synthesis rate for each fatty acid. The $N$ number was adjusted until the $p$ was 0.08 (8% deuterated water), which was experimentally set, thus providing the $N$ number. The amount of newly synthesised palmitate was calculated by multiplying the $f$ value by the concentration of the fatty acid in the well.

## Cell density assay

1. On day 6 post-differentiation, cells were trypsinized and reseeded into 12-well plates or 96-well plates at different densities.
2. To quantify seeding density, cells were cultured until day 10 post-differentiation, washed 3 times with cold PBS and scraped in 500 μL lysis buffer (1% Triton in PBS) for DNA quantification.
3. Samples were freeze-thawed three times and sonicated for 10 s at 30% power.
4. Then, samples were centrifuged at $16,000 \times g$ for 5 min at 4 °C to remove cell debris.
5. 10 μL of each sample, as well as a standard curve made from salmon sperm DNA (0–25 μg/mL) was loaded in duplicates into a black-bottom, black-walled 96-well plate.

6. Then, 200 μL of SYBR Green assay buffer (10,000 x dilution of SYBR Green in TNE buffer) was added to all samples.
7. After 5 min incubation at room temperature, the fluorescence intensity was measured using a Tecan M1000 Pro Plate Reader at 497 (ex)/ 520 (em).

## Measurements of oxygen consumption rate in a closed system

High-resolution $O_2$ consumption measurements were conducted using the Oroboros Oxygraph-2K (Oroboros Instruments, Innsbruck, Austria) in intact and digitonin-permeabilised cells.

1. For the cell experiment, cells were centrifuged at $300 \times g$ for 7 min at room temperature, washed in PBS, centrifuged once more and then suspended in assay medium at a cell concentration of $\sim 1 \times 10^6$ viable cells/mL.
2. For experiments designed to assess basal respiratory kinetics in intact cells, differentiated 3T3-L1 adipocytes were suspended in bicarbonate-free DMEM, supplemented with Glutamax (glucose 4.5 g/L).
3. Respiration and oxygen tension were continuously evaluated until the cells consumed all the available oxygen.
4. For experiments in permeabilized cells, intact 3T3-L1 adipocytes, exposed to low or high medium for 16 h, were suspended in respiration buffer. Respiration buffer consisted of potassium-MES (105 mM; pH 7.2), KCl (30 mM), $KH_2PO_4$ (10 mM), $MgCl_2$ (5 mM), EGTA (1 mM), and BSA (2.5 g/L).
5. Following basal respiration, cells were permeabilised with digitonin (0.02 mg/mL).
6. Physiological ATP-free energy ($-54.16$ kJ/mol) was applied via the creatine kinase (CK) clamp (Fisher-Wellman et al, 2018).
7. Cytochrome $c$ (0.01 mM) was added to assess the integrity of the outer mitochondrial membrane, followed by sequential additions of carbon substrates and respiratory inhibitors [pyruvate/malate (Pyr/Mal 5 mM/1 mM); Glutamate (10 mM); octanoyl-carnitine (Oct 0.2 mM); rotenone (Rot 0.5 μM); Succinate (10 mM); Malonate (10 mM); Calcium (0.6 μM); glycerol-3-phosphate (G3P 10 mM); Antimycin (0.5 μM)].
8. Data were normalised to total protein.

## Immunoblotting

1. Cells were washed three times in ice-cold PBS and lysed in RIPA buffer containing protease and phosphatase Inhibitors (Thermo Fisher).
2. Scraped lysates were then sonicated and centrifuged at $16,000 \times g$ at 4 °C for 30 min.
3. Protein concentration of the supernatant was quantified using the BCA Assay (Thermo Scientific).
4. Lysates were diluted in 4x Laemmli Sample Buffer (Bio-Rad), reduced with TCEP (Thermo Fisher) and heated at 37 °C for 30 min (Figs. 2C and EV2A; GLUT1, GLUT4, mitochondrial respiratory complexes), or 65 °C for 10 min (Fig. 2D; pPDH, PDH).

5. 10–20 µg of protein was resolved by SDS-PAGE and transferred to nitrocellulose membranes (Bio-Rad).

6. Membranes were blocked in 5% skim milk powder in Tris-buffered saline for 1 h at room temperature, followed by overnight incubation at 4 °C with specific primary antibody solutions.

7. Subsequently, membranes were incubated with the appropriate secondary antibodies for 1 h at room temperature before signals were detected using enhanced chemiluminescence (ECL) (Thermo Scientific) on the Chemidoc MP (Bio-Rad).

8. Bands were quantified using ImageJ2.

Antibodies used are listed in the Reagents and Tools table.

## Immunoblotting of HIF1α

1. Cells were washed three times in ice-cold PBS and lysed in 2% SDS containing protease and phosphatase Inhibitors (Thermo Fisher), 500 µM $CoCl_2$, and 10 µM MG132.

2. Scraped lysates were then sonicated and centrifuged at $16,000 \times g$ at room temperature for 30 min.

3. Protein concentration of the supernatant was quantified using the BCA Assay (Thermo Scientific).

4. Lysates were diluted in 4x Laemmli Sample Buffer (Bio-Rad), reduced with TCEP (Thermo Fisher) and heated at 95 °C for 10 min.

5. 40 µg of protein was resolved by SDS-PAGE and transferred to nitrocellulose membranes (Bio-Rad).

6. Membranes were blocked in 5% skim milk powder in Tris-buffered saline for 1 h at room temperature, followed by an overnight incubation at 4 °C with anti-HIF1α (1:500, #14179 Cell Signaling Technology).

7. Subsequently, membranes were incubated with the goat anti-rabbit IgG for 1 h at room temperature before signals were detected using ECL (Thermo Scientific) on the Chemidoc MP (Bio-Rad).

8. Bands were quantified using ImageJ2.

9. For detecting the rate of HIF1α degradation (Fig. 3A), 0.67 mL of medium was carefully removed from 1 mL without agitating the plate to prevent reoxygenation, leaving 0.33 mL as the low medium volume.

## Assessment of PDH flux

PDH flux was assessed using the FASA package implemented in Matlab (Argus et al, 2018). FASA was performed on isotope-corrected isotopomer distributions for palmitate (M0-M16). Isotope correction was performed using the IsoCorR package implemented in R. The value "D" from the FASA analysis was the proportion of lipogenic acetyl-CoA derived from glucose that was used to produce palmitate.

## ¹³C-glucose tracing and metabolite extraction

1. Prior to the experiment, 3T3-L1 adipocytes were washed once with glucose-free DMEM and labelled with 25 mM 13C-glucose in DMEM for either 4 or 16 h.

2. Then, the medium was removed from the wells and cells were washed twice with room temperature PBS and placed on dry ice.

3. 250 µL metabolite extraction buffer (50% methanol, 30% acetonitrile, 20% ultrapure water, 5 µM valine-d8) was added to each well of a 12-well plate and incubated for 5 min on a dry ice-methanol bath to lyse the cell membranes.

4. After that, extracts were scraped and mixed at 4 °C for 15 min in a thermomixer at 2000 rpm.

5. After final centrifugation at maximum speed for 20 min at 4 °C, 80 µL of the supernatant from each sample was transferred into labelled LC-MS vials.

## LC-MS analysis

Hydrophilic separation of metabolites was achieved using a Millipore Sequant ZIC-pHILIC analytical column (5 µm, 2.1 × 150 mm) equipped with a 2.1 × 20 mm guard column (both 5 mm particle size) with a binary solvent system. Solvent A was 20 mM ammonium carbonate, 0.05% ammonium hydroxide; Solvent B was acetonitrile. The column oven and autosampler tray were held at 40 and 4 °C, respectively. The chromatographic gradient was run at a flow rate of 0.200 mL/min as follows: 0–2 min: 80% B; 2–17 min: linear gradient from 80% B to 20% B; 17–17.1 min: linear gradient from 20% B to 80% B; 17.1–23 min: hold at 80% B. Samples were randomised and the injection volume was 5 µl. A pooled quality control (QC) sample was generated from an equal mixture of all individual samples and analysed interspersed at regular intervals.

Metabolites were measured with Vanquish Horizon UHPLC coupled to an Orbitrap Exploris 240 mass spectrometer (both Thermo Fisher Scientific) via a heated electrospray ionisation source. The spray voltages were set to +3.5 kV/−2.8 kV, the RF lens value at 70, the heated capillary held at 320 °C, and the auxiliary gas heater held at 280 °C. The flow rate for sheath gas, aux gas and sweep gas were set to 40, 15 and 0, respectively. For MS1 scans, the mass range was set to $m/z = 70–900$, the AGC target was set to standard and the maximum injection time (IT) set to auto. Data acquisition for experimental samples used full scan mode with polarity switching at an Orbitrap resolution of 120,000. Data acquisition for untargeted metabolite identification was performed using the AcquireX Deep Scan workflow, an iterative data-dependent acquisition (DDA) strategy using multiple injections of the pooled sample. DDA full scan-ddMS2 method for AcquireX workflow used the following parameters: full scan resolution was set to 60,000, fragmentation resolution to 30,000, and fragmentation intensity threshold to 5.0e3. Dynamic exclusion was enabled after 1 time, and the exclusion duration was 10 s. Mass tolerance was set to 5 ppm. Isolation window was set to 1.2 $m/z$. Normalised HCD collision energies were set to stepped mode with values at 30, 50 and 150. Fragmentation scan range was set to auto, AGC target at standard and max IT at auto. Mild trapping was enabled.

Metabolite identification was performed in the Compound Discoverer software (v 3.2, Thermo Fisher Scientific). Metabolite identities were confirmed using the following parameters: (1) precursor ion m/z was matched within 5 ppm of theoretical mass predicted by the chemical formula; (2) fragment ions were matched within 5 pm to an in-house spectral library of authentic compound standards analysed with the same ddMS2 method with a best match score of over 70; (3) the retention time of metabolites was within 5% of the retention time of a purified standard run with the same chromatographic method. Chromatogram review and peak area integration were performed using the Tracefinder software (v 5.0,

Thermo Fisher Scientific), and the peak area for each detected metabolite was normalised against the total ion count (TIC) of that sample to correct any variations introduced from the sample handling to instrument analysis. The normalised areas were used as variables for further statistical data analysis.

For $^{13}C_6$-glucose tracing analysis, the theoretical masses of $^{13}C$-labelled isotopes were calculated and added to a library of predicted isotopes in Tracefinder 5.0. These masses were then searched with a 5-ppm tolerance and integrated only if the peak apex showed less than 1% deviation in retention time from the [U-$^{12}C$] monoisotopic mass in the same chromatogram. The raw data obtained for each isotopologue were then corrected for natural isotope abundances using the AccuCor algorithm (https://github.com/lparsons/accucor) before further statistical analysis.

### $^3$H-2-deoxyglucose uptake measurements

1. 3T3-L1 adipocytes were differentiated in 24-well plates with 500 µL medium throughout differentiation.
2. 48 h prior to the assay, medium volumes were changed to either 500 µL (high) or 167 µL (low). On the day of the assay, cells were serum-starved for 2 h with either 500 or 167 µL DMEM containing 0.2% BSA at 37 °C, 10% $CO_2$.
3. Following 2 h serum-starvation, cells were washed and incubated in pre-warmed Krebs–Ringer phosphate (KRP) buffer containing 0.2% bovine serum albumin (BSA, Bovostar, Bovogen) (KRP buffer; 0.6 mM $Na_2HPO_4$, 0.4 mM $NaH_2PO_4$, 120 mM NaCl, 6 mM KCl, 1 mM $CaCl_2$, 1.2 mM $MgSO_4$ and 12.5 mM Hepes (pH 7.4)) for 10 min.
4. Again, KRP volumes were maintained at either 500 or 167 µL, and 200 µM indinavir (Murata et al, 2002) was added to at this point where indicated.
5. Adipocytes were then stimulated with 100 nM insulin for 20 min.
6. To determine non-specific glucose uptake, 25 µM cytochalasin B (in ethanol, Sigma-Aldrich) was added to control wells before addition of 2-[$^3$H]deoxyglucose (2-DG).
7. During the final 5 min, 2-DG (0.25 µCi, 50 µM) was added to cells to measure steady-state rates of 2-DG uptake.
8. Note that the volumes of 2-DG added were altered to account for different medium volumes.
9. Cells were then moved to ice, washed with ice-cold PBS, and solubilised in PBS containing 1% (v/v) Triton X-100. Tracer uptake was quantified by liquid scintillation counting on the TriCarb 2900TR (PerkinElmer), and data normalised for protein content.

### qRT-PCR

3T3-L1 adipocytes and scAdips were cultured using 1 mL medium (12-well plate) throughout differentiation. Medium volumes were changed to either 1 mL or 0.33 mL 16 h prior to the assay. Hepatocytes were cultured in either 1 mL or 0.5 mL medium (12-well plate) throughout differentiation, and changed to either 1 mL or 0.5 mL 24 h prior to the assay. RNA extractions were performed using the RNeasy Mini kit (Qiagen 1152 #74104). Concentrations of RNA samples were quantified using NanoDrop. cDNA synthesis from 500 ng RNA was performed using the GoScript Reverse Transcriptase kit (Promega #A2801). Real-time (RT)-polymerase chain reaction (PCR) was performed with SYBR Green Master Mix on the ABI QuantStudio 5. qPCR primers used in this study are listed in Table 1.

### 1% hypoxia chamber experiments

All 1% hypoxic experiments were performed in a Whitley H35 Hypoxystation (Don Whitley Scientific) maintained at 1% oxygen, 94% $N_2$ and 5% $CO_2$ at 37 °C. All media used were equilibrated in the chamber for 24 h, and all harvesting was also conducted in the chamber.

### 5% hypoxia incubator experiments

All 5% hypoxic experiments were performed in a standard incubator maintained at 5% oxygen, 90% $N_2$, 5% $CO_2$ at 37 °C. Cells were exposed to brief reoxygenation during medium changes and harvesting.

### Subcutaneous white adipose tissue (scWAT) from normoxic and hypoxic mice

Animal work was carried out in accordance with United Kingdom Home Office regulations under the Animals in Scientific Procedures (1986) Act and underwent review by the University of Cambridge Animal Welfare and Ethical Review Board. All procedures involving live animals were carried out by a personal licence holder in accordance with these regulations.

Subcutaneous adipose tissue (scWAT) was collected from 80-day-old male mice exposed to either 10% $O_2$ or maintained under normoxic atmospheric conditions, from a larger experimental cohort as previously published (O'Brien et al, 2019; Horscroft et al, 2019). 129Ev/Sv mice were housed in conventional cages from birth in a temperature- (23 °C) and humidity-controlled environment with a 12 h photoperiod and given *ad libitum* access to standard rodent chow (RM1(E), Special Diet Services, UK) and water. At 7 weeks of age, mice were randomly assigned to remain under normoxic conditions, or transferred to a normobaric hypoxia chamber (10% $O_2$; PFI systems, Milton Keynes, UK). Mice were maintained in hypoxia or normoxia for 28 days, after which they were killed by cervical dislocation. The scWAT was removed from the inguinal region, snap-frozen and stored at -80 °C until further analysis. The inguinal lymph node was removed from scWAT depots prior to RNAseq analysis.

### RNA extraction and sequencing

3T3-L1 adipocytes were seeded and differentiated in 12-well plates. RNA extraction from adipocytes was performed using the RNeasy Mini Kit (Qiagen 1152 #74104).

1. RNA extraction from mice scWAT was performed by homogenisation in RNA STAT-60.
2. Samples were broken up using a pestle and mortar under liquid nitrogen.
3. The broken pieces were then put in 2 mL screw-top tubes with Lysing Matrix D (MP Biomedicals).
4. 1 mL STAT-60 was added to the tissue samples, followed by homogenisation using Precellys 24 Touch.
5. The homogenate was then transferred to 1.5 mL eppendorfs.
6. Tubes were spun at 13,000 rpm for 5 min at 4 °C to remove debris.

**Table 1. qPCR primer sequences.**

| Gene | Species | Forward primer | Reverse primer |
|------|---------|----------------|----------------|
| Slc16a3 | Mouse | CTTGGATCTCCTCCATCC | GGTGAGGTAGATCTGGATAA |
| Pgk1 | Mouse | GGATGTTCTGTTCTTGAAG | CCTTCTTCCTCTACATGAA |
| Pdk1 | Mouse | AGTCGCATCTCAATTAGA | TCGCAGTTTGGATTTATG |
| Car9 | Mouse | CCTTCTCTTTGCTGTAC | GCTCCAGTTTCTGTCATC |
| Pkm2 | Mouse | TGCTGAAGGAGATGATTAAG | CCTTGATGAGTCCAGTCC |
| Slc2a1 | Mouse | GCTTCCTGCTCATCAATCGTAAC | CATCGGCTGTCCCTCGAA |
| Actin | Mouse | GCTCTGGCTCCTAGCACCAT | GCCACCGATCCACACAGAGT |
| 36B4 | Mouse | AGATGCAGCAGATCCGCAT | GTTCTTGCCCATCAGCACC |
| PGK1 | Human | CTGACAAGTTTGATGAGAATG | GCCTCAGCATACTTCTTG |
| PKM2 | Human | GAGGTGGAGCTGAAGAAG | CAGGATGTTCTCGTCACA |
| GAPDH | Human | GGAAGCTTGTCATCAATG | CCCCACTTGATTTTGGAG |
| LDHA | Human | CCGATTCCGTTACCTAATG | GCAGAGTCTTCAGAGAGA |
| SLC2A1 | Human | CACTGGAGTCATCAATGC | AAGCGGTTAACGAAAAGG |
| PDK1 | Human | CCGAACTAGAACTTGAAG | CGTGACATGAACTTGAATA |
| ALB | Human | CCTTTGGCACAATGAAGTGGGTAACC | CAGCAGTCAGCCATTTCACCATAG |
| SERPINA1 | Human | CCACCGCCATCTTCTTCCTGCCTGA | GAGCTTCAGGGGTGCCTCCTCTG |
| HNF-4α | Human | CATGGCCAAGATTGACAACCT | TTCCCATATGTTCCTGCATCAG |
| Cd206 | Mouse | GCATGGGTTTTACTGCTACTTGATT | CAGGAATGCTTGTTCATATCTGTCTT |
| Tnfα | Mouse | CATCTTCTCAAAATTCGAGTGACAA | TGGGAGTAGACAAGGTACAACCC |

7. The supernatant was removed and transferred to fresh tubes containing 200 µL of Chloroform.

8. Samples were vortexed at maximum speed for 15 s, and spun at $12{,}000 \times g$ for 15 min at 4 °C.

9. A clear aqueous supernatant of around 600 µL for every 1 mL of STAT-60 added will have formed.

10. The aqueous phase was transferred to fresh tubes containing 500 µL of isopropanol.

11. Tubes were mixed by inversion several times and left at −20 °C overnight.

12. After overnight incubation, the precipitates were spun at $12{,}000 \times g$ for 10 min at 4 °C to pellet RNA.

13. After removing the supernatant, 1 mL of cold 70% ethanol (stored at −20 °C) was added to the pellet and washed by vortexing briefly.

14. Samples were then spun at $8000 \times g$ for 5 min at 4 °C before discarding the supernatant.

15. Pellets were air-dried for 10–15 min on ice until transparent.

16. Nuclease-free water was added to the pellet and incubated at 60 °C for 10 min.

17. Samples were placed on ice and re-suspended by pipetting up and down.

18. RNA samples from both 3T3-L1 adipocytes and mice scWAT were quantified using nanodrop.

19. 1 µg of total RNA was quality checked (RIN >7) using an Agilent Bioanalyser 2100 system and then used to construct barcoded sequencing libraries with Illumina's TruSeq Stranded mRNA Library Prep Kit following manufacturer's instruction.

20. All the libraries were then multiplexed and sequenced on one lane of Illumina NovaSeq6000 at PE50 at the CRUK Cambridge Institute Genomics Core Facility.

## RNAseq data analysis

The NGS data were processed through a customised pipeline. FastQC software (v. 0.11.9) was used for generating quality-control reports of individual FASTQ files (http://www.bioinformatics.babraham.ac.uk/projects/fastqc). RNA sequencing reads were aligned to the *mus musculus* (mouse) GRCm38 genome using hisat2 (v2.1.0) (Kim et al, 2015). HTseq-count (v 0.11.1) (Anders et al, 2015) was then used for gene counting and DESeq2 (v 3.15) (Love et al, 2014) for differential gene expression analysis (Wald Test). The raw p values were adjusted by the Benjamini–Hochberg procedure to control the false discovery rate (FDR). Pathway enrichment analysis was performed with FGSEA (fast gene set enrichment analysis) (Sergushichev, 2016) using pre-ranked gene lists and pathways from the KEGG database (Kanehisa, 2019; Kanehisa et al, 2021).

For the inference of the upstream transcriptional regulators, we used VIPER (Virtual Inference of Protein-activity by Enriched Regulon analysis) (Alvarez et al, 2016). The estimation of their activation/inhibition status is based on the differential expression of their known target genes and is represented by a positive or negative NES (normalised enrichment score), respectively. The network that links the transcriptional regulators with their target genes was derived from DoRothEA (https://saezlab.github.io/dorothea/) (Garcia-Alonso et al, 2019).

## Measurement of extracellular leptin and adiponectin

3T3-L1 adipocytes were cultured as described above. Cells were transferred to low or high medium at the volumes stated above for

48 h, before media was collected and analysed by the Core Biochemical Assay Laboratory (Addenbrooke's Hospital, Cambridge). Leptin concentration was determined using the MesoScale Discovery Mouse Leptin Kit (K152BYC-2, Rockville, MD, USA), and adiponectin concentration using the MesoScale Discovery Mouse Adiponectin Kit (K152BXC, Rockville, MD, USA) according to manufacturer's instructions.

## Immunofluorescence analysis of GLUT4 translocation in 3T3-L1 adipocytes

1. Cells were differentiated in black CellCarrier-96 Ultra Microplates (PerkinElmer).
2. Cells were cultured in specified medium volumes for 48 h prior to assay and medium volumes were kept the same throughout the assay until cells were fixed.
3. Cells were serum-starved in DMEM containing 0.2% BSA for 2 h at 37 °C, 10% $CO_2$, and then stimulated with 0.5 nM or 100 nM insulin for 20 min.
4. Cells were then washed in an ice-cold PBS bath and instantly fixed with 4% paraformaldehyde on ice for 5 min, followed by 15 min at room temperature.
5. Cells were then quenched with 50 mM glycine for 10 min, then washed twice with PBS at room temperature, and incubated in blocking buffer (5% normal swine serum in PBS) for 20 min.
6. After blocking, cells were incubated with anti-GLUT4 (LM048) (2 μg/mL, Integral Molecular, PA, USA) and Lectin-FITC conjugate (Sigma #L4895) for 1 h at room temperature. This antibody recognises an external epitope on GLUT4, and so staining in non-permeabilised cells will only label GLUT4 present in the plasma membrane (Tucker et al, 2018).
7. Then, cells were washed in PBS and incubated with anti-human Alexa-647 (1:500, Thermo Fisher), and Hoechst 33342 (1:5000) room temperature.
8. After 1 h, the plates were washed three times in PBS, then stored and imaged in PBS, 5% glycerol and 2.5% DABCO.
9. Imaging was performed with the Opera Phenix (PerkinElmer) using a 20x NA1.0 water immersion objective.
10. Nine fields of view were imaged per well and analysed using a custom pipeline in Harmony High-Content Imaging and Analysis Software (PerkinElmer).
11. Cell plasma membrane regions were defined using the lectin-FITC signal and the anti-GLUT4 signal within this region calculated as a measure of GLUT4 translocation.

## CL dose-response and lipolysis assay

3T3-L1 adipocytes were cultured as described above. All lipolysis experiments were performed on day 10 following differentiation. Medium volume interventions were carried out 48 h prior to the experiment, and cells were kept in the appropriate medium volumes (high or low) for the duration of the lipolysis experiments.

1. Cells were serum-starved in DMEM containing 0.2% BSA for 2 h at 37 °C, 10% $CO_2$, and then stimulated with the appropriate agonist (insulin or CL316,243 (Tocris)) in KRP buffer for 30 min at 37 °C.

2. Glycerol release into the KRP buffer was determined using the free glycerol reagent (Sigma, #F6428) and absorbance was measured at 595 nm.
3. Glycerol release was normalised to cellular protein content (determined by BCA) of cells lysed in PBS containing 1% Triton X-100.

## Cell culture of pBAT (murine brown adipose tissue)

pBAT cells were kindly provided by Associate Professor Ángela María Martínez Valverde, IIBm Alberto Sols (CSIC-UAM) and differentiated as previously described (Miranda et al, 2010). Cells were tested for mycoplasma but were not authenticated; however, they were differentiated into adipocytes and only used if they reached >90–95% differentiation as observed by lipid accumulation.

1. Briefly, immortalised brown preadipocytes were grown in DMEM supplemented with 10% FBS, 20 nM insulin and 1 nM T3 (differentiation medium, DM) and grown until confluence.
2. Then, the cells were cultured for 2 days in induction medium consisting of DM supplemented with 0.5 μM dexamethasone, 0.125 μM indomethacin and 0.5 mM 3-isobutyl-1-methylxanthine.
3. After 2 days, the cells were cultured with DM again until they exhibited a fully differentiated phenotype with numerous multi-locular lipid droplets in their cytoplasm.

## Cell culture of L6 myotubes

1. Cells were cultured in MEM-α supplemented with 10% FBS at 37 °C, 10% $CO_2$.
2. For differentiation to myotubes, cells were moved to 2% FBS for 4 days, after which 2% FBS medium was replenished every 2 days.
3. Cells were used for 7–8 days after switching to a 2% FBS medium.

## Cell culture of iPSC-derived neurons

1. Human-derived iPSCs were cultured on Matrigel-coated dishes (Corning 354277) in complete Essential 8 Medium (Thermo Fisher Scientific A1517001).
2. Cells were routinely passaged when ~90% confluent using 0.5 mM EDTA in PBS, and culture media was renewed daily.
3. Accutase lifting was only performed when iPSCs needed to be singularised according to experimental requirements, and in this instance, seeded in the presence of 50 nM Chroman 1 (ROCK inhibitor, Tocris, 7163).
4. i3Neuron differentiation was carried out as described previously (Fernandopulle et al, 2018).
5. In summary, iPSCs were cultured for 3 days in Induction Medium (DMEM/F12 with HEPES Gibco, 11330032; 1% N2 supplement, Gibco, 17502048; 1% NEAA, Gibco, 11140050; 1% GlutaMAX, Gibco, 35050061; Doxycycline 2 μg/mL, Sigma, D9891) supplemented with 50 nM Chroman 1 for the first day only, with daily medium changes.

6. Day 3 neuro-precursor cells were reseeded in poly-L-ornithine-coated (Sigma, P3655) six-well plates and further differentiated in Cortical Medium (CM) (BrainPhys neuronal medium, STEMCELL Technologies, 05790; 2% B27 supplement 50x, Gibco, 17504044; BDNF (10 ng/mL), PeproTech, 450-02; NT-3 (10 ng/mL), PeproTech, 450-03; Laminin (1 mg/mL), Gibco, 23017015).

7. Each well contained $1.5 \times 10^6$ cells in a final volume of 2 ml of CM, and half-medium changes were carried out every third day.

8. In the 3 days preceding the medium-volume experiments, half-medium changes were carried out daily to acclimate the cells to a new glucose concentration.

9. Here, fresh CM was supplemented with 17.5 mM glucose.

10. Experiments with medium volumes were carried out on day 14.

11. Fresh CM containing 17.5 mM glucose was mixed 1:1 with conditioned CM (recovered from the neuronal culture during the half-medium changes and filtered sterile), to make up Neuronal Medium with a final concentration of 10 mM glucose.

12. On day 14, the total amount of medium was replaced by the experimental volumes of conditioned Neuronal Medium (Fig. EV1A), and the neurons were incubated for 24 h.

13. At the end of this period the medium was recovered for downstream extracellular glucose and lactate analysis, and the neuronal cells for qPCR.

## Cell culture of iPSC-derived hepatocytes

Maintenance of iPSCs:

1. Human induced pluripotent stem cells (iPSCs) were maintained on vitronectin XFTM (10 µg/mL, StemCell Technologies)-coated plates and in Essential 8 (E8) medium consisting of DMEM/F12 (Gibco), L-ascorbic acid 2-phosphate (1%), insulin-transferrin-selenium solution (2%, Life Technologies), sodium bicarbonate (0.7%), and Penicillin/Streptomycin (P/S) (1%), freshly supplemented with TGFβ (10 ng/mL, R&D) and FGF2 (12 ng/mL, Qkine).

2. Cells were split every 5–7 days by incubation with 0.5 µM EDTA-PBS (Thermo Fisher Scientific) for 4 min at room temperature, and clumps were dissociated into small clumps by pipetting.

Differentiation of FSPS13B hepatocytes:

1. For the FSPS13B hepatocytes, cells were differentiated towards hepatocytes as previously described (Hannan et al, 2013) with minor modifications.

2. iPSCs were split into single cells using Accutase (StemCell Technologies) and seeded at a density of 50,000 cells per cm² in E8 media with 10 µM ROCK Inhibitor Y-27632 (Selleckchem).

3. For foregut specification, cells were incubated for 5 days in RPMI media with 2% B27 and 50 ng/mL Activin (R&D).

4. Cells were differentiated into hepatocytes using HepatoZYME media (Thermo) with 2 mM L-glutamine, 1% P/S, 2% non-essential amino acids (Thermo), 2% chemically defined lipids (Thermo), 30 µg/mL transferrin (Roche), 14 µg/mL insulin (Roche), 20 ng/mL Oncostatin M (R&D) and 50 ng/mL hepatocyte growth factor (R&D).

5. Cells were analysed on day 32 of differentiation.

Differentiation of FOP hepatocytes:

1. For the FOP hepatocytes (Tomaz et al, 2022), iPSCs were dissociated into single cells following incubation with StemPro Accutase (Thermo Fisher) for 3–5 min at 37 °C and seeded at a density of 20,000 cells/cm² in E8 medium supplemented with 10 µM ROCK Inhibitor Y-27632 (Selleckchem).

2. E8 medium was replenished the following day.

3. 48 h after seeding, initial induction of the transgenes was achieved by incubation in E6 medium (E8 without growth factors) supplemented with 1 mg/mL doxycycline for 24 h at 5% $O_2$.

4. Cells were then maintained in HepatoZYME complete medium (HepatoZYME-SFM (Thermo Fisher), 2% chemically defined lipid concentrate (Thermo Fisher), 2% MEM non-essential amino acids solution (Thermo Fisher), 2 mM L-glutamine, 1% P/S, 14 µg/mL human insulin (Sigma), and 30 µg/mL human apo-transferrin (R&D)) supplemented with 1 mg/mL doxycycline, 50 ng/mL hepatocyte growth factor (HGF) (Preprotech), and 20 ng/mL oncostatin M (OSM) every day for the next 4 days, and every other day thereafter.

5. 9 days post-induction with doxycycline, the hepatocytes were fed with HepatoZYME complete medium containing low insulin (2 µg/mL), supplemented with 50 ng/mL HGF and 20 ng/mL OSM every other day until complete differentiation.

6. 14 days post-induction with doxycycline, the cells were moved back to 18% $O_2$.

7. Cells were analysed on day 20 of differentiation.

For medium-volume experiments, cells were differentiated towards hepatocytes in either 0.5 mL or 1 mL of complete HepatoZYME media. Cells were tested for mycoplasma and authenticated by genotyping, morphology and ability to differentiate into three germ lineages.

## Albumin secretion assay

The assay was an in-house microtitre plate-based three-step electrochemiluminescence immunoassay using the MesoScale Discovery assay platform (MSD, Rockville, USA).

1. A standard-bind MSD microtitre plate was coated overnight with a polyclonal goat anti-human albumin capture antibody (Bethyl Laboratories) diluted in PBS.

2. After coating, the plate was washed 3x with PBS/Tween using a Thermo Fisher automated plate washer.

3. Assay diluent was added to each well.

4. After 30 min incubation at room temperature on a plate shaker, 20 µL standards, QCs and undiluted sample were added to the plate in duplicate.

5. Two levels of QC were run at the beginning and end of each plate.

6. After 2 h incubation at room temperature on a plate shaker, the plate was washed 3x with PBS/Tween.

7. The polyclonal rabbit anti-human albumin detection antibody (Agilent Dako) diluted in MSD Diluent 100 was added to each well.

8. After 1 h incubation at room temperature on a plate shaker and washing with PBS/Tween, goat anti-rabbit IgG SULFO-TAG (MSD) labelled diluted in MSD Diluent 100 was added to the plate.

9. The plate was incubated on a plate shaker for a further 30 min.

10. After another wash step, the MSD read buffer was added to all the wells and the plate was immediately read on the MSD s600 plate reader.

11. Luminescence intensities for the standards were used to generate a standard curve using MSD's Workbench software package.

12. Results for study samples and QCs were read off this standard curve.

## Immunofluorescence analysis of albumin in iPSC-derived hepatocytes

1. Cells were fixed in 4% PFA for 15 min at room temperature, washed 3x in PBS and blocked for 1 h in PBS with 10% donkey serum and 0.1% Triton X-100 for permeabilisation.

2. Anti-albumin antibody (Bethyl Labs #A80-229A) was applied in blocking solution overnight at 4 °C.

3. Cells were then washed 3x in PBS and incubated in Alexa Fluor 488-conjugated secondary antibody (Life Technologies) in blocking solution, with 3 μM DAPI for 1 h at room temperature.

4. Cells were then washed 3x in PBS and visualised in the plate using a Zeiss inverted 710 confocal microscope.

## CYP3A4 activity assay

1. CYP3A4 enzymatic activity was measured using the P450-Glo kit (Promega).

2. Cells were incubated with 1:1000 luciferin-IPA in Hepatozyme complete for 1 h at 37 °C.

3. 50 μL of cell culture supernatant was mixed with 50 μL detection reagent and incubated for 20 min at room temperature in Greiner white 96-well microplates (Sigma-Aldrich).

4. Luminescence was measured in triplicate on a GloMax plate reader. Hepatozyme complete medium was used as background control.

## Bone marrow-derived macrophage (BMDM) cell culture

BMDM isolation from male C57BL/6 J mice and in vitro culture was performed as previously described (Bidault et al, 2021). Animal work was carried out in the Disease Model Core, and was regulated under the Animals (Scientific Procedures) Act 1986 Amendment Regulations 2012 following ethical review by the University of Cambridge Animal Welfare and Ethical Review Body according to UK Home Office guidelines.

## Fatty acid oxidation (FAO) assay

FAO measurements were conducted as previously described (Bidault et al, 2021).

1. BMDMs were seeded in 24-well plates at a density of 250,000 cells per well.

2. At 24 h after IL-4 stimulation, cells were washed with PBS and incubated in the presence of 0.5 mL FAO medium (RPMI containing 12.5 mM HEPES, 1 mM L-carnitine, 10% heat-inactivated (HI)-FBS and 0.2 μCi of [1-$^{14}$C]oleate (NEC317250UC, PerkinElmer)).

3. Plates were then immediately sealed with parafilm M and incubated at 37 °C for 3 h.

4. Meanwhile, $CO_2$ traps were prepared by adding 200 μL of concentrated HCl into the 1.5 mL Eppendorf tubes containing paper discs, wetted with 20 μL of 1 M NaOH, in the inner side of their lids.

5. Once the FAO reaction was complete, 400 μL of medium from BMDMs was transferred into the $CO_2$ traps, lids were immediately closed, and tubes were incubated for 1 h at room temperature, allowing $CO_2$ to escape the medium and react with NaOH in the paper disc.

6. Paper discs were then transferred to scintillation vials containing 5 ml of Hionic-Fluor scintillation liquid, and radioactivity was measured using a liquid scintillation counter.

## hPSC-derived cardiac organoids

Ethical approval for the use of human embryonic stem cells (hPSCs) was obtained from QIMR Berghofer's Ethics Committee (P2385), and was carried out in accordance with the National Health and Medical Research Council of Australia (NHMRC) regulations. Female HES3 (WiCell) were maintained in mTeSR PLUS (Stem Cell Technologies)/ Matrigel (Millipore) and passaged using T ReLeSR (Stem Cell Technologies). Quality control was performed with karyotyping (G-banding) and mycoplasma testing. hPSC-derived cardiac organoids were generated as recently described (Mills et al, 2021).

Cardiomyocyte and stromal cell differentiation was achieved using previously described protocols (Mills et al, 2017, 2019; Voges et al, 2017; Hudson et al, 2012).

1. hPSCs were seeded on Matrigel-coated flasks at $2 \times 10^4$ cells/cm$^2$ and cultured in mTeSR-1 for 4 days.

2. To induce cardiac mesoderm, hPSCs were cultured in RPMI B27-medium (RPMI 1640 GlutaMAX+ 2% B27 supplement without insulin, 200 μM L-ascorbic acid 2-phosphate sesquimagnesium salt hydrate (Sigma) and 1% penicillin/streptomycin (Thermo Fisher Scientific), supplemented with 5 ng/ml BMP-4 (RnD Systems), 9 ng/ml Activin A (RnD Systems), 5 ng/ml FGF2 (RnD Systems) and 1 μM CHIR99021 (Stemgent or Stem Cell Technologies).

3. Mesoderm induction required daily medium exchanges for 3 days.

4. This was followed by cardiac specification using RPMI B27-containing 5 μM IWP-4 (Stem Cell Technologies) for another 3 days, and then further 7 days using 5 μM IWP-4 RPMI B27+ (RPMI 1640 Glutamax + 2% B27 supplement with insulin, 200 μM L-ascorbic acid 2-phosphate sesquimagnesium salt hydrate and 1% penicillin/streptomycin) with medium change every 2–3 days.

5. For the final 2 days of differentiation, hPSCs were cultured in RPMI B27+.

6. Harvest of differentiated cardiac cells involved enzymatic

digestion, firstly in 0.2% collagenase type I (Sigma) in 20% foetal bovine serum (FBS, Thermo Fisher Scientific) in PBS (with $Ca^{2+}$ and $Mg^{2+}$) at 37 °C for 1 h, and secondly in 0.25% trypsin-EDTA at 37 °C for 10 min.

7. The trypsin was neutralised, and then cells were filtered through a 100-µm mesh cell strainer (BD Biosciences), centrifuged at $300 \times g$ for 3 min, and resuspended in α-MEM Glutamax, 10% FBS, 200 µM L-ascorbic acid 2-phosphate sesquimagnesium salt hydrate and 1% penicillin/streptomycin (MEM + +++).

8. Previous flow cytometry analysis indicated that differentiated cardiac cells were ~70% α-actinin$^+$/CTNT$^+$ cardiomyocytes, ~30% CD90 stromal cells (Voges et al, 2017).

9. Human cardiac organoid (hCO) culture inserts were fabricated using SU-8 photolithography and PDMS moulding (Mills et al, 2017).

10. Acid-solubilized bovine collagen 1 (Devro) was salt balanced using 10x DMEM (Thermo Fisher Scientific) and pH neutralised using 0.1 M NaOH before combining with Matrigel and then the cell suspension on ice.

11. Each hCO contained $5 \times 10^4$ cells, a final concentration of 2.6 mg/ml collagen I and 9% Matrigel.

12. 3.5 µL of suspension was pipetted into the hCO culture insert and incubated at 37 °C with 5% $CO_2$ for 45 min in order to gel.

13. After gelling, α-MEM GlutaMAX (Thermo Fisher Scientific), 10% foetal bovine serum (FBS), 200 µM L-ascorbic acid 2-phosphate sesquimagnesium salt hydrate (Sigma) and 1% Penicillin/Streptomycin (Thermo Fisher Scientific) was added.

14. hCO were subsequently cultured in maturation medium (MM) (Mills et al, 2017) with medium changes every 2 to 3 days for 5 days (7 days old hCO).

15. To better approximate adult metabolic provisions a 'weaning medium' (WM) was utilised.

16. hCO were cultured in WM containing 4% B27 – insulin, 5.5 mM glucose, 1 nM insulin, 200 µM L-ascorbic acid 2-phosphate sesquimagnesium salt hydrate, 1% P/S, 1% GlutaMAX (100x), 33 µg/mL aprotinin and 10 µM palmitate (conjugated to bovine serum albumin in B27) in DMEM without glucose, glutamine and phenol red (Thermo Fisher Scientific) with medium changes every 2–3 days.

The elasticity of the Heart-Dyno poles enables the contractile properties to be determined via tracking pole deflection, which directly correlates with force (Mills et al, 2017). Videos of 10 s were made of each hCO with the Nikon ANDOR WD Revolution Spinning Disk microscope (magnification 4x). While imaging, hCO were incubated at 37 °C, 5% $CO_2$ to prevent changes in contractile behaviour. This facilitated the analysis of the contractile properties of the organoids and the production of time-force graphs (Mills et al, 2017). Moreover, data was obtained regarding additional important functional parameters, including the contraction rate and the activation and relaxation time of the organoids.

## Statistical analysis

All statistical analyses, unless otherwise stated in figure legends, were carried out using GraphPad Prism 9 or Excel. Data were presented as mean ± SEM, of at least three independent biological replicates as stated in the figure legends. Processing of GC-MS samples was randomised, and immunofluorescence analysis of

GLUT4 translocation in 3T3-L1 adipocytes was automated on the Opera Phenix to ensure unbiased results. Investigators were not blinded to samples. All samples were included in the analysis. Two-tailed paired/unpaired Student's t-tests were used to compare the means between the two groups. One/two-way ANOVA with Šidák correction for multiple comparisons was performed for multigroup (at least three) comparisons. Variations among replicates are expected to have normal distributions and equal variances.

## Graphics

The graphical synopsis, Figs. 1D and EV3C were created with BioRender.com.

## Data and materials availability

The datasets and computer code produced in this study are available in the following databases: RNAseq data: Array Express (www.ebi.ac.uk/arrayexpress), with accession numbers E-MTAB-12298 and E-MTAB-12299. Metabolomics data: EMBL-EBI MetaboLights database (https://doi.org/10.1093/nar/gkz1019, PMID:31691833) with the identifier MTBLS6677. There are no restrictions on data availability.

## Peer review information

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

## Acknowledgements

These studies were supported by the Wellcome-MRC Institute of Metabolic Science (IMS) Metabolic Research Laboratories, Imaging Core (Wellcome Trust Major Award (208363/Z/17/Z)), and the MRC MDU Mouse Biochemistry Laboratory (MC_UU_00014/5). RNAseq was performed by the IMS Genomics and transcriptomics core facility and supported by the UK MRC Metabolic Disease Unit (MRC_MC_UU_00014/5) and a Wellcome Trust Major Award (208363/Z/17/Z). S.V. was supported by BHF (RG/18/7/33636). O.J.C. was supported by a Wellcome Trust PhD studentship. I.K. was supported by a Medical Research Council (MRC) PhD studentship. C.P. was supported by a BBSRC project grant (BB/W005905/1). A.J.M was supported by BBSRC [BB/F016581/1] and BHF [FS/17/61/33473]. D.C.G. is funded by a Sir Henry Dale Fellowship from the Wellcome Trust/Royal Society (210481). J.A.N was supported by a Wellcome Senior Clinical Research Fellowship (215477/Z/19/Z). J.E.H. was supported by a Snow Medical Fellowship. The L.V. lab is funded by the ERC advanced grant New-Chol and the core support grant from the Wellcome Trust and Medical Research Council (MRC) of the Wellcome–Medical Research Council Cambridge Stem Cell Institute. For K.H.F.-W., the work was supported in part by DOD-W81XWH-19-1-0213. C.F. and M.Y. were supported by the MRC Core award (MRC_MC_UU_12022/6). A.V-P. was supported by BHF (RG/18/7/33636) and MRC (MC_UU_12012/2). D.J.F. was supported by a Medical Research Council Career Development Award (MR/S007091/1) and a Wellcome Institution Strategic Support Fund award (204845/Z/16/Z).

## Author contributions

**Joycelyn Tan**: Formal analysis; Investigation; Visualisation; Methodology; Writing—original draft; Project administration; Writing—review and editing. **Sam Virtue**: Conceptualisation; Formal analysis; Supervision; Investigation; Visualisation; Methodology; Writing—original draft; Project administration; Writing—review and editing. **Dougall M Norris**: Conceptualisation; Investigation; Methodology. **Olivia J Conway**: Investigation. **Ming Yang**: Formal analysis; Investigation; Visualisation. **Guillaume Bidault**: Investigation. **Christopher Gribben**: Investigation. **Fatima Lugtu**: Investigation. **Ioannis Kamzolas**: Formal analysis; Investigation. **James R Krycer**: Investigation. **Richard J Mills**: Investigation. **Lu Liang**: Investigation. **Conceição Pereira**: Investigation. **Martin Dale**: Investigation. **Amber S Shun-Shion**: Methodology. **Harry JM Baird**: Investigation. **James A Horscroft**: Investigation. **Alice P Sowton**: Investigation. **Marcella Ma**: Investigation. **Stefania Carobbio**: Investigation. **Evangelia Petsalaki**: Supervision; Investigation. **Andrew J Murray**: Supervision; Investigation. **David C Gershlick**: Supervision; Investigation. **James A Nathan**: Supervision; Investigation. **James E Hudson**: Supervision; Investigation; Visualisation. **Ludovic Vallier**: Supervision. **Kelsey H Fisher-Wellman**: Investigation; Visualisation. **Christian Frezza**: Supervision. **Antonio Vidal-Puig**: Supervision; Writing—review and editing. **Daniel J Fazakerley**: Conceptualisation; Supervision; Investigation; Methodology; Writing—original draft; Project administration; Writing—review and editing.

## Disclosure and competing interests statement

The authors declare no competing interests.

# Expanded View Figures

**Figure EV1.  Changes in glucose metabolism in low medium is due to increased oxygen availability.**

(**A**) Table of medium volumes used in this study and the corresponding medium heights of both top and bottom menisci. Images of each plate-type containing different medium volumes were used to measure menisci heights. Known well diameters were used to convert menisci heights from pixels to mm. 'High' refers to the standard culture volumes used. (**B**) Representative trace of fluorescence intensity indicative of pericellular oxygen concentrations under different medium volumes, measured for 24 h in 96-well plates. The bar graph shows relative oxygen levels taken at 16 h (arrow in representative trace) ($n = 4$ biological replicates). AA antimycin A. (**C**) Western blot of mitochondrial respiratory complexes I–V after 16 h of medium volume change in 12-well plates ($n = 3$ biological replicates). (**D**) Oxygen consumption rate (OCR) of permeabilised 3T3-L1 adipocytes upon different substrate stimulation after 16 h of medium volume change ($n = 5$ biological replicates). (**E**) Extracellular medium glucose and lactate measurements in 24-well or Lumox plates after 16 h culture in high or low medium volumes ($n = 3$ biological replicates). (**F**) Medium glucose concentration after 16 h of medium volume change in 12-well plates ($n = 6$ biological replicates). (**G**) Extracellular medium glucose and lactate measurements after 16 h medium volume change with different starting glucose concentrations in 12-well plates. ($n = 4$ biological replicates). (**H**) OCR measurements from different cell densities in 96-well plates ($n = 3$ biological replicates). (**I**) Extracellular medium glucose and lactate measurements after 16 h medium volume change with different cell densities in 12-well plates measured by DNA concentration. ($n = 3$ biological replicates). Data information: Data were represented as mean ± SEM (**B, D–I**). ****$p < 0.0001$ by two-way ANOVA with Šidák correction for multiple comparisons (**E**).

A

| Plate | 12-well | | | 24-well | | | 96-well | | |
|---|---|---|---|---|---|---|---|---|---|
| Well area (cm²) | 3.8 | | | 1.9 | | | 0.32 | | |
| | Medium volume (µL) | Top men. (mm) | Bottom men. (mm) | Medium volume (µL) | Top men. (mm) | Bottom men. (mm) | Medium volume (µL) | Top men. (mm) | Bottom men. (mm) |
| High (standard) | 1000 | 4.7 | 2.4 | 500 | 3.9 | 2.2 | 100 | 3.3 | 2.5 |
| Mid | 666 | 3.8 | 1.7 | 333 | 3.5 | 1.6 | 50 | 2.2 | 1.5 |
| Low | 333 | 3.0 | 0.8 | 167 | 1.9 | 0.8 | 33 | 2.0 | 0.7 |

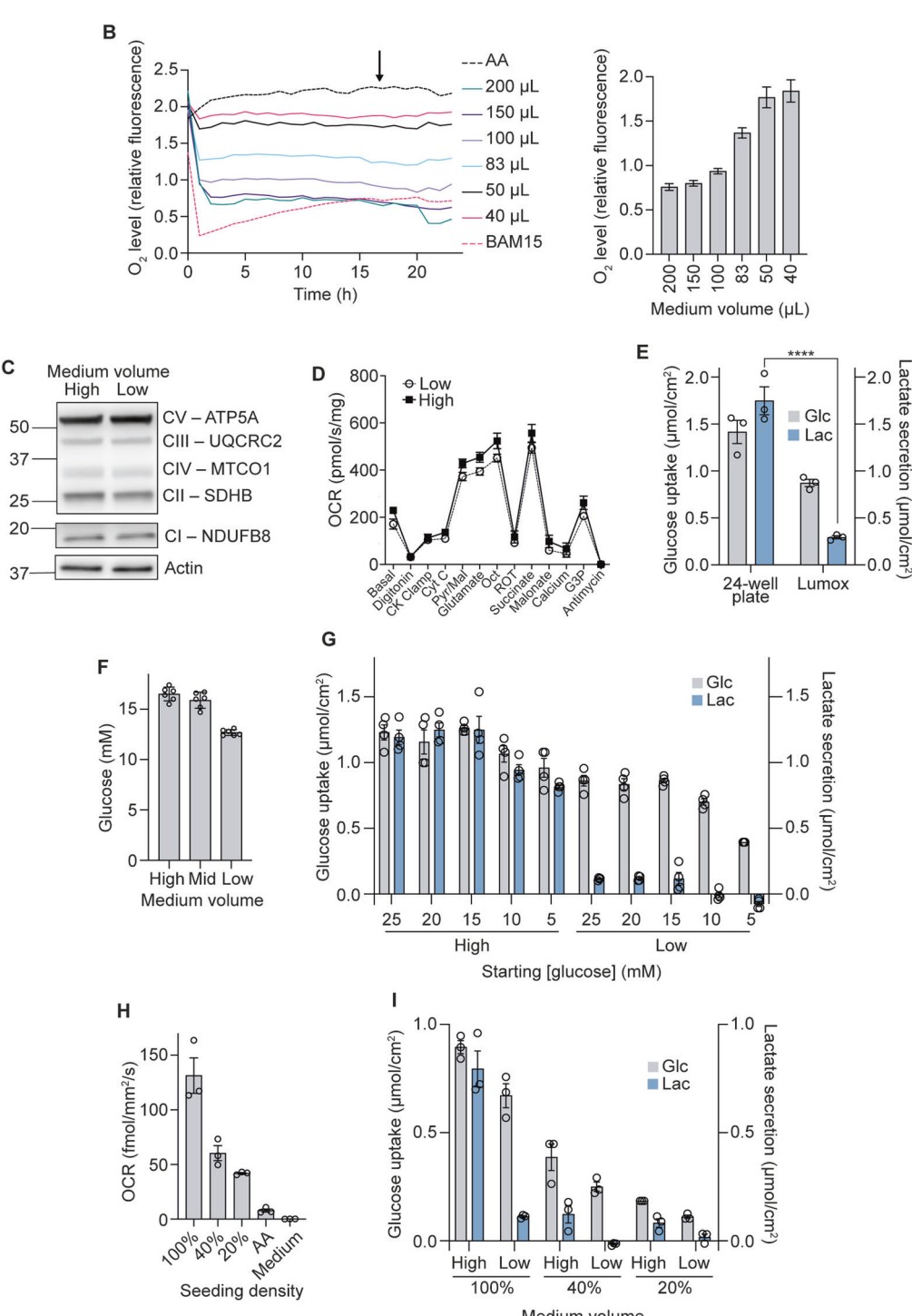

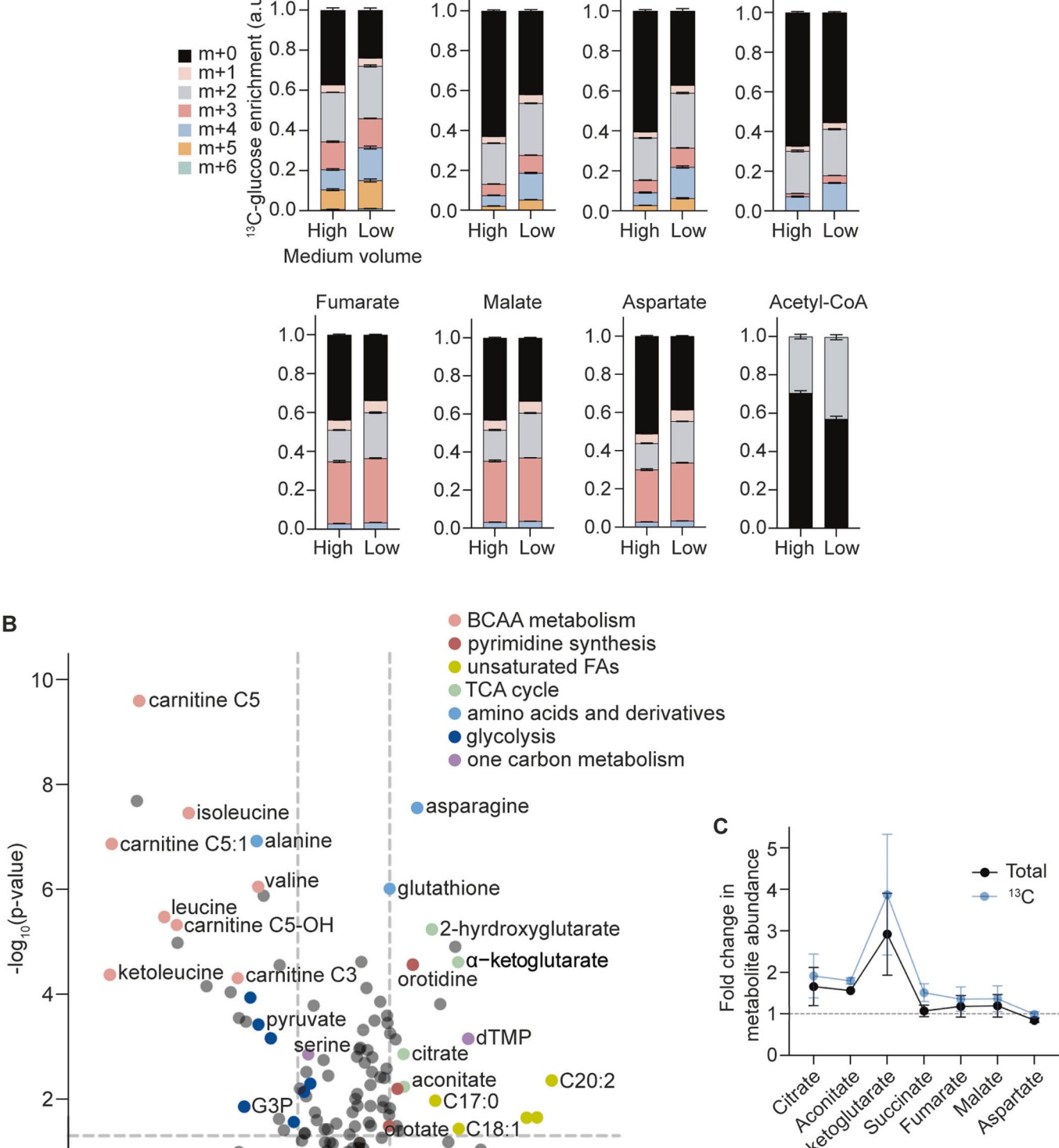

Figure EV2. Lowering medium volumes rewires cellular metabolism.

(A) Fractional abundance of each isotopologue after 4 h medium volume change ($n = 6$ biological replicates). (B) Volcano plot of differentially regulated metabolites after 16 h medium volume change. Metabolites of interest which are significantly changed ($p < 0.05$) are highlighted according to their metabolic pathways ($n = 6$ biological replicates). BCAA branched-chain amino acid, FA fatty acid, G3P glyceraldehyde−3−phosphate. (C) Fold change of total and U$^{13}$C-glucose labelled TCA metabolite abundance after 16 h medium volume change in 12-well plates ($n = 4$ biological replicates). Data information: Data were represented as mean ± SEM (A, C). $p$ value threshold of 0.05 (B) was determined using differential metabolite analysis (DMA) with Student's $t$-test.

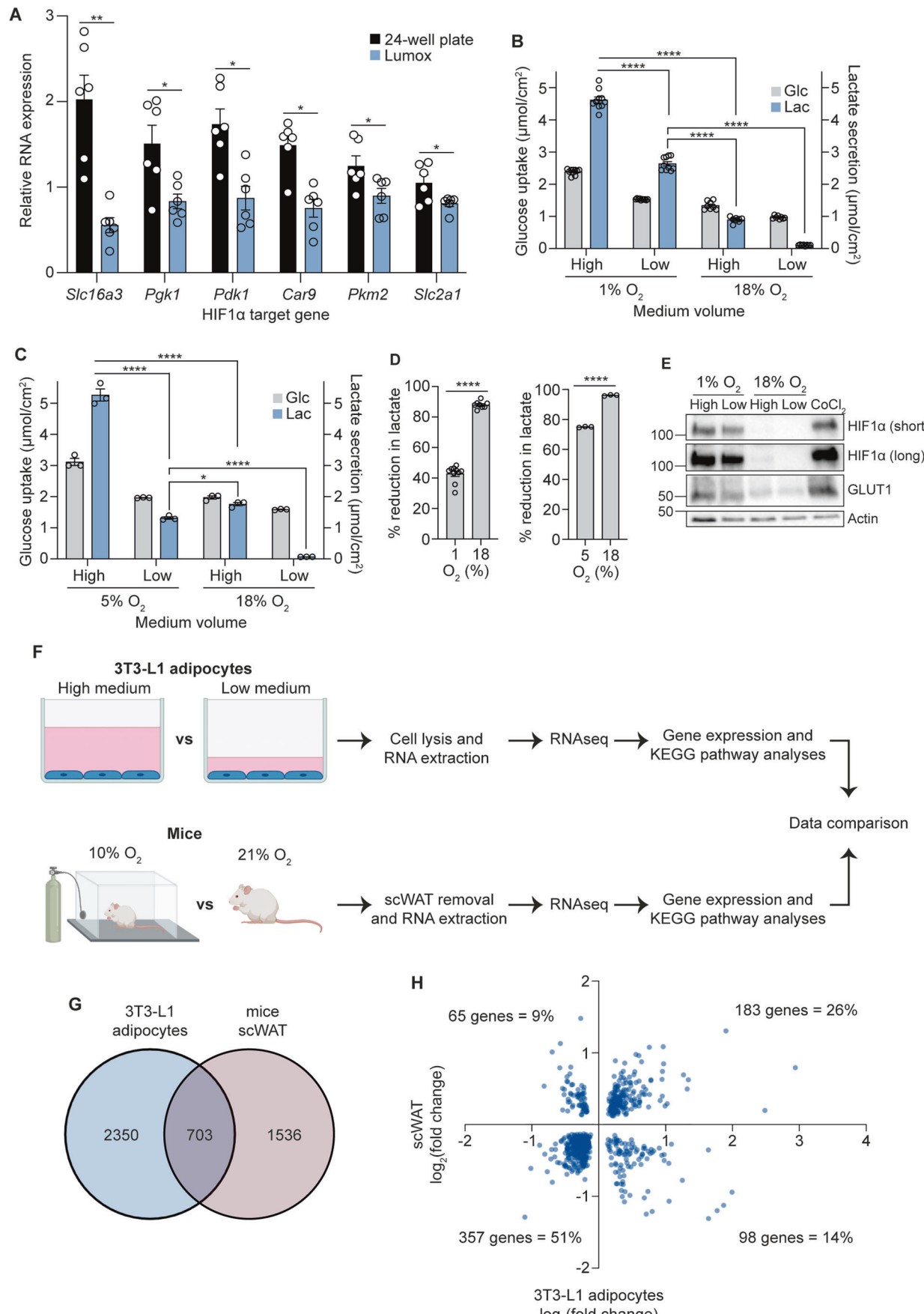

◀

**Figure EV3. Reducing oxygen availability causes metabolic and transcriptional rewiring.**

(A) Relative RNA expression of HIF1α target genes in 3T3-L1 adipocytes cultured in either 24-well or gas-permeable Lumox plates ($n = 6$ biological replicates). (B) Extracellular medium glucose and lactate measurements after 16 h medium volume change (12-well plate) in 1 or 18% oxygen incubators ($n = 8$–10 biological replicates). (C) Extracellular medium glucose and lactate measurements after 16 h medium volume change (12-well plate) in 5 or 18% oxygen incubators ($n = 3$ biological replicates). (D) Percentage reduction in lactate production calculated from Fig. EV3B (1% $O_2$) ($n = 8$–10 biological replicates) and EV3C (5% $O_2$) ($n = 3$ biological replicates). (E) Western blot of HIF1α (after both short and long imaging exposures) and GLUT1 after 16 h medium volume change at 1 or 18% $O_2$ in 12-well plates ($n = 3$ biological replicates). (F) Schematic representation of the RNAseq experimental workflow. RNA extracted from 3T3-L1 adipocytes (cultured in high or low medium for 16 h in 12-well plates) or scWAT (obtained from mice kept in 10 or 21% $O_2$ for 4 weeks) were sequenced. The two sets of analysed data were then compared. (G) Venn diagram showing the overlapping differentially expressed genes ($p$-adj < 0.05) from both 3T3-L1 adipocytes (high $vs$ low medium) ($n = 6$ biological replicates) and mice scWAT (10 vs 21% $O_2$) ($n = 10$ biological replicates). (H) Fold change of the 703 differentially expressed genes in 3T3-L1 adipocytes (y-axis) and mice scWAT (x-axis) from the intersection in Fig. EV3D, showing a 77% directional concordance. Data information: Data were represented as mean ± SEM (A–D). *$p < 0.05$, **$p < 0.01$, ****$p < 0.0001$ by two-way ANOVA with Šidák correction for multiple comparisons (B, C), or by paired/unpaired Student's $t$-tests (A, D).

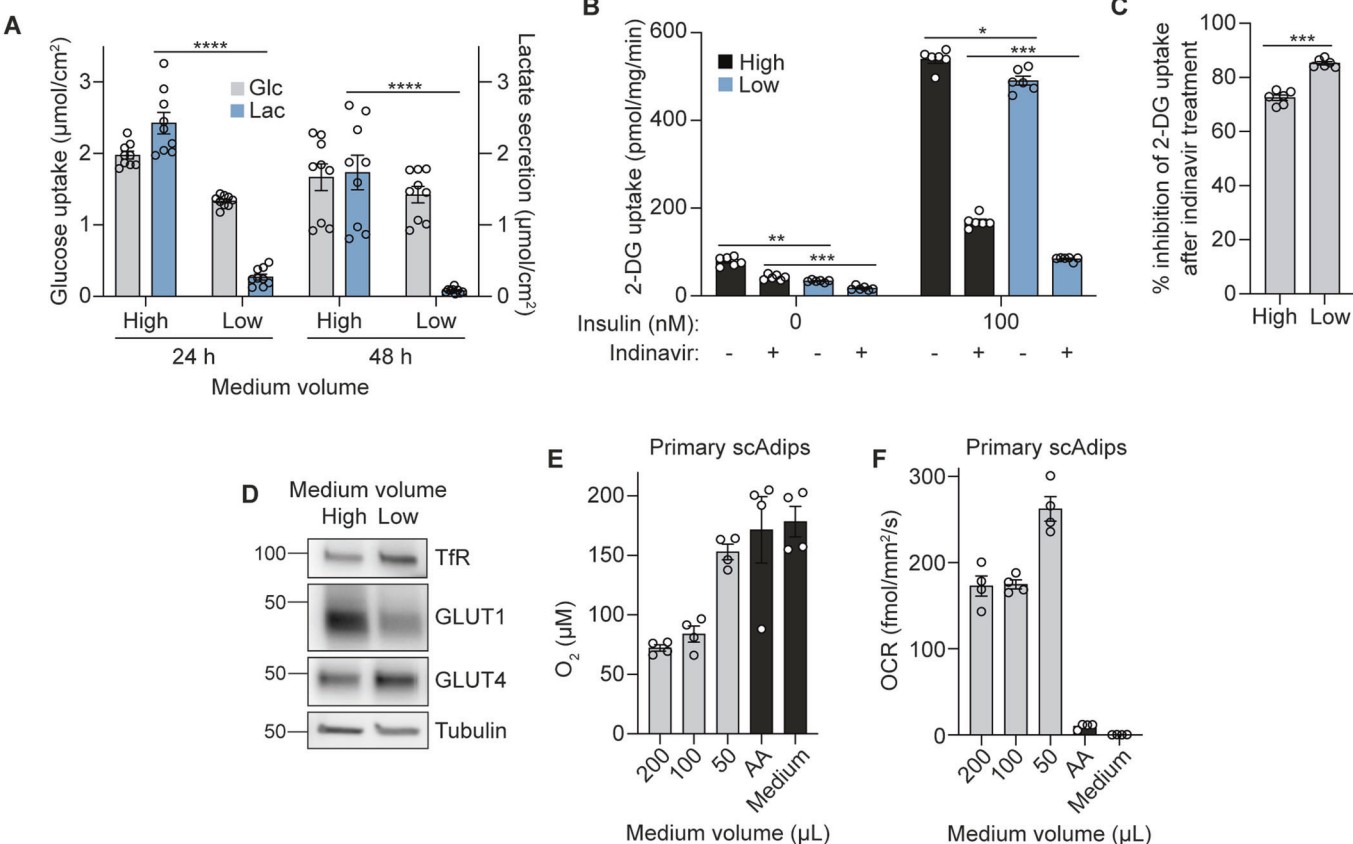

**Figure EV4. Effects of lowering medium volumes on 3T3-L1 adipocyte glucose metabolism and oxygen use in primary scAdips.**

(A) Extracellular measurements of 3T3-L1 medium glucose and lactate 24 or 48 h after medium volume change in 12-well plates ($n = 3$ technical replicates from $n = 3$ biological replicates). (B) 2-deoxyglucose (DG) uptake after insulin stimulation and 200 μM indinavir (GLUT4 inhibitor) treatment. Cells were cultured in high or low medium for 48 h in 24-well plates prior to the experiment ($n = 6$ biological replicates). (C) Percentage inhibition of 2-DG uptake after indinavir treatment, calculated from the difference between $+/-$ indinavir treated conditions, as a percentage of -indinavir 2-DG uptake upon 100 nM insulin stimulation. Graph shows the percentage of 2-DG uptake that is GLUT4-dependent (i.e. inhibited by indinavir) ($n = 6$ biological replicates). (D) Western blot of GLUT1 and GLUT4 in 3T3-L1s after 48 h medium volume change (12-well plate) ($n = 6$ biological replicates). (E) The pericellular oxygen concentration of primary scAdips cultured with different medium volumes in 96-well plates ($n = 4$ biological replicates). AA antimycin A. (F) OCR of primary scAdips cultured with different medium volumes in 96-well plates (n = 4 biological replicates). Data information: Data were represented as mean ± SEM (A–D). *$p < 0.05$, **$p < 0.01$, ***$p < 0.001$, ****$p < 0.0001$ by two-way ANOVA with Šidák correction for multiple comparisons (A. B), or by paired two-tailed Student's $t$-test (C).

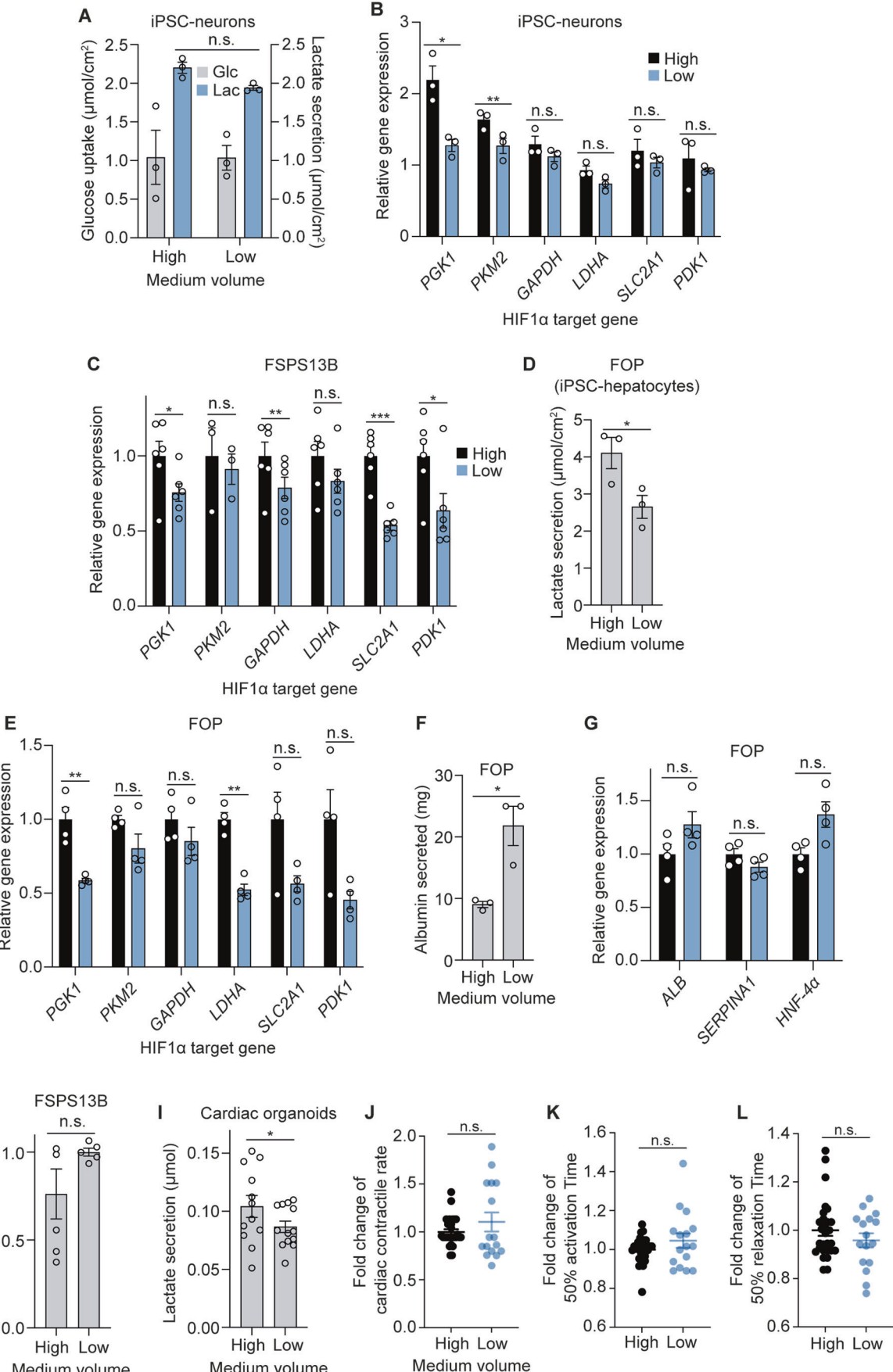

◄  **Figure EV5.  Effects of low medium volumes on other primary and hPSC-derived cell types.**

(A) Extracellular medium glucose and lactate measurements from iPSC-derived neurones after 16 h medium volume change in 6-well plates ($n = 3$ biological replicates). (B) Relative RNA expression of HIF1α target genes in iPSC-derived neurones after 16 h medium volume change in 12-well plates ($n = 3$ biological replicates). (C) Relative RNA expression of HIF1α target genes in FSPS13B hepatocytes cultured under different medium volumes in 12-well plates throughout differentiation ($n = 3–6$ biological replicates). (D) Lactate secretion in FOP hepatocytes after medium volume change in 12-well plates throughout differentiation ($n = 3$ biological replicates). (E) Relative RNA expression of HIF1α target genes in FOP hepatocytes cultured under different medium volumes in 12-well plates throughout differentiation ($n = 4$ biological replicates). (F) Albumin secretion over 24 h by FOP hepatocytes after medium volume change in 12-well plates throughout differentiation ($n = 3$ biological replicates). (G) Relative RNA expression of hepatocyte differentiation marker genes in FOP hepatocytes after medium volume change in 12-well plates throughout differentiation ($n = 4$ biological replicates). (H) Relative CYP3A4 activity in FSPS13B hepatocytes after medium volume change throughout differentiation ($n = 5$ biological replicates). (I) Lactate secretion by cardiac organoids after 48 h of medium volume change ($n = 3–6$ technical replicates from $n = 3$ biological replicates). High $= 150$ μL, Low $= 50$ μL. (J) Contractile rate (normalised) ($n = 2–17$ technical replicates from $n = 4$ biological replicates). (K) Time from 50% activation to peak (normalised) ($n = 2–17$ technical replicates from $n = 4$ biological replicates). (L) Time from peak to 50% relaxation (normalised) ($n = 2–17$ technical replicates from $n = 4$ biological replicates). Data information: All data were represented as mean ± SEM (A–L). n.s. non-significant; $*p < 0.05$, $**p < 0.01$, $***p < 0.001$ by two-way ANOVA (A) or by paired/unpaired two-tailed Student's $t$-test (B–L).

