## [Peer Review File · The EMBO Journal]

Limited oxygen in standard cell culture alters metabolism and function of differentiated cells

Joycelyn Tan, Sam Virtue, Dougall Norris, Olivia Conway, Ming Yang, Guillaume Bidault, Christopher Gribben, Fatima Lugtu, Ioannis Kamzolas, James Krycer, Richard Mills, Lu Liang, Conceição Pereira, Martin Dale, Amber Shun-Shion, Harry Baird, James Horscroft, Alice Sowton, Marcella Ma, Stefania Carobbio, Evangelia Petsalaki, Andrew Murray, David Gershlick, James Nathan, James Hudson, Ludovic Vallier, Kelsey Fisher-Wellman, Christian Frezza, Antonio Vidal-Puig, and Daniel Fazakerley

Corresponding author(s): Daniel Fazakerley (djf72@medschl.cam.ac.uk) , Antonio Vidal-Puig (ajv22@medschl.cam.ac.uk), Sam Virtue (sv234@cantab.ac.uk)

Review Timeline:

Submission Date:	12th Jan 24
Editorial Decision:	6th Feb 24
Revision Received:	20th Feb 24
Accepted:	3rd Mar 24

Editor: Daniel Klimmeck

Transaction Report:

Please note that the manuscript was transferred from another journal where it was originally reviewed. Since the original reviews are not subject to EMBO's transparent review process policy, they cannot be published.

Dear Dr Fazakerley,

Thank you again for the submission of your amended manuscript (EMBOJ-2024-116647) to The EMBO Journal. We have carefully assessed your manuscript and the point-by-point response provided to the referee concerns that were raised during review at a different journal. In addition, and as mentioned before, we decided to involve two arbitrating experts to evaluate the revised version of your work, with respect to technical robustness, conceptual advance and overall suitability of your work for publication in The EMBO Journal.

As you will see from their comments enclosed below, the advisors are in favour of the work stating the interest and value of your results and they are supportive of publication at The EMBO Journal, pending minor revision.

We are thus pleased to inform you that we can offer to swiftly move forward towards acceptance of this work at The EMBO Journal, pending minor revision of the following remaining issues, which need to be adjusted in a re-submitted version.

- We are applying a Structured Methods format for the Materials and Methods of articles published at EMBO Press, which have a string focus on methods advances. Adhering to this format is optional for research articles. However, considering the strong methodological aspect of your study, we would strongly encourage you to use it. Specifically, the Material and Methods section should include a Reagents and Tools Table (listing key reagents, experimental models, software and relevant equipment and including their sources and relevant identifiers) followed by a Methods and Protocols section in which we encourage the authors to describe their methods using a step-by-step protocol format with bullet points. More information on how to adhere to this format as well as downloadable templates (.doc or .xls) for the Reagents and Tools Table can be found in the author guidelines of our sister journal Molecular Systems Biology <https://www.embopress.org/page/journal/17444292/authorguide#methodguide>. An example of a paper with Structured Methods can be found here: <https://www.embopress.org/doi/full/10.15252/emj.2018100300>. We encourage you to be even more explicit in adding details on the experimental procedures, as this should be valuable in ensuring reproducible application if the approach.

- Revisit annotation of the Western blot analysis of HIF-1alpha (Fig3B,C; adv#2).
- Amend literature references and discussion of the context along the comments made by the advisors.

Based on the positive views of the advisors together with our own assessment, we decided to proceed with publication of your work at The EMBO Journal pending the above points in a time frame of two weeks.

Once we have received the revised version, we should then be able to swiftly proceed with formal acceptance and expedited production of the manuscript.

We also need you to take care of a number of minor issues related to formatting and data annotation as detailed below, which should be addressed at re-submission.

Further, I will share additional changes and comments from our production team and my colleague H. Sonntag (CC'ed) for Source Data requests shortly.

Please submit a revised version of the manuscript using the link enclosed below, addressing the advisor's comments.

As you might have seen on our web page, every paper at the EMBO Journal now includes a 'Synopsis', displayed on the html and freely accessible to all readers. The synopsis includes a 'model' figure as well as 2-5 one-short-sentence bullet points that summarize the article. I would appreciate if you could provide this figure and the bullet points.

Thank you again for giving us the chance to consider your manuscript for The EMBO Journal, I look forward to hearing from you and receiving your final revised version of the manuscript.

Kind regards,

Daniel Klimmeck

Daniel Klimmeck PhD
Senior Editor
The EMBO Journal.

EMBOJ-2024-116647, arbitrating advisor #1's comments:

Overall, I think this manuscript adds to the literature regarding in vitro culture conditions, which are mis-conceived to be hyperoxic. Indeed, previous studies have documented this nuance; however, the authors specifically emphasize this point in their manuscript to highlight the importance of hypoxia in standard culture conditions.

I am pointing out a study (PMID: 25114222) using a hypoxia-responsive element reporter, which indicates some hypoxia (ie, detectable by an HRE-GFP reporter) exists under 21% oxygen culture condition. Another paper (, see Figure 1D, x-axis), which the authors could consider adding to the discussion to further emphasize their point and enrich the literature with supportive evidence for the concept the authors wish to underscore.

It is also notable that the converse, that is respiring cells exist under hypoxia - which was recently documented in another collaborative study published (PMID: 36462504) using the same reporter system.

Overall, the authors' emphasis on the nuance of 'hypoxia' in vitro is important and should be noted in the literature to diminish misconceptions of in vitro culture conditions.

Overall, I think the manuscript should be published and it is up to the editor and the authors whether additional discussions (relating to the items I discuss above) are necessary to be included in the manuscript.

EMBOJ-2024-116647, arbitrating advisor #2's comments:

While there is a substantial amount of work presented in this manuscript, it aligns closely with what we've been practicing in the lab for years. Back in 2008, we published an article in the Journal of Cellular Physiology titled "Activation of HIF-1 α in exponentially growing cells via hypoxic stimulation is independent of the Akt/mTOR pathway." Our research demonstrated that despite cells being incubated in a 21% environment, the actual oxygen levels can range from 2 to 3% after 24 to 48 hours, depending on cell confluency.

Since then, I've consistently emphasized to my students that the real challenge lies not in hypoxia, but in maintaining "normoxia." We're meticulous about cell culture conditions, ensuring appropriate medium volumes and cell densities. We never culture cells densely from Friday to Monday, opting instead for multiple plates with low-density cultures. I've rejected numerous articles that fail to adhere to these standards, especially those claiming "normoxic" conditions after a week of high-density culture.

To put it simply, the findings presented by Tan et al. don't come as a surprise to me. It is not ground-breaking. Nevertheless, they have achieved a more comprehensive level of explanation compared to our previous work. Moreover, publishing in the EMBO Journal may help raise awareness about the importance of understanding cell culture practices. My only recommendation to Tan et al. is to include HIF-1 α Western blot data. It is feasible.

Nathalie Mazure'

>> Author Contributions: Please remove the author contributions information from the manuscript text. Note that CRediT has replaced the traditional author contributions section as of now because it offers a systematic machine-readable author contributions format that allows for more effective research assessment. and use the free text boxes beneath each contributing author's name to add specific details on the author's contribution.

>> Author Contributions: Please confirm corresponding sco-authorship (S.V., A.V-P.).

>> >> Introduce ORCID IDs for all corresponding authors (S.V., A.V-P.) via our online manuscript system. Please see below for

additional information.

More information is available in our guide to authors.
<https://www.embopress.org/page/journal/14602075/authorguide>

>> Please add a completed Author Checklist to your manuscript.

>> Figures: Main figures and supplementary figures should be removed from the manuscript text and uploaded as individual, high-resolution figure files. The supplementary figures should be renamed "Figure EV1" to EV5. The legends should remain in the manuscript text, legends of the suppl. figures should be after the main figure legends under the heading "Expanded View Figure Legends".

>> Funding: The "Rights and Retention Statement" should be merged with the funding info in the Acknowledgements; the sentence about the cc licence is not needed.

>> Dataset EV Legends: Tables S1-S6 should be renamed Dataset EV1-6 and their callouts need to be updated; the list with their short descriptions should be removed from the manuscript text as these should only be in the excel files themselves.

>> Revisit publication status of the bioRxiv references Rogers et al (2023), Sergushichev et al (2016) and update in case of formal journal publication.

>> Data availability section: please privacy from your data sets; move section to the end of Materials & Methods.

>> The note about BioRender should be moved from the Acknowledgements to a Graphics section that should be part of Materials & Methods, using the format "(some of the... OR Figure #... OR synopsis) Graphics were created with BioRender.com."

Please note that as of January 2016, our new EMBO Press policy asks for corresponding authors to link to their ORCID iDs. You can read about the change under "Authorship Guidelines" in the Guide to Authors here: <http://emboj.embopress.org/authorguide>

In order to link your ORCID iD to your account in our manuscript tracking system, please do the following:

1. Click the 'Modify Profile' link at the bottom of your homepage in our system.
2. On the next page you will see a box half-way down the page titled ORCID*. Below this box is red text reading 'To Register/Link to ORCID, click here'. Please follow that link: you will be taken to ORCID where you can log in to your account (or create an account if you don't have one)
3. You will then be asked to authorise Wiley to access your ORCID information. Once you have approved the linking, you will be brought back to our manuscript system.

We regret that we cannot do this linking on your behalf for security reasons. We also cannot add your ORCID iD number manually to our system because there is no way for us to authenticate this iD number with ORCID.

Thank you very much in advance.

Please use the link below to submit your revision:

Response to Reviewer comments (responses in red)

Please find Reviewer comments from our previous submission to another venue, and our responses, below. We note that Reviewers 2 and 3 were overwhelmingly positive and were convinced of the robustness of our methodology as well as the validity and importance of our findings. Reviewer 2 highlights how “compelling” our data are, and Reviewer 3 even encourages us to provide guidelines on how our findings should alter the day-to-day practices of cell culture users. We have addressed Reviewer 1's suggestion to conduct our experiments in more physiologically relevant cell types. However, Reviewer 1's suggestions to study proliferative (namely cancer) cells is beyond the scope of this manuscript which is focused on terminally-differentiated cell-types. Overall, we have taken on board the Reviewers valuable critique, which we believe have substantially improved the manuscript. Please see details of our responses below. However, we note here the four major changes to the manuscript in response to these reviews:

1. We now include data from human adipocytes and primary mouse adipocytes alongside our extensive data from 3T3-L1 adipocytes. We also include data from cultured brown adipocytes, myotubes, human iPSC-neurons, macrophages, human iPSC-derived cardiac organoids, and human iPSC-derived hepatocytes. Our observations from 3T3-L1s hold true for these cell models too. Therefore, while we use the 3T3-L1 adipocyte as an example cell line, we show that our findings relating to pericellular oxygen and the phenotypic consequences of culturing cells in suboptimal oxygen conditions are broadly applicable to multiple adipocyte cell lines and a range of other cultured cell lines, including human-derived cells. This highlights that our findings related to the importance considering oxygen tension is highly generalisable. As such, we do not doubt that our work would be of very high interest to the broad readership of *EMBO Journal*. These are the first data on the *phenotypic* significance of culturing cells in low oxygen we are aware of.
2. We acknowledge our conclusions linking the transcriptional responses to hypoxia in adipose tissue (Fig. 3F) and lowering medium volumes in 3T3-L1 were overstated. We have rectified this in the text. Though we agree our transcriptomics analysis may be insufficient to claim that we are making 3T3-L1s more like *in vivo* adipose tissue without additional supporting evidence, we think it does demonstrate that the changes driven by hypoxia *in vivo* and *in vitro* have significant overlap in terms of the impacted transcriptional pathways. This is a non-trivial observation since these results demonstrate that the same pathways induced or suppressed by hypoxia *in vivo* are similarly altered in our cell line. The broader relevance of this observation is that conclusions drawn about the impact of oxygen tension *in vitro* are likely to be relevant to *in vivo* adipose tissue.
3. We have included reference to *in vivo* studies to compare, and provide context for, oxygen tensions in cultured cells (measured here). The ppO_2 in adipose tissue in normoxia is ~60 mmHg (Midha *et al.*, 2023 *Cell Metabolism*; similar to observations in human adipose tissues Fleischmann *et al.*, 2005 *Obes Surg*). This is substantially higher than the 11 mmHg cells experience in high medium conditions. In our manuscript, the low medium condition more accurately recapitulates the *in vivo* oxygen tension (56 mmHg). In addition, glucose-derived lactate release in human subcutaneous abdominal adipose tissue showed that only 15-30% of total glucose uptake was partitioned to lactate (Hodson *et al.*, 2013 *Diabetes*). Our work shows that in high media, a substantially greater proportion of glucose appears to be converted to lactate (close to 50%; Fig 1H). Again, the extent of lactate production from glucose in 3T3-L1 adipocytes cultured at a higher oxygen tension (17% lactate from glucose; Fig 1H) is more similar to the *in vivo* values.
4. Finally, we have taken on board reviewer comments to provide a more direct comparison between the hypoxic response under standard culture condition and those elicited by culturing cells in 1% oxygen. These data provide important context, especially for those in the hypoxia field, confirmation that A) there is measurable HIF activity in cultured adipocytes in standard culture conditions; B) that the benefits of moving cells to lower medium volumes is attributable to increased oxygen delivery; and C) cultured adipocytes can mount a substantial future HIF response when cultured at 5% or 1% oxygen.

Reviewer #1 (Remarks to the Author):

The manuscript by Tan and collaborators proposes that under standard cell culture conditions, oxygen consumption by the cells leads to pericellular hypoxia and a hypoxic cellular phenotype. Although oxygen diffusion through water/medium is a slow process, it is difficult to envisage that oxygen demand from cells growing in monolayer is higher than the levels of oxygen present in culture medium under standard culture conditions. In particular, when the main cells used in the study are white adipocytes, known to have very reduced number of mitochondria and limited OXPHOS in vivo.

In direct contrast to the Reviewer's assertion, we provided several lines of evidence that oxygen concentrations are limiting at the cell monolayer for 3T3-L1 and primary adipocytes in this study. This includes directly measuring pericellular O₂, measuring an increase in OCR and a switch in glucose use from conversion to lactate to oxidation when more O₂ is provided, and showing stabilisation of HIF1α under standard culture conditions. These experiments indicate that the cells sense lower O₂. Together, these empirical data strongly refute the Reviewer's unsubstantiated opinion.

Adipose tissue is highly oxidative in its metabolism. For example, adipose tissue is a significant site for BCAA metabolism which can only be fully catabolised via the TCA cycle. Although the mitochondria number per g of fat tissue is likely low, this is not representative of mitochondria per adipocyte, and is hard to compare to other tissues, since a large amount of adipose tissue mass comes from lipids. Indeed, electron micrographs from white adipose tissue show a high density of mitochondria (Acín-Perez *et al.*, 2021 *Redox Biol.*; Choo *et al.*, 2005 *Diabetologia*).

Please see: Figures 1, 3A, 3C.

In the introduction, it is mentioned that cell lines are known to be highly glycolytic and that this is usually explained by a metabolic rewiring of cancer cells. The authors propose that a simpler explanation for the high glycolytic rates of many cultured cells is that oxygen is limiting. However, in the field of hypoxia is well know how challenging it is to detect HIF-1alpha. In cells under standard culture conditions in a regular incubator (21% O₂), the protein is not detectable by western blot (there are exceptions when cancer cell lines express oncogenes that target HIF-1alpha degradation pathway or HIF-1alpha translation). And even in cells exposed to hypoxia, the HIF-1alpha protein is only detected by western blot if the researchers collect the cells in special conditions.

We agree that detecting HIF1α can be challenging. Our new data in Figure 3B shows that HIF1α disappears rapidly upon reoxygenation. Therefore, we have been careful to optimise our methods to ensure that degradation does not occur in the lysates. By supplementing our 2% SDS lysis buffer with CoCl₂ and MG132 (proteasome inhibitor) we were readily able to measure HIF1α protein using western blotting (Fig 3B). Additionally, we were also able to assay HIF1α activity by qPCR (Fig 3A & S3A) under standard culture conditions in 18% O₂. It is not uncommon for specific proteins or protein post-translational modifications to require modified lysis conditions to reduce changes in protein/modification abundance after cell lysis. Our data obtained using this optimised lysis method to detect HIF1α protein closely match those from qPCR analysis of HIF1α targets.

Please see: Figures 3A, 3B, S3A.

Please see: Page 21, lines 781-783:

"Immunoblotting of HIF1α

Cells were washed three times in ice-cold PBS and lysed in 2% SDS containing protease and phosphatase Inhibitors (ThermoFisher), 500 μM CoCl₂, and 10 μM MG132."

Reoxygenation leads to degradation of the protein in a few minutes. In this context, to demonstrate their hypothesis, the authors should show that HIF-1alpha can be detected in cancer cell lines under standard culture conditions.

We agree that reoxygenation can rapidly lead to HIF1α degradation. We now show that HIF1α is degraded within 5 min of transfer to low medium (higher oxygen) (Figure 3B), consistent with the kinetics of HIF1α degradation. These data clearly support the notion that standard culture conditions result in HIF1α stabilisation in adipocytes.

We are not clear about the link the Reviewer is drawing between reoxygenation and cancer cells. Cancer cells are proliferative, typically grown only to pre-confluence, and so represent a very different model system than the post-mitotic terminally differentiated cells we have focussed on. In our cell types of interest,

under standard culture conditions (1 mL of medium in a 12-well plate), we clearly detected HIF1 α expression (Figures 3A, 3C). We do not plan to extend our analyses to cancer cell types, which would be a very distinct study. We are also unclear of the merits of such a study, as oxygen diffusion in cancers may be better addressed in organoid models. We have included discussion of the role confluence may play in whether cells experience hypoxia under standard conditions.

Please see: Figures 3A, 3C, S1H, S1I.

Please see: Page 14, lines 500-506:

“Indeed, while we have focused exclusively on confluent terminally-differentiated cell types, our findings are likely more widely applicable. However, rapidly proliferative cell-types, such as cancer or immune cells are usually grown and studied at pre-confluence. Since a key determinant of oxygen consumption at the cell monolayer is cell density (Figure S1H and S1I), whether specific cells in culture experience hypoxia and would functionally benefit from increased oxygen needs to be determined on a cell type-to-cell type-basis.”

Cell lines derived from different types of cancers should be used (e.g. breast cancer, hepatoma, colon cancer). Some of the experimental approaches regarding gene expression and metabolism should be also done with these human cancer cell lines. Of course, these experiments should be carry out without letting the cells become overconfluent since this is known to increase lactate production, decreased medium pH and promoting cell death. To avoid overconfluence, the experiments should be done with a cell confluence of 80-90%.

As noted above, while we agree that cancer cell line studies could be a new avenue of research, it is a distinct question to our work. There are major biological differences between cancer and terminally differentiated cell types, not least that cancer cells are proliferative, and thus optimisation of cancer cell culture for the assays presented in this manuscript remain out of scope. All cells used in this study were terminally differentiated and therefore confluent. We agree that confluency likely determines the rate of O₂ used by the cell monolayer, and therefore the degree of O₂ limitation per cell. To test this directly, we have re-seeded differentiated adipocytes at lower cell densities. As predicted, this resulted in lower OCR (Figure S1H), but also the amount of lactate produced relative to glucose uptake (Figure S1I). These data suggest that the difference in metabolism with increased oxygen provision is likely driven by a combination of cell-intrinsic OCR and cell density.

Please see: Figures S1H, S1I.

The large majority of the experiments done in the manuscript use 3T3-L1 adipocytes that are generated from a fibroblast precursor cell line. 3T3-L1 adipocytes have lost metabolic flexibility and in contrast to primary preadipocytes can only be differentiated into white adipocytes. Furthermore, they derived from a cell line that presents aneuploidy with an instable karyotype. We also do not know if there are mutations that target the HIF-1 α pathway. The relevance of this study can only be demonstrated if the majority of the experiments are performed using primary adipocytes. The authors can either use mouse primary adipocytes derived from preadipocytes of the stromal fraction of subcutaneous adipose tissue or use human primary adipocytes derived from commercially available human mesenchymal stem cells.

We have already provided evidence that a number of cell types other than 3T3-L1s respond to lowering media volume by switching metabolism to being less glycolytic (pBAT adipocytes, L6 myotubes, iPSC-derived cardiac organoids) and lowering HIF1 α activity (macrophages, iPSC-derived neurons and hepatocytes). However, to further address the Reviewer's concern regarding the 3T3-L1 cell line, we have included additional data using human adipocytes (hMADs) and a large number of experiments in mouse primary cultured adipocytes (from subcutaneous adipose tissue). These studies in primary cells completely phenocopy data from 3T3-L1s, validating the use of the 3T3-L1 model and the generality of our findings to cultured adipocytes. They also respond to further hypoxic intervention showing that they have a regulatable and responsive HIF system.

Please see: Figures 1H, 3D, S3B-E, 4G-I, S4E, S4F, 5, S5.

If under standard culture conditions (21% O₂), oxygen is limiting, culturing the cells in hyperoxia should overcome this limitation. The authors should present data of cells cultured in a hyperoxia incubator chamber.

The Reviewer is correct that providing more O₂ to cells under standard culture conditions overcomes limitations in O₂. We have addressed this question using the Lumox plates which have an O₂-permeable plastic base (see Figs S1E, 2C, S3A). Using these plates we showed that, compared to standard tissue

culture plastic, cells cultured in these plates had lower lactate production, increased lipid synthesis and less HIF1 α activity - all consistent with greater O₂ provision. We have now also conducted reversal experiments in hypoxia incubators (5% O₂) and chambers (1% O₂), showing that the adipocytes can still respond to even lower O₂ supply, but ambient hypoxia attenuates the metabolic and transcriptional effects of low medium intervention.

Please see: Figures S1E, 2C, S3A (Lumox), 3D, S3B-E (5%/1% hypoxia).

Regarding the studies comparing gene expression between 3T3-L1 adipocytes and adipose tissue of mice at normoxia or hypoxia, a graphic representation of HIF-1 α target gene expression should be presented.

HIF1 α target genes are upregulated under standard culture conditions in 3T3-L1 adipocytes (Fig 3A). However, this is not the case in the adipose tissue from mice kept under hypoxic conditions for 4 weeks. It is known that the HIF response caused by transitioning mice to hypoxia is transient, and therefore likely has declined by 4 weeks. We note that despite this, the overall hypoxic transcriptional response between cultured cells in standard media volumes and hypoxic adipose tissue is remarkably similar.

Please see: Figures 3A, 3F.

In a more practically aspect of this study, the optimal culture condition defined in the manuscript to avoid pericellular hypoxia is a 2,4 mm medium height. I believe there will be a high risk of evaporation even for a period of 16 h of culture. Was the volume of the medium measured after the 16 h incubation? To use such a small volume of medium, a perfusion system might be required to avoid evaporation and changes in medium concentration.

The standard media height used in cell culture is 2.4 mm (1 mL in a 12-well plate), so evaporation is not a major issue in properly humidified incubators. Under low medium conditions, this height is lower, ranging between 0.7-0.9 mm depending on the culture plate used. There will inevitably be evaporation during the 16 h of culture, and we optimised the volumes used and experimental timings to account for this. We measured medium volumes after 16 h and 24 h showing that though medium evaporation is unavoidable, it is only a <10% reduction after 16 h (when most of our experiments are conducted) (Reviewer Figure 1).

We agree that using a perfusion system is in some ways a better method to overcome oxygen and nutrient limitations due to better control over these factors, however the point of our methodology is to provide an accessible intervention to 1) study the importance of oxygen in cell culture and 2) allow laboratories across the world to use the same methodology to test their cells in culture. We have included reference to perfusion systems in our discussion.

Please see: Reviewer Figure 1.

Please see: Page 14, lines 480-483:

“Other methods to increase cellular oxygen have been employed (Place et al, 2017), but bioreactors or perfusion systems most accurately recapitulate the sophisticated oxygen delivery system of the vasculature since they continuously supply oxygenated medium (Sucusky et al, 2004). However, scalability of and accessibility to this method pose barriers to widespread use.”

Plate	12-well					
	16			24		
Time (h)	Start	End	% Evaporated	Start	End	% Evaporated
High (standard)	1000	931	7.0 \pm 1.0	1000	882	11.8 \pm 0.9
Mid	666	615	7.7 \pm 1.2	666	602	9.6 \pm 0.7
Low	333	304	8.7 \pm 1.6	333	282	15.3 \pm 1.4

Reviewer Figure 1. Start and end medium volumes in 12-well plates after 16 h or 24 h of low medium intervention. Mean percentage evaporation is shown \pm standard deviation (n = 4-8).

Reviewer #2 (Remarks to the Author):

*Key results: Please summarize what you consider to be the outstanding features of the work.

The authors sought to investigate how the amount of available oxygen present in adherent cell culture influences cellular metabolism. The authors characterized the amount of diffused oxygen present at the surface of adherent cells and found that the available pericellular oxygen differs substantially from the assumed concentration of dissolved O₂ given the amount of oxygen present in standard tissue culture incubators (21%).

Their work challenges the basic assumption that cells cultured at atmospheric oxygen (21% oxygen) are hyperoxic versus in vivo. Their data also suggests that the glycolytic phenotype common to cultured cells is an artifact of low dissolved oxygen present in culture media, at least in the case of terminally differentiated cells. The authors show that by increasing the amount of dissolved oxygen, post-mitotic adipose cells shift toward a more oxidative metabolic phenotype, with increased TCA cycle activity, oxygen consumption, and lipid anabolism. In contrast, and contrary to popular belief, cells cultured under standard conditions are limited for oxygen and display a hypoxic phenotype, including elevated lactate secretion and HIF1a stabilization.

We thank the Reviewer for such positive comments. We agree that our data challenges a basic assumption about cell culture.

*Validity: Does the manuscript have flaws which should prohibit its publication? If so, please provide details.

The work presented and the conclusions the authors reached from their data are valid and should not inhibit publication. That being said, I think the authors would do well to emphasize that they are working with terminally differentiated cells, which would have a different metabolic phenotype versus immortalized cell lines or primary cancer cells. Therefore, the authors should consider making this clear in the title of their work so as not to confuse readers.

We agree that this was not clear and have now changed our title and writing in the manuscript to highlight that we are working in terminally differentiated cells.

*Originality and significance: If the conclusions are not original, please provide relevant references. On a more subjective note, do you feel that the results presented are of immediate interest to many people in your own discipline, and/or to people from several disciplines?

The results could be viewed as interesting and unexpected given the general assumption that all cells cultured in standard tissue culture incubators are hyperoxic. Their data is compelling and supports their conclusion that terminally differentiated cells exhibit a hypoxic phenotype under standard culture conditions, and increasing the amount of available oxygen is sufficient to switch the cells to oxidative metabolism. If space allows, it would be interesting if the authors commented on how their conclusions may impact, if at all, cells cultured in suspension.

This is a very interesting idea, and have included this our discussion. We note that cells in suspension, and indeed cells grown in air-liquid interfaces, may not suffer from the same O₂ limitations and this is an important comparison to make.

Please see: Page 14, lines 480-486:

“Other methods to increase cellular oxygen have been employed (Place *et al*, 2017), but bioreactors or perfusion systems most accurately recapitulate the sophisticated oxygen delivery system of the vasculature since they continuously supply oxygenated medium (Sucosky *et al*, 2004). However, scalability of and accessibility to this method pose barriers to widespread use. Suspension cell culture employs a similar principle, whereby growing cells under conditions where the medium is continuously agitated will likely allow for greater oxygen provision (Cooper *et al*, 1958).”

*Data & methodology: Please comment on the validity of the approach, quality of the data and quality of presentation. Please note that we expect our reviewers to review all data, including any extended data and supplementary information. Is the reporting of data and methodology sufficiently detailed and transparent to enable reproducing the results?

The experiments that were performed are logical and well laid-out. The methods are thorough and the setup for most experiments is clear (with minor comments below). If the reviewers address the below minor comments, their work should be sufficiently detailed to allow for reproducibility.

Thank you.

Minor comments:

- The OCR experiments (as in Figure 1B) need to be better labeled. An arrow pointing at the 6 hour mark, where media is exchanged, would make the graphs much clearer.
- The RNA expression figures (as in Figure 4C) are difficult to interpret. The use of statistical significance is difficult to understand and interpret. It seems necessary to present the data in a different way for clarity.
- The authors should include which plastics (96-well, 24-well, etc) were used in each legend caption for clarity. This is noted on some of the figure captions, but consistency would be appreciated.
- Figure 2H is difficult to interpret. I recommend splitting the figure into two graphs, one which shows the increase in aKG in high versus low medium volumes, and another that shows the normalized fractional labeling.
- Figure 3H needs a legend to label the dark- and light-colored bars.
- Some of the data and figures are difficult to read if printed in black-and-white (e.g. Figure 1B, 2A, 2F).

We thank the Reviewer for their suggestions. We noted and have made these changes where appropriate.

*Appropriate use of statistics and treatment of uncertainties: All error bars should be defined in the corresponding figure legends; please comment if that's not the case. Please include in your report a specific comment on the appropriateness of any statistical tests, and the accuracy of the description of any error bars and probability values.

All error bars and p-values are defined at the end of each figure legend. Overall, the tests used (ANOVAs, t-tests) are appropriate and the description of error bars and p-values are clear and accurate. See minor comments below.

Minor comments:

- Some comparisons are not labeled with statistical significance (even if the comparison is not significant). These include Figures 1F, 2D, 3F, and 3H.

We have made sure comparisons have been labeled with statistical significance.

- The presentation of statistical tests used for Figures 4C and S4A are confusing. I recommend reformatting the data for these experiments to make it more clear about what groups of data are being compared and the statistical significance of any phenotypes observed.

We have reformatted these data. To increase clarity, we chose to present only data from hepatocytes cultured chronically in either high or low medium volumes throughout differentiation (rather than having both chronic and acute medium changes) to simplify the data and presented statistical analyses with clearer comparisons.

Please see: Figures 5C, S5C, S5E, S5G.

*Conclusions: Do you find that the conclusions and data interpretation are robust, valid and reliable?

Overall, the conclusions the authors draw from their experiments are valid and robust. The authors employ several orthogonal approaches to validate the observed phenotype that increasing oxygen availability shifts terminally-differentiated cells to an oxidative metabolic signature.

That being said, the conclusions the authors draw from their comparison of their cell culture model to their mouse model need to be clarified. The authors conclude that standard culture conditions mimic hypoxic

adipose tissue, which is supported by their gene expression correlation data. The authors go on to conclude that increasing oxygen tension in culture better models normoxic adipose tissue. However, this conclusion is far too strong to propose based on the data they present. Further, it would be important for the authors to include information about what oxygen tension is normoxic for adipose tissue.

We understand the Reviewer's concern and agree that our claim that standard culture conditions mimic mice hypoxic adipose tissue were overstated. We have revised this section carefully. Whilst we agree our transcriptomics analysis may be insufficient to claim that we are making 3T3-L1s more like *in vivo* adipose tissue without additional supporting evidence, we think it does demonstrate that the changes driven by hypoxia *in vivo* and hypoxia *in vitro* have significant overlap in terms of the impacted transcriptional pathways. These results demonstrate that the same pathways that are induced or suppressed by hypoxia *in vivo* are similarly altered in our cell line, and suggest that the response to hypoxia experienced in standard cell culture by cultured adipocytes is physiological. The broader relevance of this observation is that conclusions drawn about the impact of oxygen tension *in vitro* are likely to be relevant to similar situations *in vivo* adipose tissue.

We have also included reference to oxygen tension in normoxic adipose tissue. Whilst we have not made these measurements ourselves, a recent publication has shown that Clark-type microsensor measurements of ppO₂ in normoxic mice white adipose tissue is ~60 mmHg (Midha et al., 2023 *Cell Metabolism*). This is similar to our findings in mouse-derived 3T3-L1s where oxygen tension is 56 mmHg in low medium vs 11 mmHg in high medium conditions. In lean humans, oxygen levels measured in abdominal subcutaneous adipose tissue ranges between 40.5–73.8 mmHg (Pasarica et al., 2019 *Diabetes*), which again, is more comparable to our measurements in low medium conditions than in cells under standard culture conditions.

In addition, glucose-derived lactate release in human subcutaneous abdominal adipose tissue showed that 15-30% of total glucose uptake was partitioned to lactate (Hodson et al., 2013 *Diabetes*). Our work shows that in high media, a substantially greater proportion of glucose appears to be converted to lactate (close to 50%; Fig 1H). Again, the extent of lactate production from glucose by 3T3-L1 adipocytes cultured at a higher oxygen tension (17% lactate from glucose; Fig 1H) is more similar to the *in vivo* values.

Please see: Page 6, lines 172-180, under results section "Oxygen is limiting for adipocyte respiration under standard culture conditions", and page 7, lines 201-209, under results section "Increasing pericellular oxygen tension decreases lactate production in adipocytes" for comparisons of *in vitro* to *in vivo* adipose tissue oxygen tension and lactate production respectively.

Please also see: Pages 9-10, lines 315-333, under results section "Lowering medium volumes induces a widespread transcriptional response reminiscent of physiological hypoxia" for a more accurate interpretation of the RNAseq data.

*Suggested improvements: Please list additional experiments or data that could help strengthen the work in a revision.

The authors used cell culture media with high/supraphysiologic concentrations of glucose (25 mM), which may have impacted glucose metabolism of their cultured cells. Since an altered glucose utilization phenotype is one of their core conclusions, it may be worth repeating the glucose consumption / lactate secretion experiments and the OCR experiments with culture media that has more physiologically relevant glucose concentration (e.g. 5mM) to confirm that this phenotype is consistent.

We have performed this experiment and our data show that the metabolic changes observed when cells are cultured in low medium (i.e. changes in glucose uptake and lactate production) are consistent across various starting glucose concentrations (25, 20, 15, 10, 5 mM). Therefore, the observed changes in glucose use with increased O₂ is not driven by supraphysiologic medium glucose concentrations. We have included these data in the revised manuscript.

Please see: Figure S1G.

*References: Does this manuscript reference previous literature appropriately? If not, what references should be included or excluded?

It may be relevant to reference Jain et al, Genetic Screens for Cell Fitness in High or Low Oxygen Highlights Mitochondrial and Lipid Metabolism, 2020 in the introduction, since the authors talk about how oxygen availability affects gene expression and utilization. Per the above comment, it may also be worth

noting and citing additional efforts into considering (and improving) the modeling capacity of culture media (HPLM, Plasmax, etc) otherwise, particularly as the authors use a reagent containing 25 mM glucose in their work.

These are excellent suggestions and we have included discussion of this work in our revised manuscript. We agree that our findings on O₂ in cultured cells go hand-in-hand with efforts to develop better physiologic media compositions to improve cell models and their translatability.

Please see: Page 15, lines 523-529:

“Our findings ultimately highlight that cell culture is a state of variable oxygen tension depending on factors such as oxygen consumption by cells and limitations of oxygen diffusion through the media column. Manipulating oxygen levels can cause dramatic effects on many aspects of cellular metabolism and function (Jain et al, 2020). As such, these findings complement recent data on the use of more physiological media (Vande Voorde et al, 2019; Cantor et al, 2017), with potentially important implications for the translatability of both cell and tissue culture models to in vivo settings.”

*Clarity and context: Is the abstract clear, accessible? Are abstract, introduction and conclusions appropriate?

In paragraph 2 of the introduction (lines 78-89), the authors seem to imply that the cell lines they are working with are highly proliferative, and therefore glycolytic. However, their cell lines all appear to be derived from progenitors and are terminally differentiated for their experiment and therefore no longer rapidly proliferating. The authors should clearly state in the abstract (and perhaps the title) that they are working with non-proliferating cell types, which will have a different metabolic signature from proliferating cell types. Their concluding paragraph should also re-emphasize that their work is applicable to fully differentiated cells and that their conclusions should not be applied to rapidly proliferating cell types like cancer or immune cells.

We agree and have corrected to make it clear in the title, abstract, and discussion that we are working with terminally-differentiated confluent cells.

Reviewer #3 (Remarks to the Author):

Tan et al. address one of the well-known limitations in in vitro cultures, that is the importance of accounting for pericellular O₂ levels, published previously, for example, in <https://doi.org/10.1371/journal.pone.0152382>.

We agree other studies have highlighted that O₂ may be limiting under standard conditions. We have increased our references to include this paper and others. However, we note that these previous studies did not assess the *importance* of considering O₂ for cultured cell phenotypes as we have provided in detail in Figures 4 and 5. We hope that the stark changes in adipocyte hormonal responses, cardiac-organoid contraction and in hepatocyte differentiation will provide a strong impetus for researchers to assess how O₂ tension might be affecting their cells and to optimise this parameter.

Using sound methodology, a number of cell lines and organoids, the authors show that cellular oxygen consumption in conventional culture systems is high resulting to lower pericellular oxygen levels (thus hypoxia). By adjusting the media volume (LM), increased oxygen availability and altered metabolic responses and gene expression signatures. Tan et al. also found similar gene expression pathways between the conventional (HM) 3T3 cells with SAT-derived adipocytes from a chronic hypoxic mouse model.

Although, Tan et al. provide important insights on the limitations of the current widely used culture systems, in particular the importance of controlling for oxygen availability, there are some limitations in the experimental design that need to be addressed.

1. How did the 3T3 differentiation status compare between LM and HM? If HM keeps cells in a hypoxic state, then the differentiation process should be affected. 3T3s should be then kept in LM (vs HM) for differentiation and then compared.

Our work was largely focused on post-differentiated adipocytes. Performing the suggested experiment is challenging using our low medium strategy to increase oxygen provision due to both evaporation and nutrient depletion during the differentiation protocol, which involves 3-day incubations. However, we have now performed this experiment using 24-well Lumox (gas-permeable) plates, and also in standard 24-well culture plates using 0.25 mL instead of 0.17 mL as the low medium volume condition (Reviewer Figure 2). However, we saw no difference in Oil Red O staining between standard and Lumox plates. It has previously been reported that hypoxia interferes with adipocyte differentiation but this was typically conducted under severe hypoxia (e.g. 1% O₂) (Anvari & Bellas 2021 *Sci. Rep.*) Our interpretation is that the level of hypoxia experienced by 3T3-L1 adipocytes under standard culture conditions is not sufficient to inhibit differentiation, however does result in major functional changes (Figure 4). We also noted a decrease in lipid accumulation (as measured by Oil Red O) in low medium conditions both in the standard and Lumox plates, suggesting that adipocyte differentiation may be more dependent on medium nutrients than volumes. We have provided the data on differentiation below, for reviewers, but we have not included these in the manuscript.

Please see: Figure 4, Reviewer Figure 2.

Looking at comparing data in Fig3d and Fig3a, Table S2. HM is hypoxic and stabilizes HIF1, then would you not expect higher leptin expression in the HM? [https://www.jbc.org/article/S0021-9258\(19\)71890-0/fulltextdoi: 10.1677/JOE-08-0156](https://www.jbc.org/article/S0021-9258(19)71890-0/fulltextdoi:10.1677/JOE-08-0156).

Thank you for providing this reference. Based on the data in this paper, we would expect leptin mRNA expression to be decreased in the lower medium/higher O₂ conditions. However, we note that at the mRNA level, *Lep* expression is not altered between media conditions (log₂FC = 0.28 (low media/high media) adj. P 0.054), although it trends upwards. This raises the possibility that increased medium leptin stems from increased translation or increased efficiency of secretion.

One explanation for us not observing an effect on *Lep* mRNA as we moved cells to greater O₂ is that the data presented in this reference is not from adipocytes, but from a leptin reporter construct expressed in the choriocarcinoma cell line BeWo. As such, there may be other transcriptional factors at play in addition to HIF1 α in adipocytes, which express *Lepn* endogenously, to modulate *Lep* expression.

Reviewer Figure 2. Oil Red O staining of 3T3-L1 adipocytes differentiated in either standard (top) or Lumox/gas-permeable (bottom) plates. Cells were cultured under high (0.5 mL) or low (0.25 mL) medium volumes throughout differentiation (10 days).

2. Referring to Fig 3A. “We first determined if transitioning cells from high to low medium altered HIF1 α stabilisation. HIF1 α abundance was drastically decreased when cells were switched to low medium for 16 h”.

How do LM, HM HIF levels compare with hypoxia-induced HIF? Why cobalt chloride (as positive control) was used instead of hypoxia, if the point of the manuscript is focused on how important is oxygen tension/availability?

We initially used CoCl_2 as a more convenient control for our experimental set up, since using hypoxia meant having cells split across different plates. We now have data directly comparing HIF1 α protein in low medium (LM) and high medium (HM) in 1% and 18% O_2 . These data show that 1) both LM and HM cells respond to 1% O_2 with stabilized HIF1 α and 2) that LM in 1% or 5% O_2 has less HIF1 α stabilisation/activity than HM cells in 18% O_2 .

Please see: Figures 3D, S3E.

HIF1A western blot, how many times this was repeated, and how many biological replicates?

This was repeated with 3 biological replicates as stated in the legend of Figure 3.

Please see: Figure 3C.

Can you clarify, is the HM in this experiment 1.5% O₂ and your LM... 3%, 6%? How do all these HIF-target genes compare with the standard Hx conditions (what is the pericellular oxygen concentration under standard Hx conditions?).

Fully oxygenated medium is 181 μ M oxygen at 18% air oxygen (10% CO₂ incubator). Converting our medium oxygen measures from μ M to %, HM in this study corresponds to 15 μ M or 1.5% oxygen at approx 0.55 mm (the minimum measurable depth) above the cell monolayer. In LM, this is 73 μ M or 7.3 %. However, based on Fig 1E, we demonstrate that in a closed system, 3T3-L1 adipocytes can respire maximally down to ~4 μ M oxygen, suggesting that the cells can use almost all available oxygen down to almost 0 % at the cell monolayer.

We understand that the reviewer is asking about how the degree of hypoxia we measure under standard culture conditions compares to when cells are cultured under standard hypoxic conditions. We have now performed additional comparative analysis to come glucose and lactate metabolism, cellular oxygen consumption, as well as degree of HIF1 α stabilisation and activity under high and low medium volumes in standard hypoxia conditions (1%/5% O₂). We noted that the extent of HIF1 α stability and activity was much greater in hypoxia vs standard culture conditions (Figures 3D and S3E). Although adipocytes still mounted a considerable lactate response to further hypoxia, the effect of switching to low medium volumes was attenuated in cells cultured in either 1% or 5% O₂ (Figures S3B-D). These data suggest that though the HIF system is active in 3T3-L1 adipocytes under standard culture conditions, the cells still had the capacity to respond to even lower oxygen supply.

Please see: Figures 1E, 1G, 3D, S3B-E.

In Fig legend. Fig,3B analysed by paired test, what is the pairing?

Now Figure 3A, each biological replicate is a pair (e.g. n = 1 high vs low, n = 2 high vs low, etc).

What is the relevant oxygen tension that you suggest to culture cells to be comparable to... for example adipose tissue oxygen tension, providing that this is stable, uniform in the tissue?? Should we not then be accounting for cycles of intermittent changes in oxygen tension etc?

First, a recent paper has shown that Clark-type microsensor measurements of ppO₂ in normoxic mice white adipose tissue is ~60 mmHg (Midha et al., 2023 *Cell Metabolism*). This is similar to our findings in mouse-derived 3T3-L1s where oxygen tension is 56 mmHg in low medium vs 11 mmHg under high medium conditions. In lean humans, oxygen levels measured in abdominal subcutaneous adipose tissue ranges between 40.5–73.8 mmHg (Pasarica et al., 2019 *Diabetes*), which again, is more comparable to our measurements in low medium conditions than in cells under standard (high medium) culture conditions.

However, this is an excellent question and the main thing we ask ourselves in the overall study. One of the key differences between cell culture and *in vivo* tissue is that an oxygen tension gradient exists across the column of the culture medium due to relatively slow oxygen diffusion and oxygen consumption by cellular respiration, whereas a constant perfusion by the vascular system supplies oxygen to cells *in vivo*. We do not suggest that our method be comparable to the oxygen tension of adipocytes *in vivo*. As the Reviewer pointed out, during cell culture there will be fluctuations in oxygen tension during medium changes, as cells proliferate, etc. Thus, this is a major reason for this study, as we want these questions to be raised by researchers when they do their experiments, rather than adding 'typical' volumes of culture medium to their cells, or adding more/less medium for convenience (e.g. over the weekend).

Please see: Page 6, lines 172-180, under results section "Oxygen is limiting for adipocyte respiration under standard culture conditions" for comparison of *in vitro* to *in vivo* adipose tissue oxygen tension.

3. Experiments Fig S3. Please, explain the rationale of comparing 3T3s (16h in LM or HM) with sc adipocytes after whole body exposure of mice in 10% O₂ for 4weeks. In what physiological setting these 2 would be comparable? I don't think this is the right way to come to the conclusion in the abstract etc... "Importantly, pathway analyses revealed increasing oxygen tension made *in vitro* adipocytes more similar to *in vivo* adipose tissue". It seems an overstatement, if the conclusion comes from this experimental design. Specifically,

(a) why comparing gene expression of chronic whole-body hypoxia exposure (that will inevitably affect other metabolic organs with paracrine effects to adipose) with, yes widely used, but far from ideal- 3T3 cellular model (16h) would allow meaningful physiological (translational) comparisons?

Firstly, this is a fundamental point about all cell culture - how can any *in vitro* system recapitulate the complex interplay of interorgan communication? While we agree that cell culture can only ever be a model, we think it is an important and worthwhile endeavour to try to make it more applicable to the *in vivo* state. While we think we have gone some way to achieving this we do agree with the Reviewer insofar as we may have been overenthusiastic in our wording. As mentioned in response to Reviewer 2, we have readdressed this section. Specifically, we have described the rationale for this comparison clearly, and altered our conclusion to say that this comparison allows us to state that the transcriptional response in 3T3-L1s to standard culture conditions is similar to the response observed in hypoxia adipose tissue. We agree with the Reviewer that the models systems and interventions are distinct, but, despite this, there was a substantial overlap in the transcriptional responses. This suggests that standard culture conditions at least invoke a hypoxia response in cultured adipocytes that induces or suppresses the same transcriptional pathways that are affected in adipose tissue by hypoxia *in vivo*.

Please see: Pages 9-10, lines 315-333 under results section "Lowering medium volumes induces a widespread transcriptional response reminiscent of physiological hypoxia" for a more accurate interpretation of the RNAseq data.

(b) What is the oxygen concentration in the culture compared to the adipose tissue in hypoxia in your experiments?

We did not make this comparison in our own samples due to technical limitations. However, the paper mentioned above (Midha et al., 2023 *Cell Metabolism*) which measured white adipose tissue ppO_2 in mice kept in 8% hypoxia (our mice were kept in 10% O_2) found that tissue ppO_2 under hypoxic conditions was ~25 mmHg, comparable to our 3T3-L1 adipocytes under standard culture conditions (high medium) which has an oxygen tension of 21 mmHg.

Please see: Page 6, lines 172-180, under results section "Oxygen is limiting for adipocyte respiration under standard culture conditions" for comparison of *in vitro* to *in vivo* adipose tissue oxygen tension.

(c) Why 4-weeks? What was the reason for choosing this timeframe, and how does it compare with the 16h?

These tissues were collected as part of another study. All metabolic research is moving towards a greater commitment to the 3Rs (reduction, refinement and replacement). As such this sample set provided us with a useful tool that required no extra animals to be used. Given the extremely high concordance between the direction and magnitude of changes to the transcriptional pathways between our cells and the *in vivo* tissues we considered this sufficient proof of concept to demonstrate a good level of translatability between our cell data to the *in vivo* setting. That said, we agree with the Reviewer that it is hard to know how the different time frames (in different models) compare. We would need to generate both time courses of exposure of animals to hypoxic conditions as well as time courses of cells exposed to low medium depths for different periods of time in order to comprehensively answer this question, which would be beyond the scope of this study.

(d) How does the oxygen consumption of 3T3s compare with the hypoxic primary adipocytes?

We do not know. A direct comparison of OCR between hypoxic primary adipocytes and 3T3-L1s would require necropsy and digestion of adipose tissue with collagenase to release the adipocytes. This process takes several hours and will lead to a substantive reoxygenation.

While we think it is an interesting question with respect to the translatability between the two cell types, we think that perhaps the more key question is how the balance of oxygen supply vs consumption compare? The current model for oxygen sensing by cells is based on oxygen concentration, not consumption rates. Consumption is relevant as the steady state oxygen levels of a system (either cell culture or tissue) represent the balance of oxygen production and consumption. As mentioned above in our response to 3(b) the data regarding oxygen concentrations in adipose tissue and 3T3-L1 cell culture are comparable. We can therefore conclude the relative consumption of oxygen as a proportion of the supply is comparable between the two models.

(e) How do HIF1 protein levels compare in the 2 systems?

Unfortunately, we do not have tissue samples available to draw this direct comparison.

1. introduction." ...substantially higher than the oxygen tension human tissues are exposed to, which ranges between 6.5-130 μM ". Which is the reference for this?

The reference was Ast & Mootha 2019 *Nature Metabolism*, however we have now rewritten a significant part of our introduction and did not keep that statement.

2. expression "under low oxygen tension". In what context, please be specific what tPO₂ exactly (or range)?

We agree that this sentence suggests a specific point at which glycolysis switches to being more anaerobic. We do not think it is correct to suggest this system is binary. So we have reworded to indicate that the response to lower O₂ is for the cell to increasingly undertake anaerobic glycolysis as O₂ becomes limiting.

Please see: Page 7, lines 198-199:

"Anaerobic glycolysis is a hallmark of low oxygen availability, generating lactate instead of CO₂ as the end product."

3. "Medium height is not constant in cell culture wells due to the meniscus". What about evaporation of medium, especially in smaller volumes? How do you control for this?

We agree that these are important considerations. The experiments in the gas-permeable Lumox plates can control for the effect of lower volume *per se* (since these plates allow maximum O₂ provision at any medium volume). However, we cannot control for the effects of the meniscus or of evaporation. We tried to minimise the evaporation by ensuring incubators were fully humidified for each experiment, and performed optimisation experiments to work out the next volumes and experimental duration to use. We have measured medium volumes after 16 h and 24 h showing that though medium evaporation is unavoidable, it is only a <10% reduction after 16 h (when most of our experiments are conducted) (Reviewer Figure 1). We have also discussed this limitation in our manuscript.

Please see: Page 14, lines 472-474:

"Lowering medium volumes provides an accessible method to increase oxygen provision by decreasing oxygen diffusion distances, which can be easily employed by most laboratories. However, this method has its limitations, such as medium evaporation, nutrient depletion and altered concentrations of secreted factors."

4. Methods: referring to HIF1 α western blot-it is not clear from description, did you add CoCl₂ in lysis buffer?

We added CoCl₂ in the lysis buffer as stated in the methods. This was added along with MG132 to inhibit HIF1 α degradation post-lysis.

Please see: Page 21, lines 781-783:

"Immunoblotting of HIF1 α "

Cells were washed three times in ice-cold PBS and lysed in 2% SDS containing protease and phosphatase Inhibitors (ThermoFisher), 500 μM CoCl₂, and 10 μM MG132."

5. Elaborate in discussion, what exactly is proposed? New guidelines...change the way cells are cultured? Less volume of media, monitoring pericellular O₂?

We agree that this would be an important addition to our discussion and have added as suggested.

Please see: Page 15, lines 507-522:

"Second, our study highlights a critical distinction between incubator oxygen levels and the local oxygen concentration experienced by cells. This suggests that reporting of incubator oxygen % is insufficient and that measuring pericellular oxygen is required to reveal the actual oxygen concentration experienced by cells (Rogers *et al*, 2023). Notably, culturing 3T3-L1 adipocytes in low medium accurately matched *in vivo* oxygen tensions (Midha *et al*, 2023; Pasarica *et al*, 2009) as well as lactate export (Hodson *et al*, 2013). Shifting cells to a more physiological oxygen tension correlated with improved adipocyte function (Figure 4). Since cells *in vivo* exist on a spectrum of oxygen tensions (Ast & Mootha, 2019), we suggest that the best reference point for optimal oxygen tension for a specific cell-type is to match the pericellular oxygen to that of the relevant tissue *in vivo*."

Finally, the powerful effects of adjusting oxygen tension on cell phenotypes highlights the need for researchers to control and report on factors that impact oxygen availability (e.g. medium volumes and cell densities) to ensure reproducibility. For instance, acute changes in medium volumes (e.g. adding more medium to cells over the weekend, or reducing medium volumes during assays to save on expensive reagents) may lead to profound changes in experimental outcomes.”

Response to Arbitrating Advisor comments

We thank both Reviewers for their positive comments. Both reviewers pointed out additional references to add to our manuscript. Specifically:

Arbitrating Advisor #1: I am pointing out a study (PMID: 25114222) using a hypoxia-responsive element reporter, which indicates some hypoxia (ie, detectable by an HRE-GFP reporter) exists under 21% oxygen culture condition. Another paper (, see Figure 1D, x-axis), which the authors could consider adding to the discussion to further emphasize their point and enrich the literature with supportive evidence for the concept the authors wish to underscore.

Arbitrating Advisor #2: While there is a substantial amount of work presented in this manuscript, it aligns closely with what we've been practicing in the lab for years. Back in 2008, we published an article in the Journal of Cellular Physiology titled "Activation of HIF-1a in exponentially growing cells via hypoxic stimulation is independent of the Akt/mTOR pathway." Our research demonstrated that despite cells being incubated in a 21% environment, the actual oxygen levels can range from 2 to 3% after 24 to 48 hours, depending on cell confluency.

- We have now included these references on p. 14 (lines 501-506) in our discussion of the relevance of our findings to proliferative cell types.

“For example, studies in proliferative cell types have reported increased HIF1 α activity and stabilisation under standard culture conditions, particularly when grown at higher densities (Dayan et al, 2009; Le et al, 2014). Since cell density is a key determinant of oxygen consumption at the cell monolayer (Figure EV1H and EV1I) (Dayan et al, 2009), whether specific cells in culture experience hypoxia and would functionally benefit from increased oxygen will depend on both cell confluency during experiments and the oxygen use of the specific cell type.”

Arbitrating Advisor #2 also requested that we include “*HIF-1a Western blot data.*”

- As mentioned in email correspondence with Daniel Kilimbeck, these data are already included in the submitted manuscript in figures 3B and 3C.

Formatting and Editing requests:

>> *Author Contributions: Please remove the author contributions information from the manuscript text. Note that CRediT has replaced the traditional author contributions section as of now because it offers a systematic machine-readable author contributions format that allows for more effective research assessment. and use the free text boxes beneath each contributing author's name to add specific details on the author's contribution.*

- Author contributions have now been removed from the manuscript.

>> *Author Contributions: Please confirm corresponding sco-authorship (S.V., A.V-P.).*

- We confirm that S.V. and A. V-P. are corresponding co-authors.

>> >> *Introduce ORCID IDs for all corresponding authors (S.V., A.V-P.) via our online manuscript system. Please see below for additional information.*

- S.V. and A. V-P. have added their ORCID IDs.

>> *Please add a completed Author Checklist to your manuscript.*

- We include a completed Author Checklist in our re-submission.

>> *Figures: Main figures and supplementary figures should be removed from the manuscript text and uploaded as individual, high-resolution figure files. The supplementary figures should be renamed "Figure EV1" to EV5. The legends should remain in the manuscript text, legends of the suppl. figures should be after the main figure legends under the heading "Expanded View Figure Legends".*

- We have uploaded the figures as individual files. The supplemental figures are renamed as requested, and the legends included under the heading *Expanded View Figure Legends* .

>> *Funding: The "Rights and Retention Statement" should be merged with the funding info in the Acknowledgements; the sentence about the cc licence is not needed.*

- We have removed the rights retention statement.

>> *Dataset EV Legends: Tables S1-S6 should be renamed Dataset EV1-6 and their callouts need to be updated; the list with their short descriptions should be removed from the manuscript text as these should only be in the excel files themselves.*

- We have renamed the tables and callouts. The short description of the tables have been removed from the main manuscript text.

>> *Revisit publication status of the bioRxiv references Rogers et al (2023), Sergushichev et al (2016) and update in case of formal journal publication.*

- We have revisited these. Both are still preprints.

>> *Data availability section: please privacy from your data sets; move section to the end of Materials & Methods.*

- The data availability section has been moved to the end of the Material and Methods section. The gene expression data is now public. The metabolomics data is delayed being made public, as mentioned to Daniel Klimmeck in email correspondence.

>> *The note about BioRender should be moved from the Acknowledgements to a Graphics section that should be part of Materials & Methods, using the format "(some of the... OR Figure #... OR synopsis) Graphics were created with BioRender.com."*

- We have moved this comment about BioRender to a Graphics section in the Materials and Methods section.
- *We are applying a Structured Methods format for the Materials and Methods of articles published at EMBO Press, which have a string focus on methods advances. Adhering to this format is optional for research articles. However, considering the strong methodological aspect of your study, we would strongly encourage you to use it. Specifically, the Material and Methods section should include a Reagents and Tools Table (listing key reagents, experimental models, software and relevant equipment and including their sources and relevant identifiers) followed by a Methods and Protocols section in which we encourage the authors to describe their methods using a step-by-step protocol format with bullet points. More information on how to adhere to this format as well as downloadable templates (.doc or .xls) for the Reagents and Tools Table can be found in the author guidelines of our sister journal Molecular Systems Biology <https://www.embopress.org/page/journal/17444292/authorguide#methodguide>. An example of a paper with Structured Methods can be found here: <https://www.embopress.org/doi/full/10.15252/emj.2018100300>. We encourage you to be even more explicit in adding details on the experimental procedures, as this should be valuable in ensuring reproducible application if the approach.*
- We have edited our methods to include a bullet point Structured Methods format.
- *Revisit annotation of the Western blot analysis of HIF-1alpha (Fig3B,C; adv#2).*
 - We have not revised or added data to figures 3b and 3C, as our annotation is correct. As mentioned in email correspondence with Daniel Kilimmeck, HIF-1alpha blotting data was already included in the submitted manuscript in figures 3B and 3C.
- *Amend literature references and discussion of the context along the comments made by the advisors.*
 - We have included the references mentioned by the Advisors on p. 14 in our discussion on the relevance of our findings to proliferative cell types.

Additional edits by our production team:

-Data Availability Section:

1. Please note that the specific URLs for E-MTAB-12298 and E-MTAB-12299 datasets are not provided in the data availability statement.

- These URLs are now provided in the data availability statement.

2. Please note that reviewer access code for MTBLS6677 dataset is not provided in the data availability statement.

- Please note that we are waiting for the MTBLS6677 dataset to be made public. We are in correspondence with Metabolights to arrange this as a matter of urgency.

- Figure legends:

1. Please note that a separate 'Data Information' section is required in the legends of figures 1b, d, g-h; 2a-e, g; 3a, d; 4a-f, h-i; 5a-g, i-k, supplementary figures 1b, d-i; 3a-d; 4a-c; e-f; 5b-l.

2. Please note that the figure 4c, supplementary figures 5a, does not contain any statistical parameter, kindly rectify the statistical test related information in the figure legends appropriately.

3. Please indicate the statistical test used for data analysis in the legends of figure 3e, supplementary figure 2b.

4. Please note that in figures 4a-b, d-f, h-i, supplementary figures 3a-d, g; there is a mismatch between the annotated p values in the figure legend and the annotated p values in the figure file that should be corrected.

5. Please note that the error bars are not defined in the legends of supplementary figures 2a, c.

- All figure legends have been edited and updated as requested.

Dear Dr Fazakerley,

Thank you for submitting the revised version of your manuscript. I have now evaluated your amended manuscript and concluded that the remaining minor concerns have been sufficiently addressed.

I am pleased to inform you that your manuscript has been accepted for publication in the EMBO Journal.

On a different note, I would like to alert you that EMBO Press offers a format for a video-synopsis of work published with us, which essentially is a short, author-generated film explaining the core findings in hand drawings, and, as we believe, can be very useful to increase visibility of the work. Please see the following link for representative examples and their integration into the article web page:

<https://www.embopress.org/doi/full/10.15252/emboj.2019103932>

Best regards,

Daniel Klimmeck

Daniel Klimmeck, PhD
Senior Editor
The EMBO Journal
EMBO
Postfach 1022-40
Meyerhofstrasse 1
D-69117 Heidelberg
contact@embojournal.org
Submit at: <http://emboj.msubmit.net>
